# Whole-genome ancestry of an Old Kingdom Egyptian

Adeline Morez Jacobs[1,2,12✉], Joel D. Irish[1], Ashley Cooke[3], Kyriaki Anastasiadou[2], Christopher Barrington[4], Alexandre Gilardet[2,5], Monica Kelly[2], Marina Silva[2], Leo Speidel[2,6,13], Frankie Tait[2,14], Mia Williams[2], Nicolas Brucato[7], Francois-Xavier Ricaut[7], Caroline Wilkinson[8], Richard Madgwick[9], Emily Holt[9], Alexandra J. Nederbragt[10], Edward Inglis[10], Mateja Hajdinjak[2], Pontus Skoglund[2,15✉] & Linus Girdland-Flink[1,11,15✉]

Ancient Egyptian society flourished for millennia, reaching its peak during the Dynastic Period (approximately 3150–30 BCE). However, owing to poor DNA preservation, questions about regional interconnectivity over time have not been addressed because whole-genome sequencing has not yet been possible. Here we sequenced a 2× coverage whole genome from an adult male Egyptian excavated at Nuwayrat (Nuerat, نويرات). Radiocarbon dated to 2855–2570 cal. BCE, he lived a few centuries after Egyptian unification, bridging the Early Dynastic and Old Kingdom periods. The body was interred in a ceramic pot within a rock-cut tomb[1], potentially contributing to the DNA preservation. Most of his genome is best represented by North African Neolithic ancestry, among available sources at present. Yet approximately 20% of his genetic ancestry can be traced to genomes representing the eastern Fertile Crescent, including Mesopotamia and surrounding regions. This genetic affinity is similar to the ancestry appearing in Anatolia and the Levant during the Neolithic and Bronze Age[2–5]. Although more genomes are needed to fully understand the genomic diversity of early Egyptians, our results indicate that contacts between Egypt and the eastern Fertile Crescent were not limited to objects and imagery (such as domesticated animals and plants, as well as writing systems)[6–9] but also encompassed human migration.

For thousands of years, the Egyptian Dynastic civilization (approximately 3150–30 BCE) developed monumental architecture, sophisticated technology and relatively stable belief systems, becoming the longest-lasting civilization known. Following the political unification of the northern and southern regions of Egypt (Lower and Upper Egypt) at the end of the fourth millennium BCE, the Old Kingdom (2686–2125 BCE) witnessed considerable advances, including the construction of the first step pyramid complex of King Djoser and the 'Great Pyramid of Giza' built by King Khufu. The population has been considered to be of local origin, with limited input from neighbouring regions[8,10]. Yet, more recent archaeological evidence shows that trade connections existed across the Fertile Crescent since at least the sixth millennium BCE[7], if not earlier, with the advent of the Neolithic package (such as domesticated animals and plants)[6,7]. Cultural exchange continued to develop through the late fourth millennium BCE with the growing Sumerian civilization of Mesopotamia[7–9]. This period overlaps with the appearance of additional innovations in Egypt (such as the pottery wheel)[11] and the earliest evidence of hieroglyphic writing in the form of ivory tags in Tomb U-j at Abydos, dated 3320–3150 BCE[7].

Our knowledge of ancient Egyptians has increased through decades of bioarchaeological analyses[12–15], including dental morphological studies on their relatedness to other populations in North Africa and West Asia[16–18]. However, the lack of ancient genomes, particularly for the early periods of Egyptian Dynastic history, remains a barrier to our understanding of population continuity and gene flow in the region. Although individuals from ancient Egypt were subjected to the first effort to isolate ancient DNA[19], direct genome sequencing has remained elusive because of the challenging regional DNA preservation conditions. So far, only three individuals from Abusir el-Meleq (Fig. 1a) have yielded nuclear DNA, all post-dating the emergence of Dynastic Egypt by thousands of years (from 787 cal. BCE to 23 cal. CE)[20]. Moreover, these are not complete genome sequences but are limited to approximately 90,000–400,000 target-enriched genotypes. Over the millennia spanning the Dynastic Period, Egypt witnessed several wide-ranging wars, occupation by foreign rulers and dramatic episodes of internal political collapse (First, Second and Third Intermediate periods)[21]. Together, these processes may have substantially altered or reshaped the overall genetic structure and ancestry of the Egyptian population. Here we

[1]School of Biological and Environmental Sciences, Liverpool John Moores University, Liverpool, UK. [2]Ancient Genomics Laboratory, The Francis Crick Institute, London, UK. [3]World Museum, National Museums Liverpool, Liverpool, UK. [4]Bioinformatics and Biostatistics, The Francis Crick Institute, London, UK. [5]Centre for Palaeogenetics, Stockholm, Sweden. [6]Genetics Institute, University College London, London, UK. [7]Centre de Recherche sur la Biodiversité et l'Environnement (CRBE), Université de Toulouse, CNRS, IRD, Toulouse INP, Université Toulouse III–Paul Sabatier (UT3), Toulouse, France. [8]Face Lab, Liverpool John Moores University, Liverpool, UK. [9]School of History, Archaeology and Religion, Cardiff University, Cardiff, UK. [10]School of Earth and Environmental Sciences, Cardiff University, Cardiff, UK. [11]Department of Archaeology, School of Geosciences, University of Aberdeen, Aberdeen, UK. [12]Present address: Department of Biology, University of Padova, Padova, Italy. [13]Present address: iTHEMS, RIKEN, Wako, Japan. [14]Present address: Department of Archaeology, University of Reading, Reading, UK. [15]These authors jointly supervised this work: Pontus Skoglund, Linus Girdland-Flink. ✉e-mail: adelinemorez@gmail.com; pontus.skoglund@crick.ac.uk; linus.girdlandflink@abdn.ac.uk

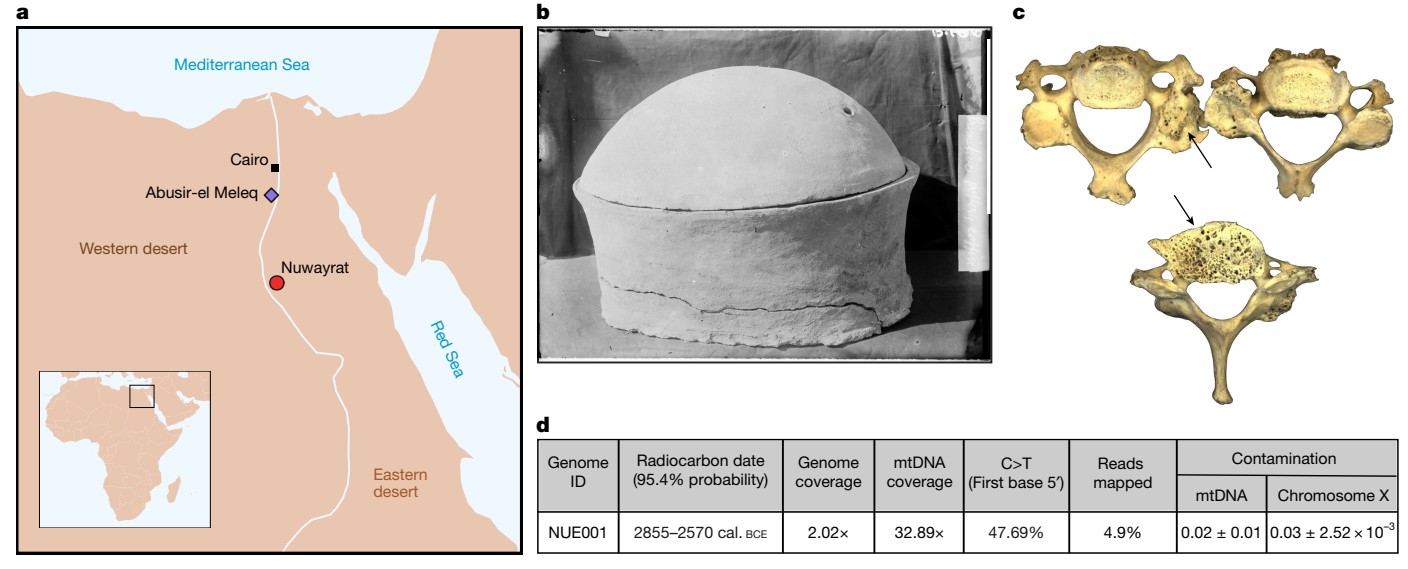

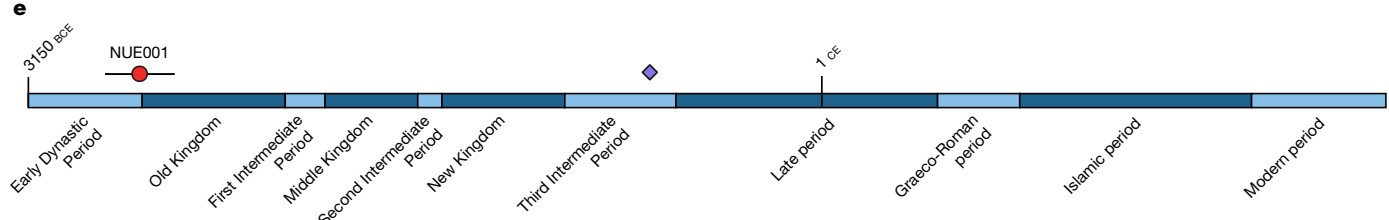

**Fig. 1 | Geographic location and date of the Nuwayrat individual in context.** **a**, Geographic location of the Nuwayrat cemetery (red dot) and the previously sequenced Third Intermediate Period individuals from Abusir el-Meleq[20] (purple diamond). **b**, Pottery vessel in which the Nuwayrat individual was discovered. **c**, Cervical vertebrae belonging to the Nuwayrat individual with evidence of extreme osteoarthritis (arrows). **d**, Summary of genomic and radiocarbon data. See the detailed breakdown of the quality indicators and calibration results for the three replicates and the combined date in Supplementary Table 2. **e**, Egyptian civilization timeline and radiocarbon date of the Nuwayrat and Third Intermediate Period individuals. mtDNA, mitochondrial DNA. Photo in **b** reproduced courtesy of the Garstang Museum of Archaeology, University of Liverpool.

present a whole-genome sequence of an ancient Egyptian individual (2.02× coverage; Supplementary Table 1), recovered from a necropolis at Nuwayrat (نويرات, Nuerat; Fig. 1a).

## The Nuwayrat individual

Nuwayrat is located near the village of Beni Hasan, 265 km south of Cairo (Fig. 1a). Radiocarbon dating of the skeletal remains showed that the Nuwayrat individual died between 2855 and 2570 cal. BCE (95.4% probability; Supplementary Information section 1 and Supplementary Table 2), which overlaps with the Early Dynastic and Old Kingdom periods (Fig. 1e). This result supports the initial archaeological assessments that material culture and funerary practices at the site were consistent with those of the Third and Fourth Dynasties of the Old Kingdom[1,22]. The body was placed in a large pottery vessel inside a rock-cut tomb (Fig. 1b and Extended Data Fig. 1). This treatment would have ordinarily been reserved for individuals of a higher social class relative to others at the site[23], as observed elsewhere during the Early Dynastic Period and at the Old Kingdom royal cemeteries near the city of Memphis (Supplementary Information section 1).

Although acknowledging known limitations in predicting phenotypic traits in understudied populations[24], the Nuwayrat individual is predicted to have had brown eyes, brown hair and skin pigmentation ranging from dark to black skin, with a lower probability of intermediate skin colour (Methods and Supplementary Table 10). The individual was genetically male (XY sex chromosomes; Supplementary Table 1), consistent with the expression of standard skeletal features[25] (Methods). Our further osteological examination revealed that he

would have stood 157.4–160.5 cm tall[26]. He lived to an advanced age for the time (approximately 44–64 years; the upper end of this range is the most probable[25,27]), as evidenced by his heavily worn teeth and age-related osteoarthritis in most joints and vertebrae, in some cases severe (Fig. 1c). This and various activity-induced musculoskeletal indicators of stress revealed that he experienced an extended period of physical labour, seemingly in contrast to his high-status tomb burial. The patterns of osteoarthritis and stress indicators further imply the form of physical activity that he routinely engaged in, which some researchers maintain can provide clues concerning occupation[28,29]. In this case, although circumstantial, they are not inconsistent with those of a potter, as depicted in ancient Egyptian imagery. Estimates of biological affinity based on dental morphological features and cranial measurements parallel the genomic results (below). More detailed information about the Nuwayrat individual is presented in Supplementary Information section 2, with a facial depiction in Supplementary Information section 3 (Extended Data Fig. 2).

Multi-isotope analysis ($\delta^{13}$C, $\delta^{15}$N, $\delta^{18}$O and $^{87}$Sr/$^{86}$Sr) was conducted on dental enamel and dental collagen from the lower-left second molar to determine his childhood diet and geographic origin (Supplementary Information section 5). All results are consistent with having grown up in the hot, dry climate of the Nile Valley ($\delta^{18}$O$_{carb\,VSMOW}$ = 23.6‰, where VSMOW indicates Vienna Standard Mean Ocean Water; $^{87}$Sr/$^{86}$Sr = 0.707888)[30–32] and consuming an omnivorous diet based on terrestrial animal protein and plants, such as wheat and barley ($\delta^{13}$C$_{VPDB}$ = −19.6‰, where VPDB indicates Vienna Pee Dee Belemnite; $\delta^{15}$N$_{AIR}$ = 12.3‰)[33], typical for Egyptians until the Coptic period[34]. An elevated $\delta^{15}$N value, frequently observed in isotope studies of

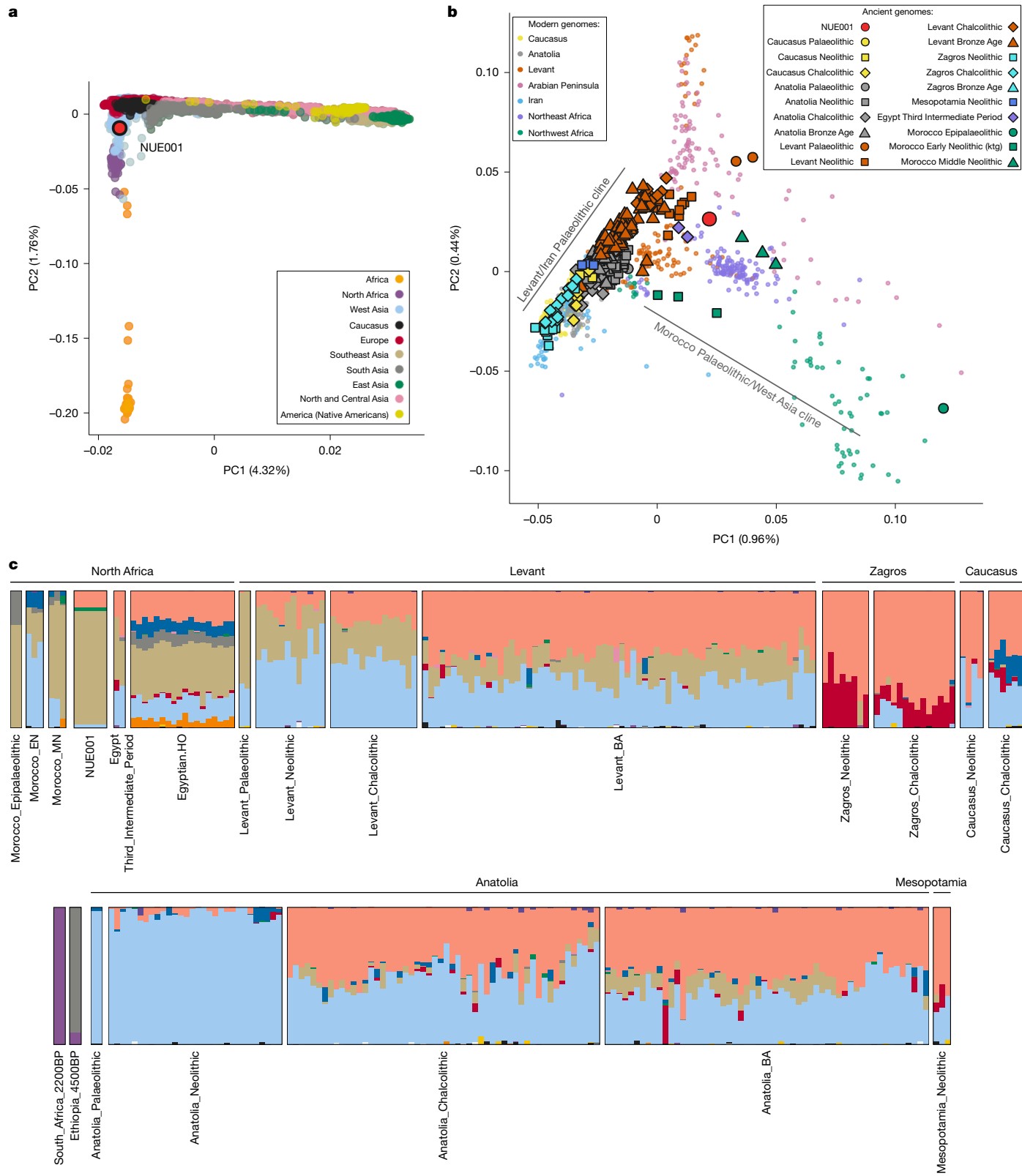

**Fig. 2 | Genetic ancestry of the Nuwayrat genome. a**, PCA of present-day worldwide populations, with projection of the Old Kingdom Egyptian genome from Nuwayrat (NUE001). **b**, PCA of present-day populations from North Africa and West Asia, with projection of ancient North African and West Asian genomes. **c**, ADMIXTURE clustering analysis of the Old Kingdom Egyptian genome in the context of ancient African, West Asian and present-day Egyptian genomes at $K = 14$ ancestral populations. Only a subset of genomes corresponding to those used in the qpAdm analysis (Fig. 3) are displayed. The full output of the ADMIXTURE analysis is shown in Extended Data Figs. 4 and 5.

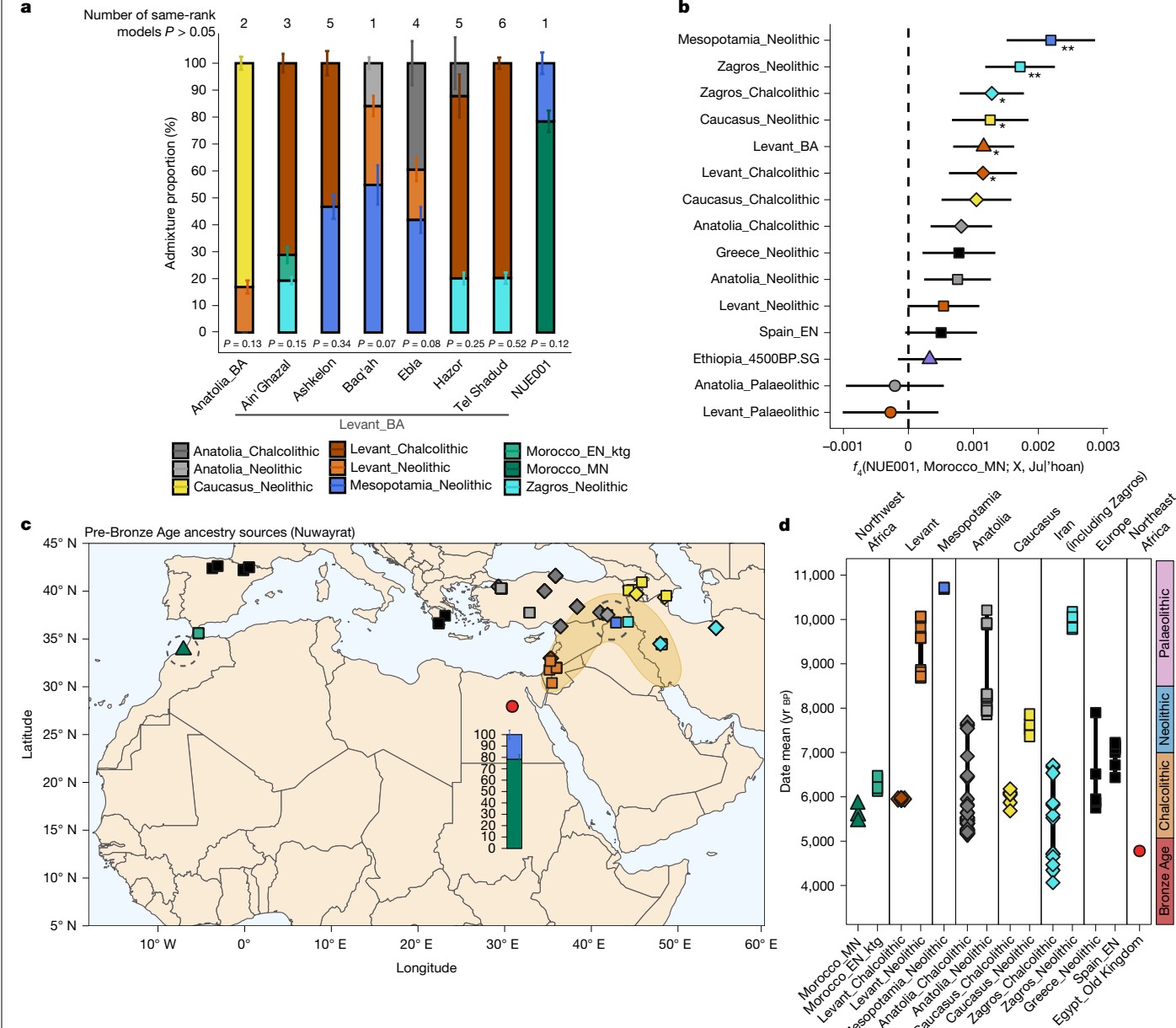

**Fig. 3 | Ancestry models of the Nuwayrat genome. a**, Ancestry proportion of Nuwayrat and comparative Bronze Age Levantine and Anatolian genomes for the best-fit full model (qpAdm). Alternative same-rank models passing P > 0.05 with a lower P value are shown in Supplementary Table 6. Values represent best-fitting model estimates ± 1 standard error. This analysis was conducted over n = 537,543 SNPs for NUE001, n = 518,994 SNPs for Anatolia_BA, n = 554,622 SNPs for Ain' Ghazal, n = 493,274 SNPs for Ashkelon, n = 578,969 SNPs for Baq'ah, n = 574,452 SNPs for Ebla, n = 552,505 SNPs for Hazor and n = 513,561 SNPs for Tel Shaddud. **b**, Estimation of the best source responsible for

deviation of the Nuwayrat genome from the Middle Neolithic Morocco group as $f_4$(NUE001, Morocco_MN; Mesopotamia_N, Ju|'hoan). Symbols represent $f_4$ value ± 1 standard error. *Z score > 2; **Z score > 3. The analysis was conducted over 280,544 SNPs. **c,d**, Map (**c**) and timeline (**d**) of rotating sources used to infer the proximal ancestry of the Nuwayrat and Bronze Age Levantine and Anatolian genomes (shown in **a**), with the dark-yellow area corresponding to the Fertile Crescent. The timeline in **d** is based on Egyptian cultural transition dates.

ancient Egyptians, may have been caused by the arid environment[35–37], eating foods raised on manured fields[38] and/or inclusion of Nile fish in the diet[34].

## Ancient genome sequencing

Seven cementum-enriched DNA extracts were prepared into single-stranded DNA sequencing libraries[39] and screened on an Illumina platform. Five of these libraries showed degradation patterns expected for ancient DNA with evidence of elevated rates of cytosine-to-thymine

substitutions at the first base of the sequence alignments (more than 30%) and low contamination estimates for both nuclear and mitochondrial DNA (0–3%; Fig. 1d); the two remaining libraries were discarded because of elevated contamination estimates (Extended Data Fig. 3 and Supplementary Table 1 (Y11473 and Y11476)). The two libraries (Y11475 and Y11477) with the highest proportion of reads mapping to the reference human genome (6.0% and 3.2% of all sequences) were further sequenced on Illumina NovaSeq 6000 and NovaSeq X platforms to generate a total of 8.3 billion 2 × 100 sequence read pairs.

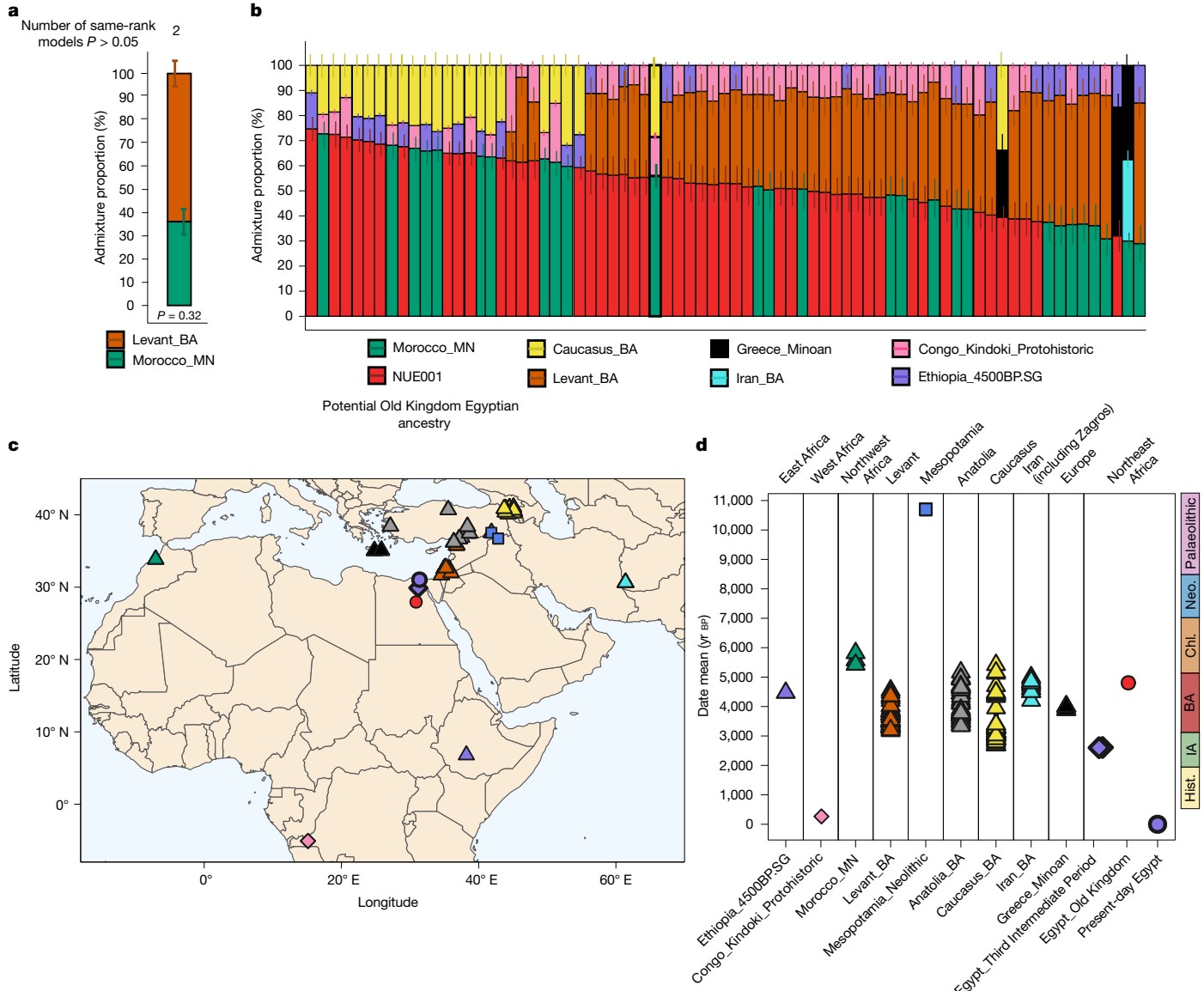

**Fig. 4 | Ancestry models of later Egyptians. a**, Ancestry proportions of the Third Intermediate Period genomes for the best-fit model (qpAdm). Alternative two-source and three-source models passing $P > 0.05$ are reported in Supplementary Table 7. This analysis was conducted over 290,262 SNPs. **b**, Ancestry proportions of the present-day Egyptian genomes for the best-fit model (qpAdm). For **b**, alternative two-source and three-source models passing

$P > 0.05$ are reported in Supplementary Table 8. This analysis was conducted over 767,305 SNPs. Values represent best-fitting model estimates ± 1 standard error (**a**,**b**). **c**,**d**, Map (**c**) and timeline (**d**) of rotating sources used to infer the ancestry of the two Third Intermediate Period and/or the present-day Egyptian genomes. The timeline in **d** is based on Egyptian cultural transition dates. BA, Bronze Age; Chl., Chalcolithic; Hist., historical period; IA, Iron Age; Neo., Neolithic.

We merged the Nuwayrat genome (NUE001) with those of 3,233 present-day individuals that were either whole-genome sequenced or genotyped on the Human Origins Array and 805 ancient individuals with either whole-genome or 1.2 million single nucleotide polymorphism (SNP) capture data. We first projected the Nuwayrat genome in a principal component analysis (PCA) using a population panel representing present-day worldwide genetic diversity. The Nuwayrat individual is genetically most similar to present-day people in North Africa and West Asia (Fig. 2a and Extended Data Fig. 4), which is consistent with the results from ADMIXTURE clustering[40] (Extended Data Fig. 5). The mitochondrial DNA (haplogroup I/N1a1b2) and chromosome Y (haplogroup E1b1b1b2b-) haplogroups of the Nuwayrat individual are most common in present-day North African and West Asian groups (Supplementary Table 4), consistent with the whole-genome affinities. Furthermore, the Nuwayrat genome had no extended runs of homozygosity above 4 cM, indicating no recent consanguinity in his ancestry[41].

## Ancestry of the Nuwayrat genome

We used the qpAdm[42] framework to model the genetic ancestry components that best represent the Nuwayrat genome using a fully rotating model competition approach, in which a set of candidate populations are iteratively used as sources to construct one-source, two-source and three-source population ancestry models, whereas the remaining candidates are set as outgroup (right) populations[43,44] (Supplementary Information section 4). We used a set of 13 populations from Neolithic and Chalcolithic West Asia, North Africa and the North Mediterranean region that predate the Nuwayrat individual as potential sources (Fig. 3c,d and Methods). No single-source model fitted the data (maximum $P$ value observed $= 2.39 \times 10^{-6}$ for a model with Morocco_MN as a single source). Instead, a single two-source model ($P = 0.12$) met the significance criteria ($P > 0.05$), which consisted of a mixture of 77.6 ± 3.8% ancestry represented by genomes from the Middle

Neolithic Moroccan site of Skhirat-Rouazi dated to 4780–4230 BCE (Morocco_MN), and the remainder most closely related to genomes from 9000 to 8000 BCE Neolithic Mesopotamia (22.4 ± 3.8%; Fig. 3a). In addition, two three-source models showed similar ancestry proportions but with a minor contribution of a third ancestry component represented by genomes from the Neolithic/Chalcolithic Levant (4.7 ± 8.2% at $P$ = 0.11 and 1.1 ± 8.7% at $P$ = 0.07, respectively; Supplementary Table 6).

All accepted qpAdm models showed the presence of ancestry related to Middle Neolithic Morocco in the Nuwayrat genome; therefore, our results could indicate shared ancestry across North Africa during this period and, consequently, that local Egyptian Neolithic populations contributed genetically to the Early Dynastic and Old Kingdom people, as indicated from material culture[7,8,10] and bioarchaeological analyses[14,15]. However, because the genomes from Middle Neolithic Morocco have previously been modelled to comprise both Iberomaurusian-like and Levantine Neolithic ancestry components[45], which we corroborated (Extended Data Fig. 6, Supplementary Information section 4 and Supplementary Table 5), the affinity to Levantine Neolithic groups could reflect several migration events. To explore these alternative hypotheses in detail, further ancient DNA studies on pre-Bronze Age genomes from North Africa are required.

The second genetic ancestry component detected in the Nuwayrat individual is most closely related to Neolithic Mesopotamians, out of the potential sources included in the model competition (Methods). To further examine the putative affinity to Neolithic Mesopotamia, we computed a series of $f_4$ statistics testing whether a set of groups share more derived alleles with Nuwayrat than with Middle Neolithic Morocco in the form $f_4$(NUE001, Morocco_MN; X, Ju_hoan_North.DG); here X represents the rotating sources of the qpAdm model with the addition of Levantine Palaeolithic, Anatolian Palaeolithic and Bronze Age Levantine genomes. The statistic was maximized and statistically significant with Neolithic Mesopotamia as X ($Z$ score = 3.2; Fig. 3b). The affinity is also seen in the statistic $f_4$(NUE001, Morocco_MN; Mesopotamia_N, X), which is positive for all tested populations as X, consistent with an ancestry affinity between the individual from Nuwayrat and Neolithic Mesopotamia, with $Z$ scores > 2 for all, except Zagros and Caucasus groups and Chalcolithic and Bronze Age Levantine groups (Supplementary Table 9).

Although we caution that these results are based on a single Egyptian genome, they mirror another study that found evidence of gene flow from the Mesopotamian and Zagros regions into surrounding areas, including Anatolia, during the Neolithic[2]. Together with archaeological evidence for cultural exchange[6,7], these findings open the possibility that wider cultural and demographic expansion originating in the Mesopotamian region reached both Egypt and Anatolia during this period. However, more recent migrations from the eastern Fertile Crescent during the Chalcolithic and Bronze Age further altered the Anatolian and Levantine genetic landscapes[3–5]. Related movements may have introduced the Mesopotamian-like ancestry more recently in Egypt. We tested this by applying the same full qpAdm model to target groups from the Bronze Age Anatolia and Levant (genomes from the Bronze Age Levant were grouped into eight archaeological sites[3–5,46,47], of which all models for Megiddo and Yehud were rejected; Supplementary Table 6). Although we replicated previous findings that all Levantine Bronze Age groups trace 18.7–79.8% of their ancestry to Neolithic or Chalcolithic Levantine groups[3–5,46,47], we also detected ancestry from Neolithic Mesopotamia at three sites (Ebla, Baq'ah and Ashkelon), considering the best-fit models, in proportions (41.8–54.8%) exceeding those in the Nuwayrat genome (Fig. 3a and Supplementary Table 6). However, the initial full qpAdm model, extended to include the Bronze Age Levant as a potential source, can effectively be rejected for the Nuwayrat genome ($P$ = 0.013; Supplementary Information section 4). Notably, the best model for the Nuwayrat genome fits worse when these groups are included as reference groups ($P$ = 0.021; Supplementary Information section 4). This means that we cannot exclude the possibility that the Neolithic

Mesopotamian-like ancestry in the Nuwayrat genome could have arrived by means of more recent unsampled intermediaries in the Levant.

Although the timing of the admixture event cannot be estimated directly (Supplementary Table 11 and Supplementary Note 4), this finding provides direct evidence of genetic ancestry related to the eastern Fertile Crescent in ancient Egypt. Archaeological evidence lends support to the Early Neolithic shared regional ancestry between Egypt and West Asia. Given its proximity, Egypt was one of the first external areas to adopt the Neolithic package that emerged across West Asia as early as the sixth millennium BCE or before[6,48,49], which could have corresponded with movements of people. This period is concomitant with the observed gene flow from Mesopotamia to Anatolia[2], which may have expanded into Egypt as well. In support, a substantial change in odontometric and dental tissue proportions occurred approximately 6000 BCE in the Nile Valley, with general continuity thereafter[50]. Along with marked temporal differences in subsistence (such as domesticated plants and greater sedentism) and material culture (such as the introduction of pottery), this is indicative of discontinuity between the Mesolithic (eighth to seventh millennium BCE) and Neolithic populations[50]. Cultural exchange and trade then continued through the fourth millennium BCE when Mesopotamian Late Uruk period features filtered into the Nile Valley during the later Predynastic Period[7–9,51]. Trade might have been routed through the Mediterranean and Red Seas rather than the Sinai Desert[7,52]. Such seaborne mobility could explain a scenario in which the source population did not come into contact with the Chalcolithic/Bronze Age Levantines. Our results indicate that this millennia-long process might not have only included cultural transmission but also migration and subsequent admixture.

Moreover, it is notable that both our qpAdm modelling and ADMIXTURE clustering excluded any substantial ancestry in the Nuwayrat genome related to the 4,500-year-old genome from Mota, Ethiopia or other individuals in central, eastern or southern Africa (Figs. 2 and 3 and Extended Data Fig. 5)[53]. Nevertheless, we found that the Nuwayrat genome fits as an equally good source as Levant Chalcolithic groups for the West Eurasian-related component of East African pastoralist genomes, but ancient DNA data are still missing for many putative source regions (Extended Data Fig. 7, Supplementary Information section 4 and Supplementary Table 12)[54,55].

## Ancestry in later Egypt

The Nuwayrat genome extends the genetic record of ancient Egypt beyond previously published data from the Third Intermediate Period (787–544 BCE; Fig. 1e). We modelled these latter individuals[20] using qpAdm with putative sources from a set of nine populations from North Africa (including Nuwayrat), West Asia and Greece, who lived between the Old Kingdom and the Third Intermediate Period and also included genomes from the Middle Neolithic Morocco and Neolithic Mesopotamia (see Supplementary Information section 4 for more models that tested the potential overfitting of these sources). We can reject all one-source models, including one with 100% continuity from Nuwayrat to the Third Intermediate Period ($P$ = 3.00 × 10$^{-7}$). Two similar two-source models fit the data (Fig. 4a and Supplementary Table 7), differing only in whether the Nuwayrat or Middle Neolithic Moroccan individuals are one of the best-fit sources. In both models, the main source of ancestry is the Bronze Age Levant (for example, 64.5 ± 5.6% in the model with Middle Neolithic Morocco; $P$ = 0.32; Supplementary Information section 4). These results are consistent with the Third Intermediate Period genomes deriving part of their ancestry from local groups related to the Nuwayrat individual while evidencing a significant increase in Levantine ancestry.

Evidence of gene flow from the Levant by the time of the Third Intermediate Period could be linked to the proposed Bronze Age Canaanite expansion, starting at the end of the Middle Kingdom period. On the basis of archaeological findings, whether this was a gradual

assimilation process[56] or a rapid shift, such as the settlement of Hyksos rulers[56,57], is still debated. Overall, this period also overlaps with the well-characterized Late Bronze Age collapse that witnessed rapid societal and economic upheaval across the Mediterranean region, leading to or being caused by widespread population movements[58,59]. However, the temporal and geographical limitations of the current genomic data do not allow firm conclusions to be drawn.

We next tested how present-day Egyptian ancestry could be traced to the Bronze Age populations living in North Africa, including the Nuwayrat individual, West Asia, Europe and sub-Saharan Africa, using qpAdm. Despite substantial heterogeneity, most present-day Egyptian genomes can be modelled as deriving their ancestry from five sources related to (1) Nuwayrat (32.1–74.7%); (2) Middle Neolithic Morocco (28.9–72.7%); (3) Bronze Age Levant (11.6–57.1%); (4) the 4,500-year-old individual from Ethiopia ('Mota') (7.4–56.0%); and (5) two approximately 230-year-old individuals from Congo (4.8–52.0%) (Fig. 4b and Supplementary Table 8). Thus, if tracing the ancestry of many present-day Egyptians in our study to the Bronze Age, much of it would be found in groups related to Nuwayrat or alternatively to sources best represented by Middle Neolithic Morocco from which approximately 80% of Nuwayrat's ancestry derives. The second most common ancestry component is related to the Bronze Age Levant, consistent with the ancestry detected in the Third Intermediate Period individuals. Bronze Age Caucasus ancestry is present in a fraction of the present-day Egyptians but is similar to the Bronze Age Levant ancestry[60]. Our models show a more recent arrival of East and West African ancestries in present-day Egyptians, which has also been previously suggested[20] and dated to 27 generations ago using linkage disequilibrium-based admixture dating[61]. Moreover, we note that there is a substantial diversity in ancestry across Egypt; approximately 20% of the present-day Egyptian genomes included here did not fit the model described above.

## Conclusions

Our results demonstrate the feasibility of ancient genome sequencing from the earliest stages of the Egyptian Dynastic civilization. One possible explanation for the successful whole-genome retrieval is the pot burial, which may have favoured a degree of DNA preservation not previously reported in Egypt. This contributes to the road map for future research to obtain ancient DNA from Egypt[62]. Although our analyses are limited to a single Egyptian individual who, on the basis of his relatively high-status burial, may not be representative of the general population, our results revealed ancestry links to earlier North African groups and populations of the eastern Fertile Crescent. Analogous links were indicated in our biological affinity analyses of dental traits and craniometrics of the Nuwayrat individual, as well as in previous morphological studies based on full samples. The genetic links with the eastern Fertile Crescent also mirror previously documented cultural diffusion (such as domesticated plants and animals, writing systems and the pottery wheel), opening up the possibility of some settlement of people in Egypt during one or more of these periods. The Nuwayrat genome also allowed us to investigate the Bronze Age roots of ancestry in later Egypt, highlighting the interplay between population movement and continuity in the region. Future whole-genome sequencing of DNA from more individuals will allow for a more detailed and nuanced understanding of ancient Egyptian civilization and its inhabitants.

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

## Methods

### Provenance and ethics

The human remains were excavated from the Nuwayrat necropolis near Beni Hasan, Egypt. They were donated between 1902 and 1904 by the Egyptian Antiquities Service to the members of the Beni Hasan excavation committee and subsequently donated to the Institute of Archaeology, University of Liverpool and exported under the John Garstang export permit. The human remains were then donated to the World Museum (previously the Liverpool City Museum) in 1950. Sampling permit was granted by the World Museum.

### Ancient DNA extraction, library preparation and sequencing

Sampling and DNA extraction of seven permanent teeth belonging to an individual from Nuwayrat were carried out in dedicated ancient DNA facilities at Liverpool John Moores University. Library preparation and sequencing were carried out at The Francis Crick Institute (Supplementary Table 1). Before subsampling, the teeth were decontaminated by wiping with 1% sodium hypochlorite, followed by wiping with molecular biology grade water and ethanol. Approximately 44–66 mg of cementum-enriched powder was extracted from each tooth using a Dremel drill at the lowest possible rotations per minute (5,000 rpm).

DNA was extracted using 1 ml of extraction buffer consisting of 0.45 ml of 0.5 M EDTA (pH 8.0) and 10 μl of 10 mg ml$^{-1}$ proteinase K per 50 mg of bone powder. The mixture was incubated overnight (approximately 18 h) at 37 °C and purified on the High Pure Viral Nucleic Acid Large Volume Kit (Roche) using a binding buffer described in ref. 63 and QIAGEN buffer PE. DNA was eluted in approximately 100 μl of QIAGEN elution buffer.

Extracts were turned into single-stranded DNA libraries[39] (without treatment to remove uracils), double-indexed[64] and then underwent paired-end sequencing on a HiSeq 4000 to approximately seven million reads per library for initial screening (Supplementary Table 1). All samples were processed alongside negative lysate and extraction controls and positive and negative library controls. On the basis of the assessment of the initial sequencing results, two libraries were selected for extra rounds of deeper sequencing on the NovaSeq 6000 and NovaSeq X platforms, following the selection of fragments greater than 35 bp using polyacrylamide gel electrophoresis[65] for the library built from the NUE001b5e1 extract (Supplementary Table 1), with a resulting total of 8.3 billion 2 × 100 sequence pairs.

### Radiocarbon dating

New radiocarbon dating was generated for the individual that yielded DNA from Nuwayrat (NUE001) by Beta Analytic using accelerator mass spectrometry. We directly dated the upper-left third molar (NUE001b3) and lower-left first premolar (NUE001b5), both of which yielded DNA that was deep sequenced (Supplementary Table 1). The results are reported in Supplementary Table 2. The femur of this individual was previously radiocarbon dated[66,67] (Supplementary Information section 1 and Supplementary Table 2). All dates were calibrated using OxCal v.4.4.4 (ref. 68) with atmospheric data in IntCal20 (ref. 69). We also combined the three independent dates using the R_Combine() function in OxCal[70] (Supplementary Table 2). We rounded the calibrated dates outwards to 10 years unless error terms were smaller than ±25 bp, in which case we rounded outwards to 5 years[71].

### Isotope analysis

Dental collagen and enamel were extracted from the lower-left second molar. Dentine collagen was extracted for carbon ($\delta^{13}$C) and nitrogen ($\delta^{15}$N) isotope analysis following a modified Longin method[72,73]. Mass spectrometry was performed using a Flash 1112 series elemental analyser coupled with a Finnigan DELTA V Advantage (Thermo Fisher Scientific) using established protocols[74]. Analytical precision (1$\sigma$) of the in-house calibrated standards[74] were 0.08 and 0.07 for $\delta^{13}$C and $\delta^{15}$N, respectively.

For enamel, after surface abrasion, a slice (3.6 mm wide) was extracted, and all adhering dentine was removed. Two fragments were powdered, one of which was pre-ultrasonicated. A minimum of 3.0 mg was analysed for the oxygen isotope composition of enamel carbonate ($\delta^{18}$O$_c$). Samples were acidified for 5 min with more than 100% orthophosphoric acid (density approximately 1.9 g cm$^{-3}$) at 70 °C and analysed in duplicate using a MAT 253 dual-inlet mass spectrometer (Thermo Fisher Scientific) coupled to a Kiel IV carbonate preparation device using established protocols[74]. Isotope values are reported as per mille ($^{18}$O/$^{16}$O) normalized to the Vienna Pee Dee Belemnite (VPDB) scale using an in-house carbonate standard (BCT63) calibrated against NBS19. The long-term reproducibility for $\delta^{18}$O BCT63 is ±0.04‰ and ±0.03‰ for $\delta^{13}$C (1$\sigma$). The oxygen carbonate values ($\delta^{18}$O$_{C\,VPDB}$) were converted to the Vienna Standard Mean Ocean Water (VSMOW) scale[75] and phosphate ($\delta^{18}$O$_{P\,VSMOW}$)[76].

The remaining enamel fragment (56.3 mg) was cleaned in an ultrasonic bath, digested in 8 M HNO$_3$ and heated overnight at 120 °C. Sr-Spec was used for strontium extraction, following the revised version of Font et al.[77]. Once column-loaded in 1 ml of 8 M HNO$_3$, matrix elements were eluted in washes of 8 M HNO$_3$, and samples were placed on a hotplate (120 °C) overnight, with a repeat pass following. The sample was redissolved in 2% HNO$_3$, and the $^{87}$Sr/$^{86}$S ratio was measured using a Neoma multi-collector inductively coupled plasma mass spectrometry with tandem mass spectrometry (MC-ICP–MS/MS, Thermo Fisher Scientific). Instrumental mass bias was corrected for using the exponential law and a normalization ratio of 8.375209 for $^{88}$Sr/$^{86}$Sr (ref. 78). Residual krypton (Kr) and rubidium ($^{87}$Rb) interferences were monitored and corrected using $^{84}$Kr and $^{86}$Kr ($^{83}$Kr/$^{84}$Kr = 0.20175 and $^{83}$Kr/$^{86}$Kr = 0.66474; without normalization) and $^{85}$Rb ($^{85}$Rb/$^{87}$Rb = 2.5926), respectively. The accuracy of the method was assessed by measuring the EC-5 coral standard ($^{87}$Sr/$^{86}$Sr: 0.709171 ± 0.000016 (2$\sigma$; $n$ = 14), consistent with the expected value for seawater). The data were also corrected against a National Institute of Standards and Technology Standard Reference Material 987 value of 0.710248 (ref. 79). The procedural blank was less than 75 pg of Sr, negligible relative to sample Sr.

### Osteological analyses

Following element inventory, our determination of the Nuwayrat individual's sex was based on standard morphological indicators across the skeleton (protocol in Buikstra and Ubelaker[25]). Ageing was estimated from the dentition, cranium and postcrania[25,27,80–84]. For stature, several approaches were used[85–87], with the most likely estimate based on direct stature reconstruction of ancient Egyptians following ref. 26.

Biological affinity was assessed from two long-recognized methods: dental non-metric traits[88] and craniometrics (for example, Howells[89]). First, the rASUDAS application was accessed (https://osteomics.com/rASUDAS2/)[90]. It used up to 32 crown and root traits for comparison with data from seven global population samples. Second, the craniometric approach used the CRANID program CR6bIND, with 29 measurements for comparison with a database of 74 premodern through recent global samples, including Late Dynastic Egyptians and ancient West Asians[91].

Our recording and description of skeletal pathology, related primarily to age-related breakdown, follow accepted methods[25,92,93]. This and activity-induced musculoskeletal stress markers (details in previous studies[28,29,94]) were used to ascertain the level of physical activity. Although not without criticism[95,96], they have been used to infer occupation by identifying common positions and movements in life. For that purpose, the latter were compared with illustrations of individuals engaged in a range of common jobs, as depicted on ancient Egyptian tomb walls and in statuary (Supplementary Information section 2).

### Facial reconstruction and depiction

Craniofacial analysis and facial reconstruction from skeletal remains were carried out using three-dimensional laser scan data of the skull (collected using an Artec Space Spider scanner), Touch X haptic device

and Geomagic Freeform software[97]. Egyptian male data[98] were used to estimate facial tissues at anatomical points across the skull surface. The muscles of the head and neck were imported from the Face Lab database and remodelled to fit the skull following anatomical guidelines[99]. Morphometric standards were used[99,100] to estimate facial feature morphology, such as eye and nasal shape, lip and ear pattern and structural creases. A final facial depiction was produced using two-dimensional photo-editing software. It is important not to consider a facial depiction as a portrait or definitive image because it can only visualize the available information[101]. In this case, although DNA analysis indicated the most probable population of origin, there was no evidence in relation to skin colour and hair colour. Therefore, the facial depiction was produced in black and white without head hair or facial hair (Supplementary Information section 3).

## Bioinformatics data processing and authentication

Read alignment was performed following the pipeline in the study of Swali et al.[102]. Samples were processed through the nf-core/eager v.2.3.3 pipeline[103]. First, adaptors were removed, paired-end reads were merged and bases with a quality below 20 were trimmed using AdapterRemoval v.2.3.1 (ref. 104) with −trimns −trimqualities −collapse −minadapteroverlap 1 and −preserve5p. Merged reads with a minimum length of 35 bp were mapped to the hs37d5 human reference genome with Burrows-Wheeler Aligner (BWA-0.7.17 aln)[105] using -l 16500 -n 0.01 -o 2 -t 1 (ref. 106). Duplicate reads were removed using DeDup v.0.12.8 (ref. 107). Finally, we removed the alignments with mapping quality below 30 and containing indels.

We used mapDamage v.2 (ref. 108) to visualize the substitution distribution along the reads and evidence the presence of deaminated molecules typical of ancient DNA. Contamination was estimated using three different data sources: (1) genome-wide present-day contamination using the conditional substitution rate[109] computed using PMDtools v.0.60 (ref. 110); (2) present-day mitochondrial DNA-based contamination using schmutzi (commit be61017)[111]; and (3) chromosome X contamination on libraries assigned as male using ANGSD v.0.933 (ref. 112), restricted to the non-recombining region of chromosome X. All the libraries from NUE001 show little to no contamination, except two libraries with sequencing identification numbers SKO719A1706 and SKO719A1709 (Extended Data Fig. 3 and Supplementary Table 1).

## Molecular sexing

The biological sex of the sequenced individual was determined using the $R_y$ parameter[113], which is the ratio of the number of alignments to the Y chromosome ($n_y$) to the total number of alignments to both sex chromosomes ($n_x + n_y$), $R_y = n_y/(n_x + n_y)$. All libraries are consistent with NUE001 being karyotypically male, except the results from the library SKO719A1706 consistent with being female, which is probably a result of contamination (Supplementary Table 1).

## SNP calling in the Nuwayrat individual

We merged the sequencing data from five libraries from the Nuwayrat individual showing an absence of present-day human DNA contamination, yielding a total of 135,606,409 mapped unique reads of 44.63 bp on average, resulting in an average genome-wide coverage of 2.02×. We called pseudo-haploid positions using SAMTools v.1.9 mpileup[114] with options -B -R -Q30 and SequenceTools 1.5 (ref. 115) with options −randomHaploid and −singleStrandMode. This approach leverages the single-stranded library preparation to computationally remove the effects of cytosine-deamination-derived sequence errors. Specifically, at C/T SNPs, it removes all bases that are aligned onto the forward strand; at G/A SNPs, it removes all bases on that aligned to the reverse strand. This allows for a confident pseudo-haploid genotyping even also at CpG context transitions, which are mostly not repaired by the uracil-DNA glycosylase (UDG) treatment owing to methylation[116].

## Uniparental marker determination

We obtained the mitochondrial DNA consensus of the Nuwayrat individual from endogenous reads, removing the bases with quality below 20 (-q 20) using schmutzi[111]. The mitochondrial haplogroup was assigned using Haplogrep 3 (ref. 117).

The chromosome Y haplogroup was obtained using pathPhynder[118] with the parameter -m 'no-filter', on the basis of approximately 120,000 SNPs extracted from worldwide present-day and ancient male chromosome Y variation and the International Society of Genetic Genealogy v.15.73 (http://www.isogg.org).

## Comparison dataset

We merged the genome of the Nuwayrat individual with a comparison dataset of 977 ancient individuals[2–5,20,43,45–47,53–55,60,119–160] and 4,040 modern individuals[43,46,122,141,158,161–169] genotyped on either the Human Origins array[169] ('Human Origins' dataset) or the 1.2 million SNP array ('1240k' dataset)[139] (Supplementary Table 3). Most genotypes were directly accessed from the Allen Ancient DNA Resource v.54.1 (ref. 170). We added nine ancient genomes from Morocco[45] and 13 ancient genomes from Mesopotamia[120] from raw mapped Binary Alignment Map (BAM) files processed following the above-mentioned bioinformatic pipeline, with two modifications: (1) for the double-stranded UDG-treated genomes from ref. 45, we trimmed the first and last three bases of the reads and then called pseudo-haploid genotypes at both transition and transversion sites; and (2) for the non-UDG-treated genomes from ref. 120, we called pseudo-haploid genotypes at transversion sites only. We included 100 present-day Egyptian genomes from ref. 164 in both datasets. Individuals related up to the second degree, as detected in previous studies, were excluded.

## Principal component analysis

We computed two PCA on present-day individuals from the 'Human Origins' dataset using 593,124 substitutions through SMARTPCA (eigensoft v.6.1.4)[169]. For the first analysis, we kept 3,233 individuals from across the world and projected NUE001 on the resulting components. For the second PCA, we kept 722 present-day individuals from North Africa, West Asia and the Caucasus and projected NUE001 together with 781 ancient genomes from North Africa, West Asia and the Caucasus. Both analyses used transversions only (111,208 SNPs).

## ADMIXTURE clustering

We used a model-based clustering approach from the program ADMIXTURE v.1.2 (ref. 40) to estimate the ancestry components from genomes in the 'Human Origins' dataset. All genomes were transformed into pseudo-haploid sequences, and transitions were removed. The remaining 111,208 positions were subsequently pruned for SNPs in strong linkage disequilibrium using PLINK v.1.9 (ref. 171), with the parameter −indep-pairwise 200 25 0.4 to yield a final set of 71,202 transversion SNPs. ADMIXTURE was run with cross-validation enabled using --cv flag for all ancestral population numbers from $K = 3$ to $K = 20$.

## Runs of homozygosity

The presence and length of runs of homozygosity greater than 4 cM in the Nuwayrat genome were estimated using hapROH v.0.64 (ref. 41) on the 1.2 million SNP set of sites.

## qpAdm modelling

For all qpAdm modelling in this study, we estimated the ancestry proportions as a mixture of a set of left (source) rotating populations differentially related to a set of right (outgroup) populations using ADMIXTOOLS 2 (ref. 42) qpadm_rotating() with the option maxmiss = 0.1, removing genotypes missing in more than 10% of populations. We restricted the analysis to genomes with both transitions and transversions (half/plus UDG-treated libraries or single-stranded

libraries called using SequenceTools[115] --singleStrandMode), removing CpG sites, to increase the robustness of the models. We considered only models with three or less sources. We restricted the fixed set of outgroup to populations distantly related to any left populations and with genomes greater than or equal to 2×: Ju_hoan_North.DG, Ethiopia_4500BP.SG, Latvia_HG_UDG, USA_Ancient_Beringian.SG, Vanuatu_400BP_UDG, Japan_HG_Jomon_UDG and China_NEastAsia_Coastal_EN_UDG. This analysis was conducted on the 1240k dataset. These parameters are always true unless otherwise stated.

We ranked the non-rejected models first on the basis of the minimal number of source populations, assuming that a fewer number of source populations is more parsimonious. Then, if several models with the same number of sources are not rejected, we considered the $P$ value, given that the number of SNPs in the rotating models are nearly equal (10% missingness allowed between populations). Supplementary Information section 4 details all models tested.

**Nuwayrat genome ancestry modelling.** We first estimated the Nuwayrat genome (NUE001) and contemporary North African and West Asian populations (Levant_BA (also for each of the eight archaeological sites separately), Anatolia_BA and Morocco_MN) ancestry proportions as a combination of distal Neolithic populations from North Africa and West Asia (Morocco_Epipaleolithic, Anatolia_Neolithic, Levant_Neolithic, Zagros_Neolithic and Caucasus_Neolithic). This analysis was carried out on 433,280–558,848 SNPs. Then, we estimated NUE001, Bronze Age Levant (also for each of the eight archaeological sites separately) and Bronze Age Anatolia ancestry components, adding more proximal Neolithic and Chalcolithic North African and West Asian populations as potential sources (Morocco_EN_ktg, Morocco_MN, Anatolia_Neolithic, Anatolia_Chalcolithic, Levant_Neolithic, Levant_Chalcolithic, Zagros_Neolithic, Zagros_Chalcolithic, Mesopotamia_Neolithic, Caucasus_Neolithic and Caucasus_Chalcolithic) as well as two Neolithic Europeans: Spain_EN and Greece_Neolithic, referred to as the full qpAdm model. This analysis was conducted over 474,731–578,969 SNPs.

**Third Intermediate Period ancestry modelling.** We estimated the ancestry proportions of the two Third Intermediate Period Egyptians[20] using North African and West Asian populations who lived between the Old Kingdom and Third Intermediate periods as potential sources (NUE001, Anatolia_BA, Levant_BA, Iran_BA and Caucasus_BA), as well as a Bronze Age Greek population (Greece_Minoan). We also added Morocco_MN and Mesopotamia_N to test whether the Third Intermediate Period Egyptians share a closer ancestry with NUE001 or a source more related to one of these two ancestries present in NUE001. This analysis was conducted on 290,262 SNPs.

**Present-day Egyptian genome ancestry modelling.** We directly estimated the proportion of NUE001 ancestry in present-day Egyptians[164] as a whole or each individual separately, as well as ancestries from North African (Morocco_MN), West Asian (Caucasus_BA, Iran_BA and Levant_BA) and European (Greece_Minoan) populations, as well as East and West Africa (Ethiopia_4500BP.SG and Congo_Kindoki_Protohistoric). For each region, we selected the representatives closest to NUE001's lifetime. Anatolia_BA was removed from the list of West Asian groups because its inclusion led to a substantial drop-off of genomes having at least one model passing $P = 0.05$. This analysis was conducted on 767,305 SNPs.

**Ancient East African ancestry modelling.** We estimated the ancestry proportion in ancient East African[43,54,55,144] using both NUE001 and Levantine Chalcolithic genomes as competing sources for the Eurasian-like component. We used as potential left sources NUE001, Levant_Chalcolithic, Ethiopia_4500BP.SG, Dinka.DG, Congo_Kindoki_Protohistoric and South_Africa_2200BP.SG. For this model,

the following fixed right groups were used: Chimp.REF, Latvia_HG_UDG, USA_Ancient_Beringian.SG, Vanuatu_400BP_UDG, Japan_HG_Jomon_UDG and China_NEastAsia_Coastal_EN_UDG. This analysis was conducted on 141,323–350,110 SNPs.

## $f_4$ statistics

$f_4$ statistics of the form $f_4$(NUE001, Morocco_MN; X, Ju_hoan_North.DG) and $f_4$(NUE001, Morocco_MN; Mesopotamia_N, X), X being the non-North African groups used in the full qpAdm model, Palaeolithic Levant or Palaeolithic Anatolia, were estimated to confirm the probable source of admixture in the Nuwayrat genome when compared with the Middle Neolithic Moroccan group. $f_4$ statistics was computed using ADMIXTOOLS 2 (ref. 42) with the option maxmiss = 0.1. We restricted the analysis to genomes with both transitions and transversions (half/plus UDG-treated libraries or single-stranded libraries called using SequenceTools[115] --singleStrandMode). This analysis was conducted on the 1240k dataset on 280,544 SNPs.

## Imputation

The genotypes of the Nuwayrat genome were imputed together with 200 ancient genomes from North Africa and West Asia associated with the Palaeolithic, Neolithic and Bronze Age culture (Supplementary Table 3). We restricted the imputation to whole-genome sequencing data greater than 0.5× coverage or 1240k SNP capture data greater than 2× coverage, following recommendations from Sousa da Mota et al.[172].

First, we called genotypes using bcftools v.1.19 (ref. 173) with the commands bcftools mpileup with parameters -I -E -a 'FORMAT/DP' --ignore-RG and bcftools call -Aim -C alleles. We then imputed the missing genotypes using Glimpse v.1.1.0 (ref. 174). First, we used GLIMPSE_chunk to split chromosomes into chunks of 2 Mb and with a 200-kb buffer region. Second, imputation was performed with GLIMPSE_phase on the chunks with default parameters --burn 10, --main 10 and --pbwt-depth 2, with 1000 Genomes[175] as the reference panel. We then ligated the imputed chunks with GLIMPSE_ligate.

To remove transitions caused by post-mortem damage before imputation, for the genomes generated with UDG treatment, we first hard-trimmed the first and last three base pairs of each read and removed CpG sites, and for the genome generated without UDG treatment, we removed all transition sites after SNP calling.

We finally restricted the imputed genotypes to those with genotype probability ≥0.99 and minor allele frequency ≥0.01 using the command bcftools filter -i 'MAX(FORMAT/GP)>=0.99 && INFO/RAF>=0.01&&INFO/RAF<=0.99' --set-GTs '.'.

The imputed dataset was used for phenotype prediction (see below) and admixture dating using DATES (Supplementary Information section 4 and Supplementary Table 11).

## Phenotype prediction

The genotypes responsible for skin, hair and eye colour prediction were investigated using the HIrisPlex-S system[176–178] using the imputed genotypes.

## Reporting summary

Further information on research design is available in the Nature Portfolio Reporting Summary linked to this article.

## Data availability

All mapped sequence data generated for this project are available from the European Nucleotide Archive under the study accession no. PRJEB88328.

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

**Acknowledgements** A.M.J. was supported by ECR strategic support of early career researchers in the faculty of science at Liverpool John Moores University, awarded to L.G.-F., and European Research Council grant no. 852558, awarded to P.S. Open Access funding was provided by Liverpool John Moores University. This study was supported by the European Research Council (grant no. 852558 to P.S.). L.S. was supported by a Sir Henry Wellcome Fellowship (220457/Z/20/Z). P.S. was supported by the European Molecular Biology Organization, the Vallee Foundation, the Wellcome Trust (217223/Z/19/Z) and The Francis Crick Institute core funding (FC001595) from Cancer Research UK, the UK Medical Research Council and the Wellcome Trust. We thank the Genomics Science Technology Platform at the Francis Crick Institute for technical assistance. We thank M. Dee (University of Groningen) for providing advice and support on how to combine the three radiocarbon dates generated from the Nuwayrat skeletal remains and M. Stratigos (University of Aberdeen) about radiocarbon modelling and collagen decay. We are grateful to A. Eladany (University of Aberdeen) for sharing valuable knowledge on Egyptian archaeology and ethical recommendations, and B. Vanthuyne (University of Cologne) for sharing literature on the archaeological site. Liverpool John Moores University colleagues, M. Borrini, C. Eliopoulos, J. Ohman and A. Wilshaw, provided advice on the osteological profile. We also appreciate the advice from J. Kabaciński (Polish Academy of Sciences, Poznań Branch).

**Author contributions** A.M.J., J.D.I., P.S. and L.G.-F. contributed to the conception and design of the study. A.M.J. and L.G.-F. selected and sampled archaeological material for DNA extraction. A.M.J., K.A., M.K., F.T., M.W. and M.H. performed DNA laboratory work. A.M.J., C.B. and A.G. analysed the genetic data. J.D.I. performed osteological analyses. A.C. provided the archaeological context and interpretation. R.M., E.H., A.J.N. and E.I. performed isotope analysis and interpretation. C.W. provided facial reconstruction. A.M.J., J.D.I., P.S. and L.G.-F. wrote the paper with substantial inputs from A.C., K.A., C.B., M.S., L.S., F.T., N.B., F.-X.R., C.W. and M.H.

**Competing interests** The authors declare no competing interests.

**Additional information**
**Correspondence and requests for materials** should be addressed to Adeline Morez Jacobs, Pontus Skoglund or Linus Girdland-Flink.

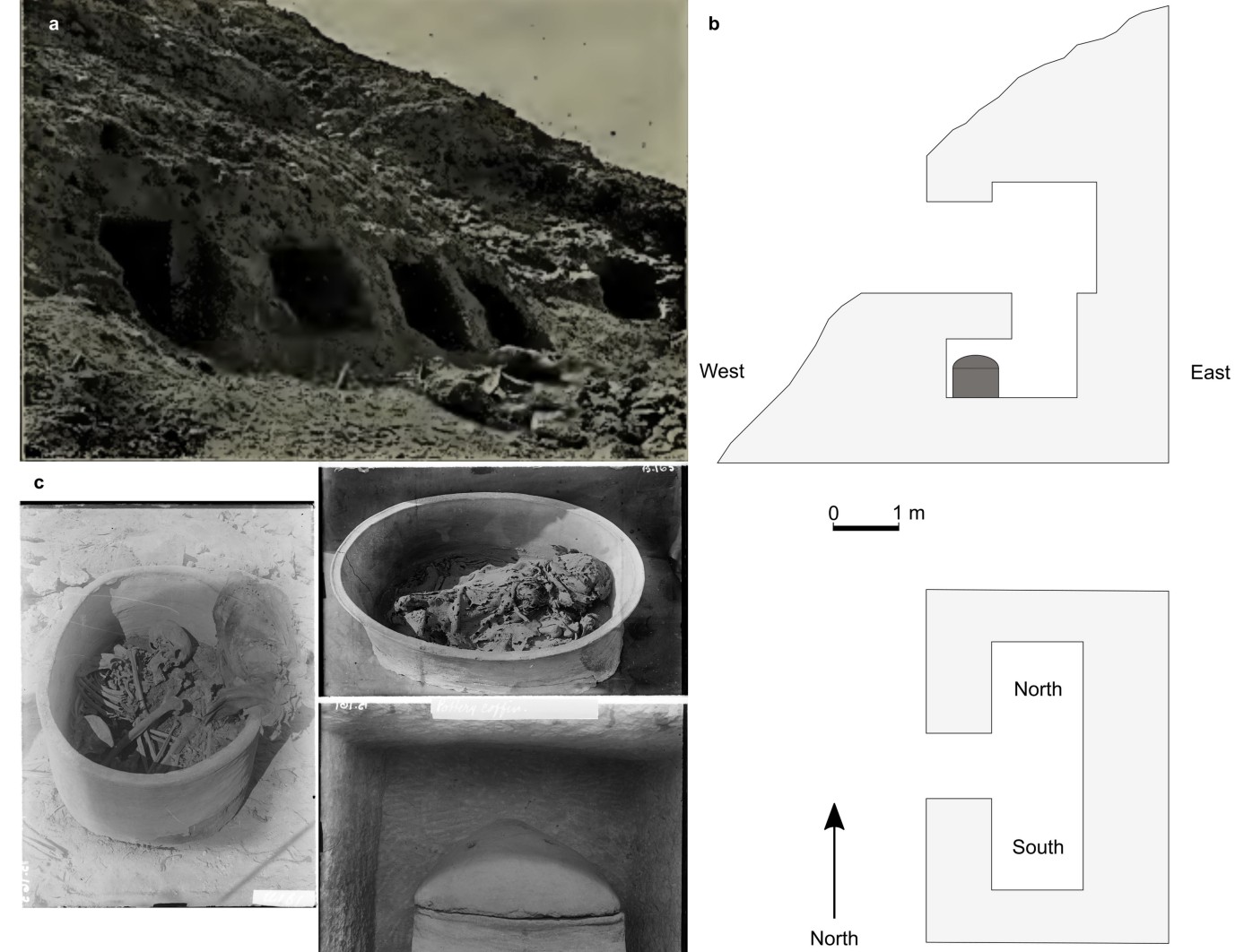

**Extended Data Fig. 1 | Archaeological context at the Nuwayrat site.**
**a**, Rock-cut tombs at Nuwayrat enclosing the pottery vessel containing the pottery coffin burial. **b**, An impression of the rock-cut tomb based on the archaeologist John Garstang's description, with the pottery coffin burial in the south burial chamber. **c**, Pottery coffin and archaeological remains of the Nuwayrat individual, as discovered in 1902. Photos in **a** and **c** reproduced courtesy of the Garstang Museum of Archaeology, University of Liverpool.

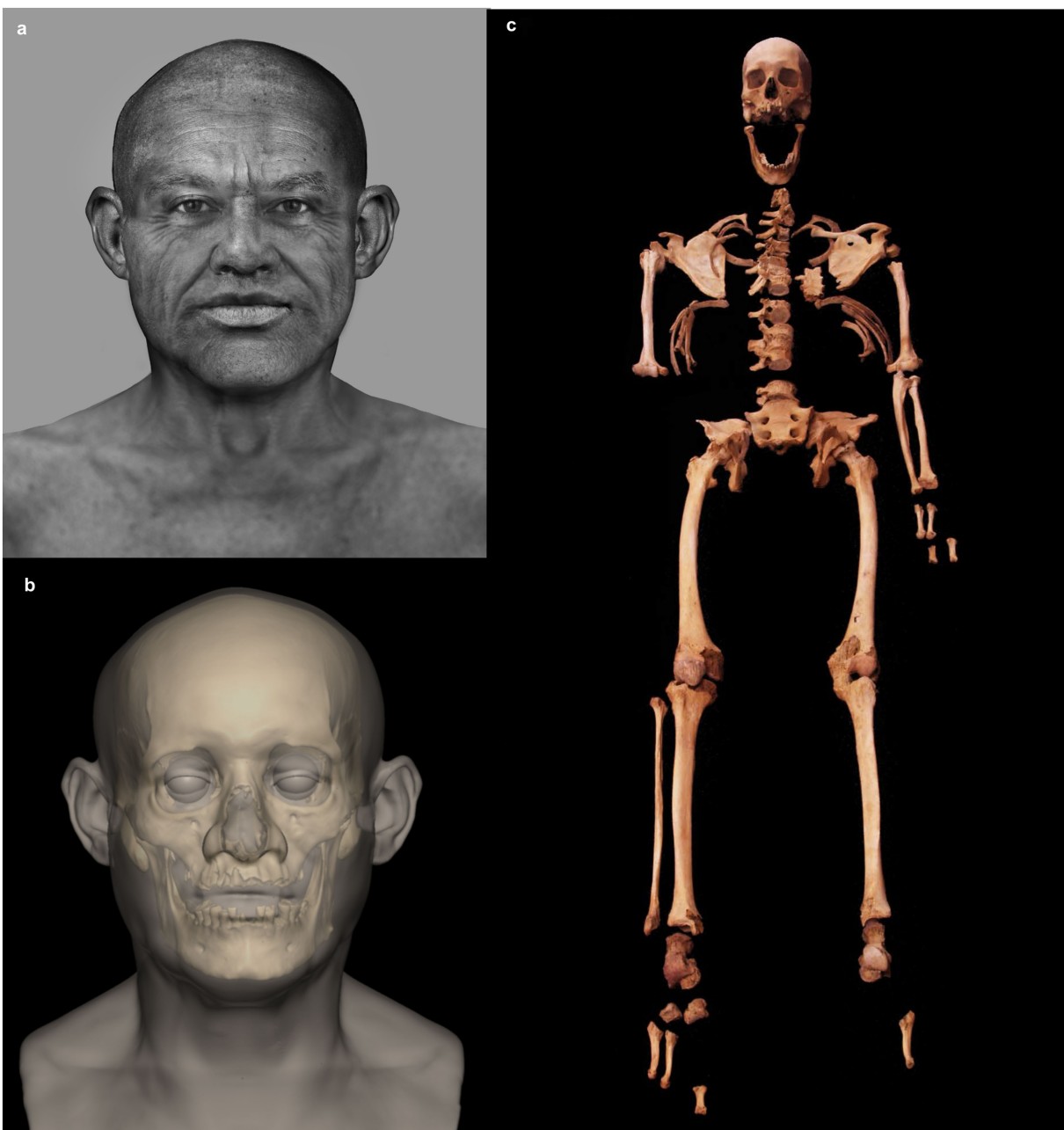

**Extended Data Fig. 2 | Facial reconstruction and depiction created from the Nuwayrat individual skull. a**, Final facial depiction of the Nuwayrat individual. **b**, Virtual fit of the skull and facial reconstruction. **c**, The Nuwayrat individual's partially complete skeleton.

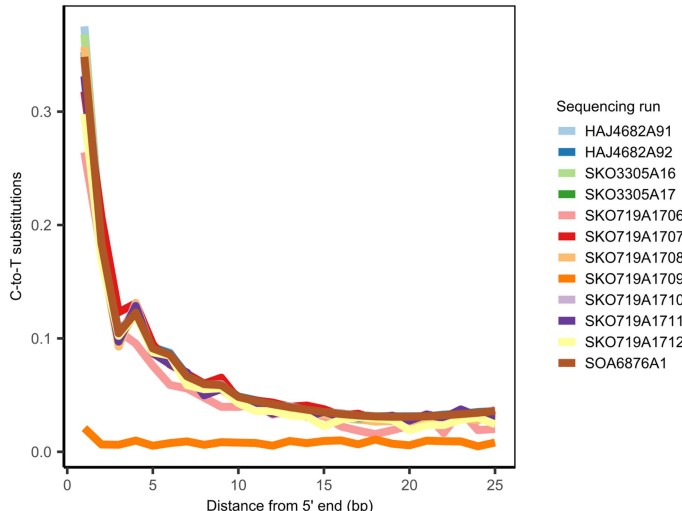

**Extended Data Fig. 3 | Damage patterns at the 5′ end of the reads in each sequencing run from the Nuwayrat individual.** All sequencing runs but one show a significant increase of C-to-T transitions at the 5′ end of DNA fragments, indicative of authentic ancient DNA.

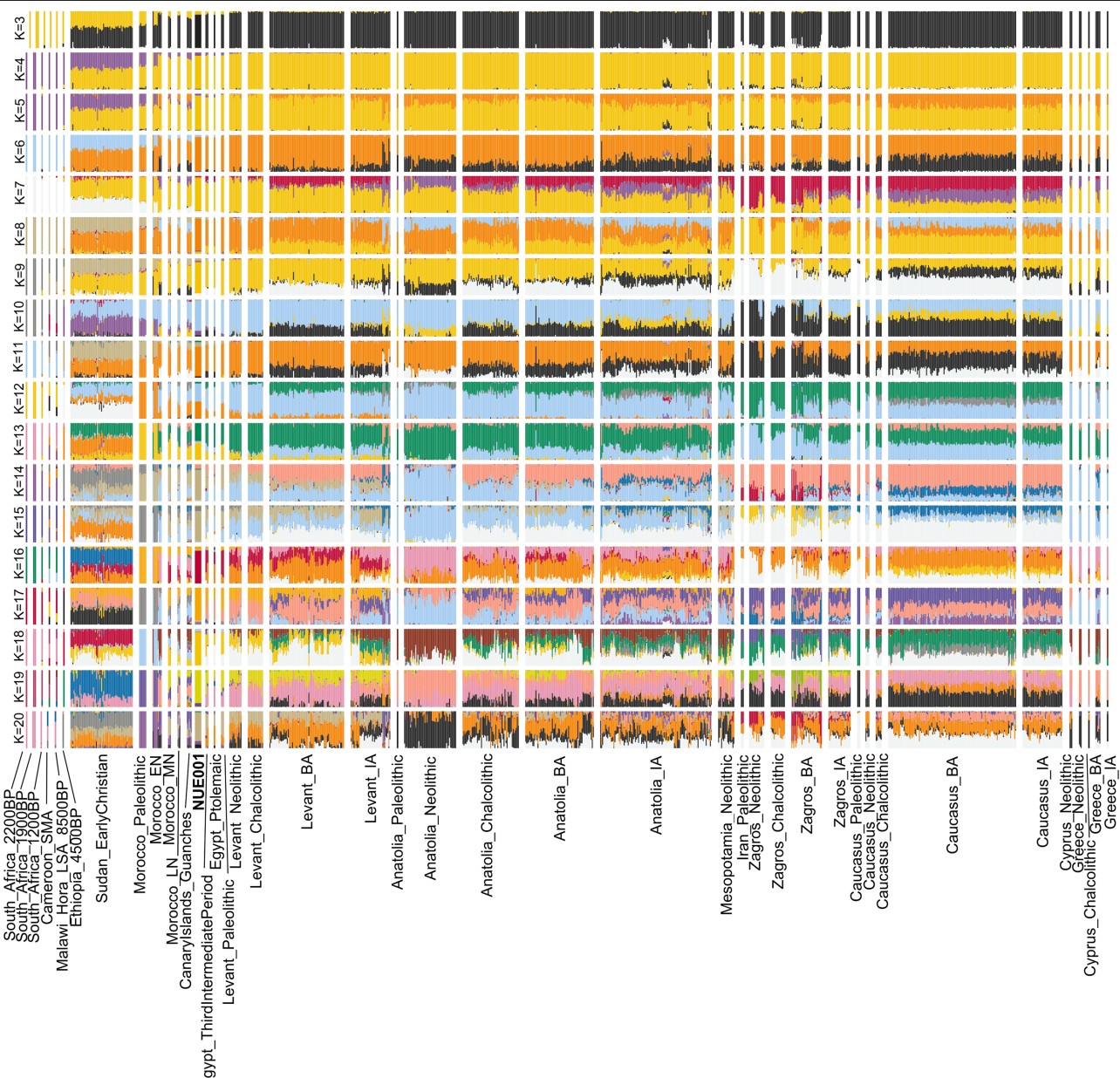

**Extended Data Fig. 4 | ADMIXTURE clustering analysis of the Old Kingdom Egyptian genome in the context of ancient genomes (K = 3 to 20).** ADMIXTURE was generated on 4,574 present-day and ancient genomes from the 'HO' dataset (Supplementary Data Table 3), over 71,202 transversion SNPs. ADMIXTURE output on the present-day genomes are displayed in Extended Data Fig. 5.

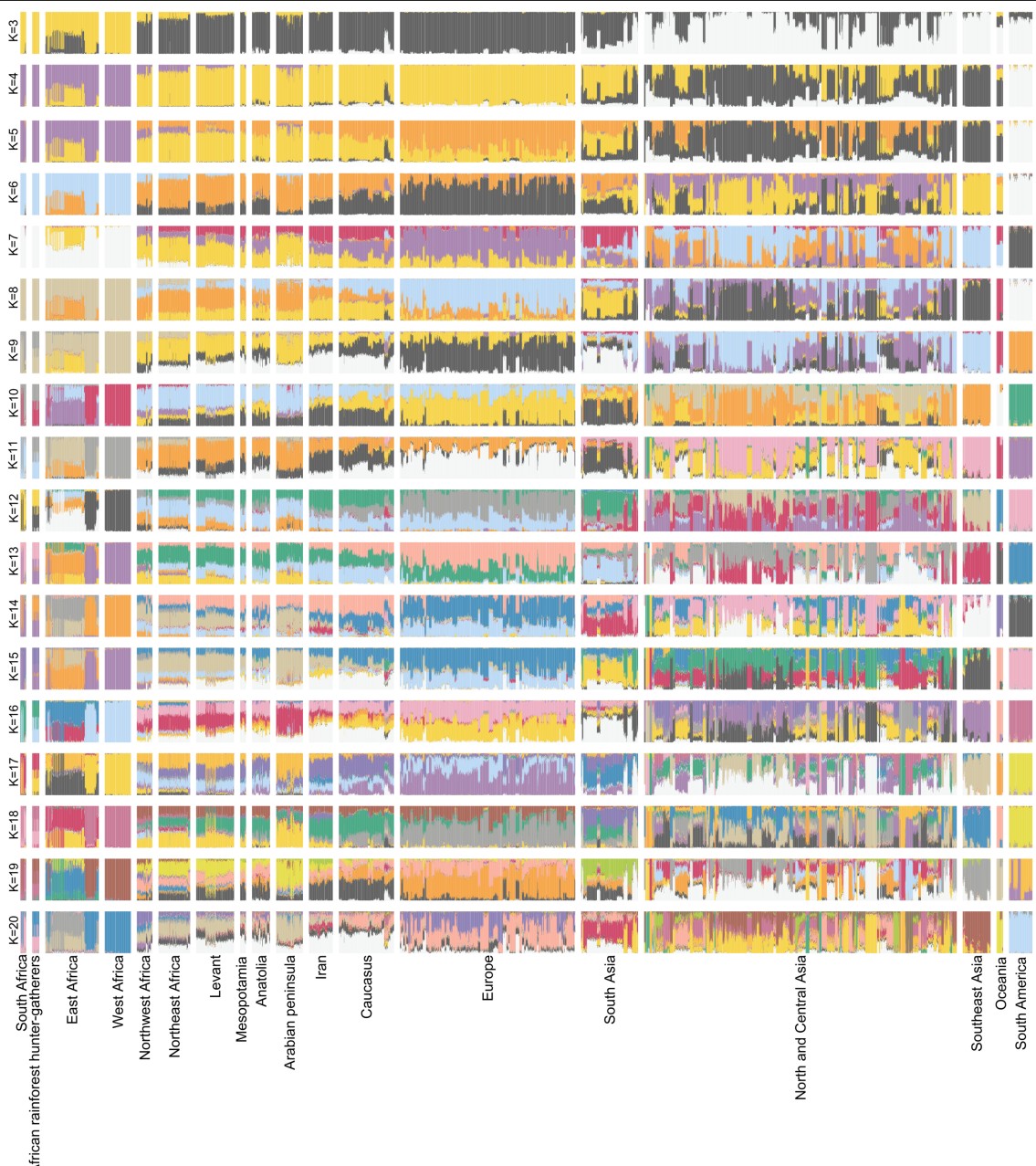

**Extended Data Fig. 5 | ADMIXTURE clustering analysis of the Old Kingdom Egyptian genome in the context of present-day genomes (K = 3 to 20).** ADMIXTURE was generated on 4,574 present-day and ancient genomes from the 'HO' dataset (Supplementary Data Table 3), over 71,202 transversion SNPs. ADMIXTURE output on the ancient genomes are displayed in Extended Data Fig. 4.

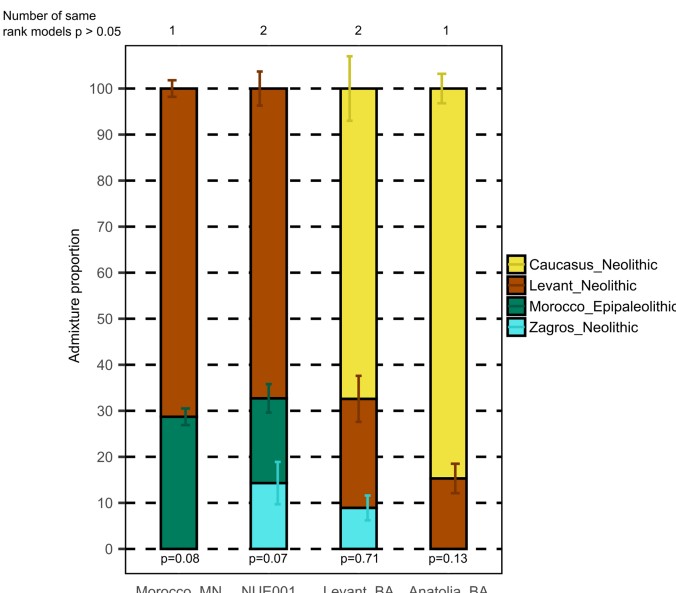

**Extended Data Fig. 6 | Distal ancestries in the Nuwayrat genome and contemporary groups.** The model included Epipaleolithic/Neolithic groups from North Africa and West Asia as rotating sources in *qpAdm* (*Morocco_Epipaleolithic*, *Anatolia_Neolithic*, *Levant_Neolithic*, *Zagros_Neolithic*, *Caucasus_Neolithic*). Details of all models passing p > 0.05 are displayed in Supplementary Data Table 5. Values represent best-fitting model estimates ±1 SE (error bars). This analysis was conducted over n = 515,802 SNPs for NUE001, n = 558,549 SNPs for Morocco_MN, n = 558,847 SNPs for Levant_BA, and n = 558,848 SNPs for Anatolia_BA.

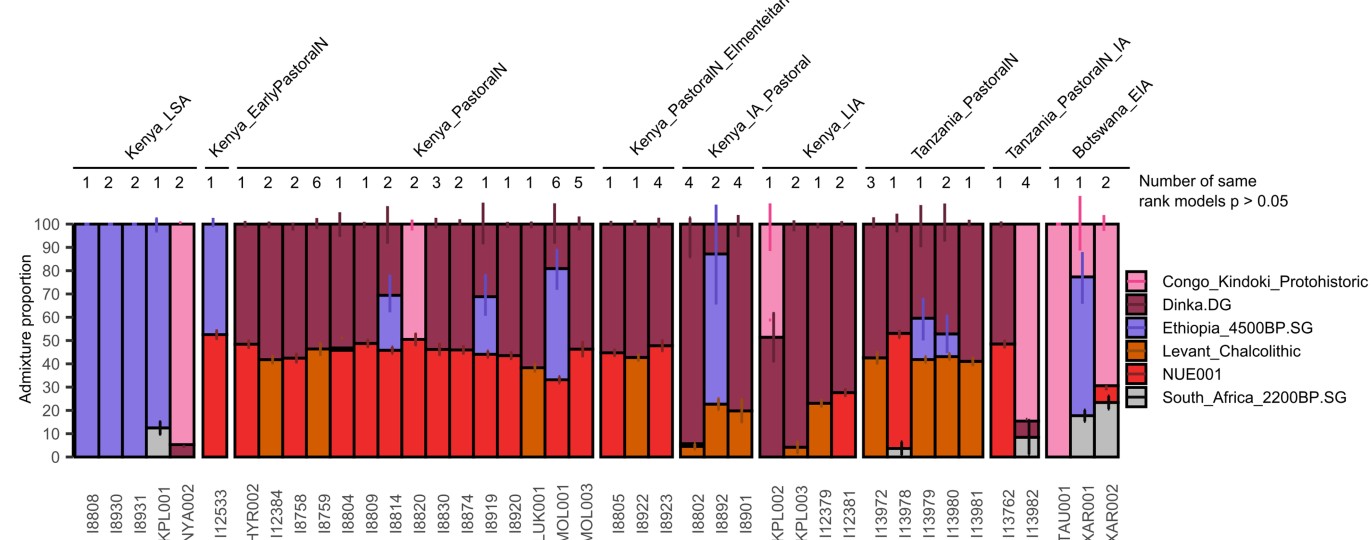

**Extended Data Fig. 7 | Ancestry modelling of ancient East African genomes with qpAdm.** Best-fit models are represented. Details of all models passing p > 0.05 are described in Supplementary Data Table 12. N, Neolithic; IA, Iron Age; EIA, Early iron Age; LIA, Late Iron Age; LSA, Late Stone Age. Values represent best-fitting model estimates ± 1 SE (error bars). This analysis was conducted over 141,323-350,110 SNPs (see Supplementary Data Table 12).

# Reporting Summary

## Statistics

For all statistical analyses, confirm that the following items are present in the figure legend, table legend, main text, or Methods section.

| n/a | Confirmed | |
|---|---|---|
| ☐ | ☒ | The exact sample size (*n*) for each experimental group/condition, given as a discrete number and unit of measurement |
| ☐ | ☒ | A statement on whether measurements were taken from distinct samples or whether the same sample was measured repeatedly |
| ☐ | ☒ | The statistical test(s) used AND whether they are one- or two-sided<br>*Only common tests should be described solely by name; describe more complex techniques in the Methods section.* |
| ☒ | ☐ | A description of all covariates tested |
| ☒ | ☐ | A description of any assumptions or corrections, such as tests of normality and adjustment for multiple comparisons |
| ☐ | ☒ | A full description of the statistical parameters including central tendency (e.g. means) or other basic estimates (e.g. regression coefficient) AND variation (e.g. standard deviation) or associated estimates of uncertainty (e.g. confidence intervals) |
| ☐ | ☒ | For null hypothesis testing, the test statistic (e.g. *F*, *t*, *r*) with confidence intervals, effect sizes, degrees of freedom and *P* value noted<br>*Give P values as exact values whenever suitable.* |
| ☒ | ☐ | For Bayesian analysis, information on the choice of priors and Markov chain Monte Carlo settings |
| ☒ | ☐ | For hierarchical and complex designs, identification of the appropriate level for tests and full reporting of outcomes |
| ☒ | ☐ | Estimates of effect sizes (e.g. Cohen's *d*, Pearson's *r*), indicating how they were calculated |

*Our web collection on statistics for biologists contains articles on many of the points above.*

## Software and code

Policy information about availability of computer code

| | |
|---|---|
| Data collection | Sequence demultiplexing and adapter removal: AdapterRemoval v2.3.1. |
| Data analysis | A full description of all software and respective packages used for data analysis can be found in the Method section and are publicly available.<br>Sequencing reads processing: nf-core/eager v2.3.3 pipeline; Genomic read mapping: Burrows-Wheller Aligner (BWA, v0.7.17); Duplicates removal: Dedup v0.12.8; Aligned reads filtering: SAMtools v1.9.2; Substitution distribution: MapDamage v2; Genome-wide contamination rate using the Conditional Substitution Rate: PMDtools v0.60; mitochondrial DNA-based contamination and consensus: Schmutzi (commit be61017); X-chromosome-based contamination in males: ANGSD v0.933; Reads count for molecular sexing: SAMtools v1.9.2; Sequencing runs merging: SAMtools v1.9.2; Pseudo-haploid SNP calling: SAMtools v1.9.2 and SequenceTools v1.5; mitochondrial haplogroup assignement: Haplogrep v3.2.1 and PhyloTree mtDNA tree build 17 (18 Feb 2016); Y-chromosome haplogroup assignement: pathPhynder and the International Society Of Genetic Genealogy (ISOGG v15.73); Genomic dataset management (including LD pruning and datasets merging): PLINK v. 1.9, EIGENSOFT 6.1.4; Principal Component Analysis (PCA): EIGENSOFT 6.1.4; ADMIXTURE clustering analysis: ADMIXTURE v1.2; Runs of homozygosity: hapROH v0.64; qpAdm modelling and F4-statistics: ADMIXTOOLS2 R package; f4-ratio: admixr R package; Phenotype predicton: HirisPlexS system. Results visualization and plot generation: R v. 4.2.3, ggplot2. Imputation: BCFtools v1.19; GLIMPSE v1.1.0. Admixture dating: DATES.<br>Radiocarbon dates calibration and combination: Oxcal v4.4.4 and IntCal20.<br>Craniodental-based biological affinity: rASUDAS, CRANID: CR6bIND.<br>Facial reconstruction: Geomagic Freeform Plus 2024.0.87 software. |

For manuscripts utilizing custom algorithms or software that are central to the research but not yet described in published literature, software must be made available to editors and reviewers. We strongly encourage code deposition in a community repository (e.g. GitHub). See the Nature Portfolio guidelines for submitting code & software for further information.

## Data

Policy information about availability of data

All manuscripts must include a data availability statement. This statement should provide the following information, where applicable:

- Accession codes, unique identifiers, or web links for publicly available datasets
- A description of any restrictions on data availability
- For clinical datasets or third party data, please ensure that the statement adheres to our policy

Human reference genome build 37 (hs37d5) (https://ftp.1000genomes.ebi.ac.uk/vol1/ftp/technical/reference/phase2_reference_assembly_sequence/)
All the generated sequence data are available as bam files of aligned reads at the European Nucleotide Archive (ENA) under the project accession number PRJEB77356. Comparative ancient and modern genetic data were downloaded from the Allen Ancient DNA Resource (AADR, https://doi.org/10.7910/DVN/FFIDCW), the European Nucleotide Archive (ENA) under the accession number PRJEB59008 (Simoes et al. 2023), PRJEB50507 (Altınışık et al. 2022), and the European Genome-Phenome Archive (EGA) under the accession number EGAS00001000480 (Pagani et al. 2015; Egyptian low coverage). The origin of each genetic data is described in the Method section and in Supplementary Data Table 3.

## Research involving human participants, their data, or biological material

Policy information about studies with human participants or human data. See also policy information about sex, gender (identity/presentation), and sexual orientation and race, ethnicity and racism.

| | |
|---|---|
| Reporting on sex and gender | N/A |
| Reporting on race, ethnicity, or other socially relevant groupings | N/A |
| Population characteristics | N/A |
| Recruitment | N/A |
| Ethics oversight | N/A |

Note that full information on the approval of the study protocol must also be provided in the manuscript.

# Field-specific reporting

Please select the one below that is the best fit for your research. If you are not sure, read the appropriate sections before making your selection.

☒ Life sciences ☐ Behavioural & social sciences ☐ Ecological, evolutionary & environmental sciences

For a reference copy of the document with all sections, see nature.com/documents/nr-reporting-summary-flat.pdf

# Life sciences study design

All studies must disclose on these points even when the disclosure is negative.

| | |
|---|---|
| Sample size | No statistical methods were used to determine ancient DNA sample size a priori. Genomic, radiocarbon and isotope (13C, 15N, 18O, 88Sr/86Sr) data from an ancient individual from Egypt analysed in this study depend on the availability of human remains from ancient Egypt with preserved and retrievable ancient DNA sequences. This specimen is very rare, given the poor molecular preservation of human remains from this period in that region. Given the millions of genetic variants analysed, information about the genetic history can be retrieved. |
| Data exclusions | For the Nuwayrat genome and the comparative ancient genomes retrieved from bam files (from Altınışık et al. 2022 and Simoes et al. 2023), sequencing reads that did not map to the human reference hs37d5, were shorter than 35 bp, were library PCR amplification duplicates, had a mapping quality < 30 or contained indels, were removed from the bam files. Sequencing data with evidence of DNA contamination (SKO719A1706 and SKO719A1709) were also excluded. For single-stranded libraries (from the Nuwayrat genome), at C/T SNPs, forward mapping reads were discarded, and at G/A SNPs in reverse mapped reads were discarded using SequenceTools. For the double-stranded damage repair UDG-treated genomes (from Simoes et al. 2023), the first and last three bases of the sequenced reads were trimmed. For the double-stranded non-damage repaired genomes (from Altınışık et al. 2022), only transition sites were discarded. SNP calling was restricted to bases with base quality >30. We called pseudo-haploid SNPs, keeping only one random allele per individual at each position.<br>For the comparative present-day Egyptian genomes from Pagani et al. 2015, individuals with more than 10% missing genotypes, genotypes with missing call rates of 2% or failing the Hardy-Weinberg equilibrium test with p<1e-06 were discarded.<br>For all genomes, we kept only biallelic SNPs present in the Human Origin array or the 1240K capture SNP set.<br>In the whole comparative dataset, for pairs of first- and second-degree relatives, the individual with the lowest genomic coverage of the pair was excluded.<br>The PCA and ADMIXTURE clustering analyses were restricted to the transversions sites in the Human Origin array (111,208 SNPs). For the ADMIXTURE clustering analysis, all genomes (present-day and ancient) were pseudo-haploidised (we kept only on random allele at each diploid site per individual), genotypes were pruned for linkage disequilibrium resulting in 84,528 transversions sites being analysed. qpAdm and F4-statistics were restricted to the 1240K capture SNP set with CpG sites excluded, with only UDG-treated genomes and the Nuwayrat |

used in the analyses.

F4-ratio to estimate Basal Eurasian ancestry restricted to the 1240K capture SNP set on transversion sites only.

Imputation was carried on the Nuwayrat genome and 200 ancient genomes from North Africa and West Asia associated with the Palaeolithic, Neolithic and Bronze Age, whole genome sequenced to >0.5X coverage or captured >2x coverage. For the genomes generated with UDG treatment, we first hard-trimmed the first and last three base pair of each reads and removed CpG sites and for the genome generate without UDG treatment, we removed all transition sites after SNP calling. We finally restricted the imputed genotypes to those with genotype probability GP ≥ 0.99 and minor allele frequency MAF ≥ 0.01, using the command bcftools filter -i 'MAX(FORMAT/GP)>=0.99 && INFO/RAF>=0.01&&INFO/RAF<=0.99' --set-GTs '.'.

Phenotype prediction and admixture dating were carried on this imputed dataset.

**Replication**

Several DNA extracts and multiple genomic libraries were generated for the newly reported sample from Nuwayrat, Egypt, and several rounds of sequencing were performed for each library (as reported in Supplementary Data Table S1), which acts as replication. Data was merged for downstream analysis after confirming similar results on contamination estimates and mitochondrial haplogroup. Thousands to millions of genetic markers were then analysed as an internal replication of the results. Detailed description of the methods used, including samples included in the dataset, software employed and respective parameters is available in the Method section.

**Randomization**

Randomization is not relevant to this study. Samples are grouped based on sampling locations, dates and genetic affinities.

**Blinding**

Blinding is not applicable for ancient specimens as the sampling locations and dates are known a-priori. In downstream data, analysis blinding is also not relevant since the newly generated ancient genome is co-analysed with previously published present-day and ancient human genomes.

# Reporting for specific materials, systems and methods

We require information from authors about some types of materials, experimental systems and methods used in many studies. Here, indicate whether each material, system or method listed is relevant to your study. If you are not sure if a list item applies to your research, read the appropriate section before selecting a response.

## Materials & experimental systems

| n/a | Involved in the study |
|-----|----------------------|
| ☒ | Antibodies |
| ☒ | Eukaryotic cell lines |
| ☐ ☒ | Palaeontology and archaeology |
| ☒ | Animals and other organisms |
| ☒ | Clinical data |
| ☒ | Dual use research of concern |
| ☒ | Plants |

## Methods

| n/a | Involved in the study |
|-----|----------------------|
| ☒ | ChIP-seq |
| ☒ | Flow cytometry |
| ☒ | MRI-based neuroimaging |

## Palaeontology and Archaeology

**Specimen provenance**

The specimen provenance is described in the Supplementary Information 1. The specimen was excavated at the Nuwayrat necropolis near Beni Hasan, Egypt. It was donated between 1902-04 by the Egyptian Antiquities Service to the members of the Beni Hasan excavation committee, and subsequently donated to the Institute of Archaeology, University of Liverpool, UK and exported under John Garstang export permit. The specimen was then given to the World Museum (previously Liverpool City Museum) in 1950

**Specimen deposition**

The World Museum of Liverpool (UK) is the sole curator of the specimen.

**Dating methods**

Three independent samples from the individual were directly radiocarbon dated using accelerator mass spectrometry (AMS) at the Beta Analytic Carbon dating laboratory (the two teeth that yielded ancient DNA) or at the Oxford Radiocarbon Accelerator Unit (femur), which date was obtained from the World Museum archive. Radiocarbon calibration was performed using OxCal v.4.4 and the IntCal20 dataset. The three dates were coherent and combined using OxCal v.4.4.

☒ Tick this box to confirm that the raw and calibrated dates are available in the paper or in Supplementary Information.

**Ethics oversight**

Permits for sampling and analyses of the archaeological material were obtained from the appropriate institutions.

Note that full information on the approval of the study protocol must also be provided in the manuscript.

## Plants

Seed stocks

N/A

Novel plant genotypes

N/A

Authentication

N/A

