## [Peer Review File · Nature]

Whole-genome ancestry of an Old Kingdom Egyptian

Corresponding Author: Dr Adeline Morez Jacobs

Parts of this Peer Review File have been redacted as indicated to maintain a contributor's confidentiality.

Version 0:

Reviewer comments:

Referee #1

(Remarks to the Author)

Morez et al. present a 1.51X whole genome of an individual from Nuwayrat, Egypt dating to the Early Dynastic-Old Kingdom period. This makes this genome the oldest genomic sequence from Egypt, thus enabling first insights - barring sample size limitation - into the genetic diversity during the early dynastic periods. The manuscript engages primarily with the genomic data and the presented analyses are fairly standard for an ancient DNA paper dealing with a single low coverage genome that precludes more involved population genetics methods. Based on the genomic analyses, the authors report that the sequenced Nuwayrat individual had genetic ancestry related primarily to North African Neolithic and eastern Fertile Crescent groups (best proxy being Neolithic Mesopotamia). The Nuwayrat individuals' ancestry may have contributed in part to later Third Intermediate period genomes as well as several present-day Egyptians, though these groups have received substantial and later gene flow from genetic sources related to Bronze Age Levant and ancient East African groups, respectively. Additionally, there are osteological and facial reconstruction analyses included, the former suggesting that despite the burial context hinting at this individual's higher social class they were likely involved in a labor-intensive occupation. Overall, this manuscript pushes past the methodological challenges inherent in a region such as Egypt and generates whole-genome data, which is commendable.

I have been through the genomic analyses presented in this manuscript. I am quite happy with the careful manner in which the data QC and filtering has been performed. I am also satisfied with the descriptions and implementation of the analyses that are feasible given the generated data. The authors have been thoughtful in the qpAdm analytical set-up, which forms the core of the results in this study. I believe they have accounted for the distal and proximal sources with the available aDNA data and show fairly convincingly the sources that are able to model the Nuwayrat individual, in addition to considering alternate models and possibilities of overfitting due to one of the sources being a compound in itself. I do not have any major concerns as such but list a few minor suggestions/questions below:

- Title: "Whole-genome ancestry" sounds a little awkward to me, perhaps: Whole-genome sequencing of an Old Kingdom Egyptian.
- Supp Table 1 lists information on 11 libraries but main text talks about 7 extracts - did some extracts have multiple libraries? Can the dropped libraries be marked clearly? Several columns in Supp Table 1 are somewhat unclear - it would be good to have descriptions of the headers.
- I was unable to locate a figure corresponding to the ROH analysis.
- Please standardize the labels for Zagros_N and Iran_N, which are used interchangeably in the qpAdm analysis (e.g. in Extended Data Fig 6 where both are used to indicate source in the figure and text legends).
- I was unable to locate the qpAdm results for the tests noted on page 6 (overfitting due to the the use of Middle Neolithic Morocco) and page 7 (BA Levantine as sources for the Nuwayrat individual).
- In the supplementary text describing the f4 ratio for investigating Basal Eurasian, it should note Extended Data Fig 7 instead of 6.

Referee #2

(Remarks to the Author)

Does the manuscript have flaws which should prohibit its publication? No.

To my knowledge, the conclusions are original.

The results presented are of immediate interest to geneticists, archaeologists, and anthropologists.

Outstanding features:

Morez and colleagues present new whole genome data from an ancient individual excavated from Nuwayrat, Egypt. The analysed archaeological remains show exceptional preservation for that region and period. Genomic data from ancient Egypt has thus far only been retrieved from three individuals – and even those are of poorer quality and from a more recent period.

The preservation of the whole skeleton has allowed for a multidisciplinary analysis of this individual, including a thorough osteological examination that reveals a great deal about this man's life, including his potential occupation, and a facial depiction.

General notes:

I applaud the authors for generating complete genome data for this rare individual, instead of enriching for a SNP panel.

The authors caution for the fact that this study is based on a single individual. Small sample sizes are common in ancient DNA studies, especially those based on sites with human remains with documented low molecular preservation, such as Egypt. The assumption for extrapolating such results to a population level is usually based on the individuals' expected representativeness of the general population genetic background. My concerns about the generalisation of the conclusions in this study are based on this individual's potential higher social status, which could hinder that extrapolation. I think that this should be more obviously stressed in the text.

I am surprised to note the absence of stable isotope analyses such as dietary isotope analysis and/or strontium analysis. The rarity of the specimen would justify investigating these aspects, which would in turn provide a greater depth to the manuscript. These could substantiate the claim of the higher social status of this individual, by possible comparison of dietary patterns with contemporary Egyptians and perhaps resolve if the individual was born locally or was a migrant.

I would expect, in the context of a culture as well-studied as the ancient Egypt (by many disciplines before archaeogenomics), a more robust and contextualised discussion, including perspectives and background from available literature/evidence (such as archaeological). I think that improving this aspect would round up the paper nicely and make it more interesting to the general audience, considering how populaancient Egypt is. As an example, the conclusions paragraph is rather dull and adds little to no information to the manuscript.

"These genetic links have ramifications for discerning early Dynastic population history, opening up the possibility that the incorporation of objects, images and ideas from outside Egypt was concomitant with some settlement of people."

Also, in the Summary "contacts between Egypt and the eastern Fertile Crescent were not limited to objects and imagery" Could the authors give examples? Is there (archaeological) evidence of objects, images or ideas that were incorporated from the regions that seem to have been connected to Egypt through the movement of people, given the genomic results?

Lastly, and related to the previous point, I am not totally convinced about Mesopotamia Neolithic ancestry being the most parsimonious source of ancestry present in NUE001. On PCA (Extended Data Fig 4), for example, NUE001 is clearly positioned between Morocco MN and Levant Neolithic/Chalcolithic, not towards Mesopotamian Neolithic. Unsupervised ADMIXTURE could be very informative, but unfortunately the figure shown is undecipherable and unhelpful. It is cautioned that "surrounding regions" are also potential sources. However, a rather strong emphasis is put on Neolithic Mesopotamia and I am concerned that this result is based mostly on the criteria implemented for model selection in the qpAdm framework, namely the minimal number of source populations criterium. Noticeably, the model selection criteria used follow recently published guidelines for the application of a qpAdm framework. However, at times two-source models are preferred to three-source models, even when the latter could possible make more sense from a population history dynamics perspective. This seems particularly relevant here given the complex Fertile Crescent demographic history and the (already) admixed nature of the source populations considered – especially proximal. Or perhaps all that is missing is contextualizing the dynamics behind the arrival of Mesopotamia Neolithic-related ancestry to Egypt – and how/why geographically and chronologically closer populations don't seem to be better proxies for that ancestry. Please see more specific comments/suggestions below.

General comments:

The use of line and page numbers would certainly help the reviewers.

A map with labels of the broader regions mentioned in the text, such as Fertile Crescent, Mesopotamia, etc. would be very helpful (I'm thinking perhaps something as seen in Lazaridis 2022 and Altınışık 2022). A more detailed timeline, including other periods mentioned in the text (e.g. Bronze Age is extensively mentioned) is essential.

A. Summary of the key results

Morez and colleagues present new whole genome data from an ancient individual excavated from Nuwayrat, Egypt. The authors stress the North African- and Fertile Crescent-related ancestry in the studied ancient Egyptian. This finding provides additional and important evidence on the links across North Africa and the Fertile Crescent, that now are shown to go beyond culture and trade, but also include migration. The unusual preservation of the whole skeleton has allowed for a multidisciplinary analysis of this individual, including a thorough osteological examination that reveals a great deal about this man's occupation and a facial depiction.

B. Originality and significance: if not novel, please include reference

The analysed archaeological remains show exceptional preservation for that region and period. Genomic data from ancient Egypt has thus far only been retrieved from three individuals – and even those are of poorer quality and from a more recent period. The newly generated data will be useful for many future studies.

C. Data & methodology: validity of approach, quality of data, quality of presentation

The authors use robust and well-established methods in ancient DNA research for generating and analysing genomic data from the ancient Egyptian individual through direct shotgun sequencing.

D. Appropriate use of statistics and treatment of uncertainties

All according to well established methods frequently used in ancient DNA analyses.

E. Conclusions: robustness, validity, reliability

The overall conclusions and the findings are sound, although at times lacking a stronger discussion/contextualisation.

F. Suggested improvements: experiments, data for possible revision

I list some suggestions and comments, which I hope the authors find helpful:

Summary

"Nuwayrat (Nuerat, نويرات)." Nuwayrat (Nuerat, نويرات).

"Most of his genome is represented by North African Neolithic ancestry"

While not incorrect, the multitude of ancestries present in Neolithic North(west) Africa makes this statement ambiguous. In the distal ancestry modelling, the majority of his genome is in fact represented by Levantine ancestry, and truly, so is the "North African Neolithic ancestry" mentioned here (I assume Morocco Middle Neolithic).

Main text

"The population has been said to be of local origin" based on what kind of evidence, if not archaeological, as that seems to be excluded, based on the next sentence?

"Exchange continued through the late 4th millennium BCE with the growing Sumerian civilisation of Mesopotamia" Please specify the kind of exchange meant.

"which some maintain can provide clues concerning occupation" some...authors?

"using a fully rotating model competition approach" It would be useful to include a very brief description of what this means.

"a single two-source model met the significance criteria ($p = 0.207$), which includes $78.4 \pm 4.0\%$ ancestry most closely related to genomes from the Middle Neolithic Moroccan site of Skhirat-Rouazi dated to 4,780-4,230 BCE (Morocco_MN), and the remainder most closely related to genomes from 9,000-8,000 BCE Neolithic Mesopotamia ($21.6 \pm 4.0\%$)." Both source populations have received geneflow from the Levant. Could this be problematic, eventually incurring into overfitting the model?

"pre-Bronze Age genomes are not available for the rest of the Maghreb" Not true for the Maghreb.

"from Mota, Ethiopia, or any other individuals in central, eastern or southern Africa (Fig. 2, Extended Data Fig. 5)" I only see Ethiopia 4500BP in Fig2 and also South Africa 2200 BP in extended Data Fig. 5. It would be very interesting to see if there is shared ancestry between NUE001 and pastoral Neolithic East Africans.

"other studies suggesting gene flow from the Mesopotamian and Zagros region into surrounding regions, such as Anatolia, during the Neolithic44" A single study is cited.

“wider expansion processes and an increase in mobility from the Mesopotamian region, into Egypt as well as Anatolia.” It is puzzling for me how such a scenario of mobility from Mesopotamia would not bring Levantine ancestry, admixed into the migrating population on its way to Egypt. Please elaborate. Is an alternative route of expansion (not encompassing the Levant) possible or supported by other lines of evidence?

“To investigate the presence of Mesopotamian ancestry in Nuwayrat relative to that in Bronze Age contemporaries of the latter, we extended the qpAdm model to adjacent regions.” It is not clear to me what was done, please rephrase.

“We detect ancestry from Neolithic Mesopotamia at three sites (Ebla, Baq’ah, Ashkelon), in proportions (41.8-54.8%) exceeding those in the Nuwayrat genome. However, like previous studies, we also find that all modelled Levantine Bronze Age groups trace 18.7-79.8% of their ancestry to previous Neolithic/Chalcolithic Levantine peoples^{46,48–51}. These results do not provide any evidence that the Mesopotamian ancestry in Nuwayrat arrived directly from Bronze Age Levantine groups, but we cannot exclude that intermediaries exist that are yet unsampled.” I miss the rationale behind this paragraph. Why is the existence of Neolithic/Chalcolithic Levant ancestry in later (BA) Levantine groups conflicting with the presence of Mesopotamia Neolithic ancestry? Also: wouldn’t evidence that Mesopotamian ancestry could have arrived through BA Levant be provided by an accepted model including BA Levant and Morocco_MN? That is not what was tested here.

“Like Middle Neolithic Morocco, this might indicate early shared regional ancestry.” Unclear. Ancestry from the eastern Fertile Crescent was not found in Morocco. Please rephrase.

“Bronze Age Caucasus, which is similar to the Bronze Age Levant ancestry detected in the Third Intermediate Period individuals⁴⁷. Other Bronze Age West Asian populations from the Levant or Anatolia cannot be rejected as potential sources when Bronze Age Caucasus is not used as a source.” Is there evidence (archaeological, for example) of migration/contacts with BA Caucasus after the third intermediate period? Do the authors mean to suggest a possible direct migration from the Caucasus into Egypt? I think these results lack a better discussion. given the justification that BA Levant ancestry is similar to the best fitting Caucasus BA ancestry. In fact, this is a good example of how I think the model selection criteria (while necessary) results in “best models” that should be better contextualised.

“Moreover, we note that there is a substantial diversity in ancestry across Egypt, as ~20% of present-day Egyptian genomes included here did not fit the model described above.” This heterogeneity could reflect some geographic structure within present-day Egypt. But there is no information on the regional origin (within Egypt) of these samples, and I think this is relevant information, omitted here.

Methods

“each skeletal element” different teeth? I am curious as for why the petrous bones were not used, if available.

In all truth, I am (very) unfamiliar with the literature cited, but I thought this was worth noting: it was only after reading the methods and supplements that it became evident to me that the results of the osteological analyses results are novel and reported first-hand in this paper (and not a collection of results published elsewhere). I think it has to do with the reference placement in the main text – articles describing the methods are cited after the results. These results are astonishing, so do make sure to take clear credit!

There are 100 Egyptian individuals sequenced in the cited article (Pagani 2015), not 216.

Importantly (maybe for several of my comments): What is the goal with using both proximal and distal sources for ancestry modelling? This should be better described.

“We estimated ancestry proportions of the two Third Intermediate Period Egyptians” Three individuals?

Basal Eurasian ancestry: please define CHS, ESN (also in Extended Data Fig. 7).

Conclusions

“...these results reveal ancestry links to North African-related groups“ ...North African groups

Figures

Maybe not a comment to the figures per se, but I find it very confusing that some sources seem to be arbitrarily chosen as both distal and proximal.

E.g. Anatolia_BA is modelled in Fig. 2b and in Extended fig 6, with vastly different results shown – but several of the populations used as sources are common between the two figures (including those that are accepted in the other figure), even if the models are intended as proximal and distal, respectively. Is this useful? How can it be interpreted?

Fig. 3a: At first I thought that each plot represented one individual. Other models have been presented based on their (best) ranking, why was that not done here? And mostly: considering that NUE001 is a mixture of Morocco_MN and

Mesopotamia_N, why including all these (redundant) sources? This plot makes it look like these individuals/populations could have very distinct demographic histories, or that a complete population replacement could have happened between NUE001 and the third intermediate period.

I suggest trying to simplify the qpAdm modelling, using a more consistent scheme – of source populations and respective colour – throughout the article.

Fig 2c and SupData9: How did Levant_BA populations score in the f4 statistics? I understand it is a good principle to observe chronological stratification of samples, but it would be informative. I don't think that a scenario of very recent admixture could be discarded.

Fig 2a PCA is very small (I can't identify Caucasian or South East Asian individuals, even though these are listed in the legend) and shows exactly what is expected in a PCA with world-wide populations, so not very revealing. I think Extended Data Fig 4 is much more informative. Consider displaying it in the main text instead.

Extended Data Fig. 5: How many K are represented in this figure? And why was this the chosen K to display? Is this the full set of populations used for ADMIXTURE analysis?

Please present all the K of the full analysis, I didn't find it in the supplements either. I also find the colour scheme confusing, especially because there are several components that are only represented in very low proportions in some individuals (so it is hard to identify their affinities) and there are too many shades of the same colour. For example, there is a blue component in NUE001 that I don't seem to identify in any other individual?

Supplements

Sup info 4

"The proportion of Zagros or Mesopotamian components (or Caucasus component, also similar to Zagros Neolithic4) increase eastward (in the Levant and Anatolia), while the Epipaleolithic Morocco ancestry disappears (Extended Data Fig. 6, Supplementary Data Table 5)." In Extended Data Fig.6, Iran-Neolithic ancestry is depicted, but it is not mentioned in the modelling methods (always Zagros). If it is the same set of samples, use a consistent label.

"74.0 ± 7.6% Middle Neolithic Moroccan, 21.2 ± 4.2% Neolithic Mesopotamian, and 4.7 ± 8.1% Neolithic Levantine ancestries (p = 0.255). This alternative model corroborates a scenario where the West Asian-related ancestry detected in the Nuwayrat individual is still most directly derived from Neolithic Mesopotamia and not from the two sources from which it derives, namely Neolithic Levantine or Caucasus/Zagros ancestries". Do you mean to say that Neo Mesopotamian ancestry derives from both Neo Caucasus and Zagros Neo? If so, please rephrase for clarity. But most importantly, why does this corroborate such scenario? This model actually has a higher p-value and would make sense from a migration route perspective.

And also: could this result reflect a bit of overfitting? The 3-source models with Morocco Middle Neolithic, Neolithic Levant, Neolithic Caucasus/Zagros fits the data as well, so I am not totally convinced about the conclusion that Nuwayrat's ancestry derives directly from Mesopotamian Neo. Including both Morocco_MN and Levant_Neolithic/Chalcolithic could have the same effect (overfitting)?

"This result is consistent with the previous distal model (Extended Data Fig. 6)" This model shows Iran Neolithic (Zagros?) instead of Mesopotamia Neolithic, even though the latter was also used as a distal source, so I don't think consistency can be claimed.

"When this reference is favoured to represent the eastern Fertile Crescent ancestry, ~13-16% ancestry is attributed to Neolithic Chalcolithic Levant." ...Neolithic/Chalcolithic Levant?

I am surprised to see that when you try to control for overfitting the model, the "Distal model" (Neolithic Levant, Epipaleolithic Morocco and Neolithic Zagros) previously accepted no longer works. I can only imagine that this is caused by populations that are now in the left populations set being rotated of the right (outgroup) set. What are the implications, if that is the case?

"the inclusion of Bronze Age Levant as an outgroup is expected to result in a lower p-value if the Bronze Age Levant in truth received gene flow from Nuwayrat-related sources." How would Noth African (related to Morocco_MN) ancestry of Nuwayrat not be observed in Levant_BA if this had happened?

"We find that the Bronze Age Levant can marginally be modelled as a source for Nuwayrat" I think that this result is worth further exploring. Can a recent admixture from BA Levant be completely excluded? Admixture dating (Morocco MN+Mesopotamia N / Morocco MN + Levant BA) could be revealing.

"We also detect ancestry from Neolithic Mesopotamia at three archaeological sites (Ebla, Baq'ah, Ashkelon)" ...at three Levantine archaeological sites...

"A part of this ancestry ultimately derives from Caucasus Neolithic-related ancestry in Neolithic Mesopotamia (Extended Data Figure 5)8,10,11, this could explain why Neolithic Mesopotamia can be a possible source with an increased proportion

compared with the best-fit model for Nuwayrat." Unclear, please rephrase.

G. References: appropriate credit to previous work?

References are appropriate and credit previous work.

H. Clarity and context: lucidity of abstract/summary, appropriateness of abstract, introduction and conclusions

The manuscript is clear, easy to follow and well written. Suggested improvements are listed in section F.

Referee #3

(Remarks to the Author)

This is an interesting article and appears well written with good supporting figures, but my review focuses solely on the radiocarbon dates as requested by the Editor, as this is my area of expertise.

The radiocarbon dates are a minor part of this study, and are presented primarily to support the age of the individual whose DNA was analysed. Within the main text of the paper, the combined radiocarbon age (from 3 dates, 2 from teeth dated as part of this study, and one previously published from the femur) is given in Fig. 1d as 2851-2573 cal BCE. However, this is an over-simplification, as calibration of the combined date results in 3 narrower ranges (2851-2809 cal BCE (20.6%), 2747-2727 cal BCE (5.2%) and 2698-2573 cal BCE (69.7%). It is not statistically correct to state that, for example, 4080 ± 30 BP calibrates to 2857-2492 cal BCE at 95.4% probability. Within the main text, this may be acceptable, but the specific date ranges should be included and explained in the supplementary information. Note that the legend to Figure 1 describes it as a summary of genomic data, without mentioning the C14 date. As stated though, the radiocarbon date is not the main focus of this study.

Within the supplementary data table 2, I would expect the version of calibration software used to calibrate and combine the dates (e.g. OxCal v4.4.4) and the calibration curve (presumably IntCal2020; Reimer et al (2020) to be stated. I recalibrated the dates on the teeth and the femur, and got the same overall ranges (at 95% probability) for the teeth but not for the femur. This suggests that they may have been calibrated using different software and/or datasets (especially if the femur calibration was cited directly from Dee (2016), prior to the release of IntCal20). Ideally, this should be recalibrated for effective comparison.

I could not access either of the references cited for the femur date (refs 64, 65) as the links took me to a Paperpile file that I could not access, and I could not easily find the documents online. However, I note that Mike Dee (an expert in C14 dating, especially from Ancient Egypt) has assisted with the date combination according to the acknowledgements. I wonder if he might be contacted to clarify the calibrated date range for the femur, and also to advise the authors on inclusion of the individual date ranges rather than stating the continuous calibrated range. As the CN ratios are included for the teeth, it would be helpful if Mike could also provide the CN for the femur (which should be readily available, although I would anticipate it would be in one of the papers cited) - that measurement provides confidence in that date.

I'm not sure why the date on the textile was included (line 6 of the table in Supplementary Data 2) as it is clearly not contemporaneous with the skeletal remains and is not discussed in the paper; I would remove this.

Could 'source' (Column L) be clarified - is this a Pers. Comm with Mike Dee, or should Dee (2016), for example, be cited?

Referee #4

(Remarks to the Author)

Summary of the key results: result push back evidence of aDNA in the Nile valley by a significant margin, capturing the dawn of Egyptian civilization, a key marker in early stages of this civilization. Other rare successful result (n=3 individuals), are much later, less comprehensive, and do not offer such insights. Although this is based on one individual, the finding is highly significant as (a) aDNA rarely survives in the Nile valley, (b) it captures a period that is key to the emergence of the Ancient Egyptian civilization and (c) the data is more comprehensive and robust than the previous study (much later, see lines 68-69 and 221-222) data. Importantly, for this early period of state formation, it is the first direct evidence of what has been hinted at by indirect genetic proxies such as dental morphology, confirming what these other data have tentatively suggested. It is also, for the first time, the key input from the fertile crescent within a stable North African population group, laying much arguments to rest.

Originality and significance: if not novel, please include reference: This is highly significant despite being based on one individual (see above). In view of the rarity of aDNA in the Nile valley, the fact that this is a single individual should not be held against it. It supports what the material culture has suggested, particularly the influx of ideas from the Fertile Crescent into Egypt from the Predynastic period onwards being accompanied with some population movement (e.g. the use of Bull Imagery reveals such an influence in the famous Predynastic tomb 100 decorations, the earliest wall paintings known to date from ancient Egypt, now at the entrance of the Cairo Museum) (i.e. cultural diffusion associated with the movement and settlement of people).

Data & methodology: validity of approach, quality of data, quality of presentation: all appear robust but the aDNA is not my expertise. Other bioarchaeological interpretations are sound.

Appropriate use of statistics and treatment of uncertainties: N/A

Conclusions: robustness, validity, reliability: All appear to be very strong

Suggested improvements: experiments, data for possible revision: perhaps add a clear paragraph (just a few lines) on how the data compares/differs from Sudanese and other 'sub-saharan' populations. I would also expand (again, just a few lines) on the population shift that appears to occur between the Mesolithic and Neolithic (something that is alluded to but would benefit the clarification for the general readers). Also perhaps expand/clarify (one or two line) what indirect sources of population biology affinity and relatedness have suggested to date (dental morphology etc...).

References: appropriate credit to previous work? yes

Clarity and context: lucidity of abstract/summary, appropriateness of abstract, introduction and conclusions. All appear to be clear and appropriate.

Version 1:

Reviewer comments:

Referee #1

(Remarks to the Author)

The authors have addressed my concerns and I have no further feedback. I recommend this manuscript for publication.

Referee #2

(Remarks to the Author)

I am overall very happy with the reviewed version of this manuscript. The authors have made significant improvements, not only according to reviewers advice, but also (and commendably) increased the genomic coverage of the Nuwayrat individual to 2x. This enabled improved results, including clearer ancestry modelling results and resolving phenotypic traits, for example. The main text is notably clearer and the discussion of the results much better contextualised and matured.

I noted, however, a couple of minor details that should be fixed before publication.

First and most importantly:

I received a version of the Supplementary Data Tables that has not been updated. For example, supplementary data table 5 still reports distal models using Mesopotamia Neolithic as a source, which as I understand, was not used as a distal source in this reviewed version. Also, still using "Iran_Neolithic" in this table; the ancestry proportions in sup data table 6 do not correspond to those reported in the main text. Please make sure to update this file.

L259 Please specify that "the initial full qpAdm model" is the initial full qpAdm model for NUE001. It was not completely clear to me, as this sentence follows some modelling results for Bronze Age Levantine sites.

L285 "Early Neolithic shared regional ancestry" It should be clearer that regional here means across North Africa. To illustrate this, it would be useful and interesting to refer to the distal qpAdm ancestry modelling results (which are now completely omitted in the main text). E.g., refer to Extended Data Fig 6 here, that shows broadly similar ancestry components for NUE001 and Morocco_MN, in spite of the absence of Zagros Neolithic in Morocco.

L547 "Caucasus_Neolithic wer used as rotating sources" ... *were* used as rotating sources

Referee #4

(Remarks to the Author)

Changes are appropriate and have clarified several elements of the paper, which is now stronger, no further comments.

Referee #5

(Remarks to the Author)

I was asked to check if the authors have answered correctly and sufficiently to the reviewer #3 (the Radiocarbon Dating scholar) points.

They authors answer the points of the reviewer and this would probably be sufficient to accept the paper for publication.

I have another suggestion in order to retrieve more information from the radiocarbon dates obtained and reduce the calibrated range.

The authors could build different Bayesian models considering that:

1. dentine 14C reflects the first 10-15 years of life
2. the person is around 60 years old

3. the femur 14C reflects the last 10-15 years of life.

4 use a gap or different gaps and see what kind of calibrated range become most probable.

Although the radiocarbon dating is not the major focus of this paper, yet the authors do mention the possible synchronization with the archaeological and historical record. I did the model and in my opinion it has improved the chronological issue. Such suggestion is not a "MUST" to get the paper accepted, but it would improve in general the meaning of the paper. Hope this helps.

Elisabetta Boaretto

Referee #6

(Remarks to the Author)

A. This article presents new and exciting genetic data from the Early Dynastic/Old Kingdom. As the authors clearly state, this pushes back our genetic understanding of ancient Egypt significantly. I very much appreciate the multi-method approach, including aDNA, isotopes, osteology, etc. Approximately 20% of this individual's genetic ancestry can be traced to the Near East. While this isn't altogether surprising, given the history of ancient Egypt--it is important to be able to quantify this connection and present empirical evidence for genetic relationships (even at Nuwayrat).

B. Of course, there have been many attempts to study aDNA in ancient Egyptian individuals. However, this has been frequently unsuccessful, due to preservation issues. Thus, the whole-genome data presented here from such an early date (Early Dynastic/Old Kingdom) is truly original and significant.

C. The data and methodology, are all sound. I'm specifically focusing on the osteological approach, but I have no suggestions here. Every method and citation is appropriate and applied accordingly.

D. Yes.

E. The conclusions are appropriate and fitting. I like the added value of the unique burial practice (jar burial), may have played a special role in preserving the aDNA. This, in my opinion, could help guide future aDNA research in the Nile Valley.

F. None.

G. Yes; again, I'm focusing mainly on osteology, but all references are good.

H. Very clear and well contextualized.

Great article--very nice work! and a great team!

Referees' comments

Referee expertise:

Referee #1: population genetics, aDNA

Referee #2: population genetics, aDNA

Referee #3: radiocarbon dating

Referee #4: Egyptology

Referees' comments:

Referee #1 (Remarks to the Author):

Morez et al. present a 1.51X whole genome of an individual from Nuwayrat, Egypt dating to the Early Dynastic-Old Kingdom period. This makes this genome the oldest genomic sequence from Egypt, thus enabling first insights - barring sample size limitation - into the genetic diversity during the early dynastic periods. The manuscript engages primarily with the genomic data and the presented analyses are fairly standard for an ancient DNA paper dealing with a single low coverage genome that precludes more involved population genetics methods. Based on the genomic analyses, the authors report that the sequenced Nuwayrat individual had genetic ancestry related primarily to North African Neolithic and eastern Fertile Crescent groups (best proxy being Neolithic Mesopotamia). The Nuwayrat individual's ancestry may have contributed in part to later Third Intermediate period genomes as well as several present-day Egyptians, though these groups have received substantial and later gene flow from genetic sources related to Bronze Age Levant and ancient East African groups, respectively. Additionally, there are osteological and facial reconstruction analyses included, the former suggesting that despite the burial context hinting at this individual's higher social class they were likely involved in a labor-intensive occupation. Overall, this manuscript pushes past the methodological challenges inherent in a region such as Egypt and generates whole-genome data, which is commendable.

I have been through the genomic analyses presented in this manuscript. I am quite happy with the careful manner in which the data QC and filtering has been performed. I am also satisfied with the descriptions and implementation of the analyses that are feasible given the generated data. The authors have been thoughtful in the qpAdm analytical set-up, which forms the core of the results in this study. I believe they have accounted for the distal and proximal sources with the available aDNA data and show fairly convincingly the sources that are able to model the Nuwayrat individual, in addition to considering alternate models and possibilities of overfitting due to one of the sources being a compound in itself. I do not have any major concerns as such but list a few minor suggestions/questions below:

We thank the reviewer for their positive comments and constructive feedback.

Title: "Whole-genome ancestry" sounds a little awkward to me, perhaps: Whole-genome sequencing of an Old Kingdom Egyptian.

We thank the reviewer for their suggestion regarding the title. However, we feel that 'Whole-genome ancestry' better captures the focus of our study on the ancestral insights derived from the genome, which goes beyond sequencing alone.

Supp Table 1 lists information on 11 libraries but main text talks about 7 extracts - did some extracts have multiple libraries? Can the dropped libraries be marked clearly? Several columns in Supp Table 1 are somewhat unclear - it would be good to have descriptions of the headers.

Two libraries were sequenced several times (information added L148). The information on the two libraries that were discarded is added to L148. We also added a column in Supplementary Data Table 1 to indicate the library ID and the sequencing platform used to generate each sequencing run. We also added the description of the columns in Supplementary Data Table 1 (L498-533).

I was unable to locate a figure corresponding to the ROH analysis.

Based on the more extensively sequenced genome (2X), no ROH segments longer than 4cM were detected. We modified this result on L164-165.

Please standardize the labels for Zagros_N and Iran_N, which are used interchangeably in the qpAdm analysis (e.g. in Extended Data Fig 6 where both are used to indicate source in the figure and text legends).

Thank you. We updated the figure with 'Zagros_Neolithic'.

I was unable to locate the qpAdm results for the tests noted on page 6 (overfitting due to the use of Middle Neolithic Morocco) and page 7 (BA Levantine as sources for the Nuwayrat individual).

Concerning 'overfitting due to Morocco_MN', we detail the qpAdm models in Supplementary Information 4 L138-159. Concerning the tests on 'BA Levantine as sources', we detail the qpAdm models L124-137 of Supplementary Information 4 and point to it in the main text on L259-263. However, in both cases, no model passed the significance threshold, so no results are provided in the Supplementary Data Table.

In the supplementary text describing the f4 ratio for investigating Basal Eurasian, it should note Extended Data Fig 7 instead of 6.

We corrected the text according to the reviewer's comment.

Referee #2 (Remark to the Author):

Does the manuscript have flaws which should prohibit its publication? No.

To my knowledge, the conclusions are original.

The results presented are of immediate interest to geneticists, archaeologists, and anthropologists.

Outstanding features:

Morez and colleagues present new whole genome data from an ancient individual excavated from Nuwayrat, Egypt. The analysed archaeological remains show exceptional preservation for that region and period. Genomic data from ancient Egypt has thus far only been retrieved from three individuals – and even those are of poorer quality and from a more recent period. The preservation of the whole skeleton has allowed for a multidisciplinary analysis of this individual, including a thorough osteological examination that reveals a great deal about this man's life, including his potential occupation, and a facial depiction.

General notes:

I applaud the authors for generating complete genome data for this rare individual, instead of enriching for a SNP panel.

We thank the reviewer for pointing this out, and we completely agree with the added value of complete genome data, reflected in our reference to "Whole-genome" in the title.

The authors caution for the fact that this study is based on a single individual. Small sample sizes are common in ancient DNA studies, especially those based on sites with human remains with documented low molecular preservation, such as Egypt. The assumption for extrapolating such results to a population level is usually based on the individuals' expected representativeness of the general population genetic background. My concerns about the generalisation of the conclusions in this study are based on this individual's potential higher social status, which could hinder that extrapolation. I think that this should be more obviously stressed in the text.

We agree, and thank the reviewer for commending our discussion on the limited sample size. We have extended the text about the limitations due to social status.

The text now reads: "While our analyses are limited to a single Egyptian individual who, based on his relatively high status burial, may not be representative of the general population, [...]" on L370-372.

I am surprised to note the absence of stable isotope analyses such as dietary isotope analysis and/or strontium analysis. The rarity of the specimen would justify investigating these aspects, which would in turn provide a greater depth to the manuscript. These could substantiate the claim of the higher social status of this individual, by possible comparison of

dietary patterns with contemporary Egyptians and perhaps resolve if the individual was born locally or was a migrant.

We agree. In the revised version of the manuscript, we now include an extensive report of several isotope analyses ($\delta^{13}\text{C}$, $\delta^{15}\text{N}$, $\delta^{18}\text{O}$, $^{87}\text{Sr}/^{86}\text{Sr}$) on L130-139 in the main text and in Supplementary Information 5. Overall, the results are consistent with the individual being local.

I would expect, in the context of a culture as well-studied as the ancient Egypt (by many disciplines before archaeogenomics), a more robust and contextualised discussion, including perspectives and background from available literature/evidence (such as archaeological). I think that improving this aspect would round up the paper nicely and make it more interesting to the general audience, considering how popular ancient Egypt is. As an example, the conclusions paragraph is rather dull and adds little to no information to the manuscript.

“These genetic links have ramifications for discerning early Dynastic population history, opening up the possibility that the incorporation of objects, images and ideas from outside Egypt was concomitant with some settlement of people.”

Also, in the Summary “contacts between Egypt and the eastern Fertile Crescent were not limited to objects and imagery”

Could the authors give examples? Is there (archaeological) evidence of objects, images or ideas that were incorporated from the regions that seem to have been connected to Egypt through the movement of people, given the genomic results?

We have expanded sections dealing with wider archaeological data and context and included additional references L52-53, L298-301 and L374-379, in addition to material previously included within the main text (e.g., L58-74, L242-249).

Lastly, and related to the previous point, I am not totally convinced about Mesopotamia Neolithic ancestry being the most parsimonious source of ancestry present in NUE001. On PCA (Extended Data Fig 4), for example, NUE001 is clearly positioned between Morocco MN and Levant Neolithic/Chalcolithic, not towards Mesopotamian Neolithic. Unsupervised ADMIXTURE could be very informative, but unfortunately the figure shown is undecipherable and unhelpful. It is cautioned that “surrounding regions” are also potential sources. However, a rather strong emphasis is put on Neolithic Mesopotamia and I am concerned that this result is based mostly on the criteria implemented for model selection in the qpAdm framework, namely the minimal number of source populations criterium. Noticeably, the model selection criteria used follow recently published guidelines for the application of a qpAdm framework. However, at times two-source models are preferred to three-source models, even when the latter could possibly make more sense from a population history dynamics perspective. This seems particularly relevant here given the complex Fertile Crescent demographic history and the (already) admixed nature of the source populations considered – especially proximal. Or perhaps all that is missing is contextualizing the dynamics behind the arrival of Mesopotamia Neolithic-related ancestry to Egypt – and how/why geographically and chronologically closer populations don't seem to be better proxies for that ancestry. Please see more specific comments/suggestions below.

We agree with this concern. In the revised version, we fully updated all analyses using newly

generated sequences, now reaching 2X coverage, which increased our statistical power to resolve this result. The new key result is that only a single two-source model passes rejection, which includes Morocco_MN and Mesopotamia_N. In turn, the two three-source models passing rejection confirm the two-source model, with Morocco_MN- and Mesopotamia_N-related ancestries present in similar proportion, with a small addition of Levant_N/ChI (4.7+/- 8.2% or 1.1+/-8.7%). Given these results and that of the distal model (Extended Data Fig 6), we can confirm the presence of ancestry ultimately deriving from the eastern Fertile Crescent. We chose the term 'eastern Fertile Crescent' in our interpretation, instead of Mesopotamia, to emphasise the uncertainty of precisely determining the source population.

Dating the admixture event would be key to answering questions related to the timing and location of the true source here. While we lack the statistical power to date the admixture event confidently, archaeological evidence is consistent with both an older and/or a more recent genetic contact; we discuss both possibilities in the main text (L242-251, L266-303). We agree with reviewer 2 that, despite that our model points toward Mesopotamia Neolithic as the best minority source for the Nuwayrat ancestry (with Morocco_MN the majority source) given the current set of reference populations, 1) the admixed nature of the BA and pre-BA West Asia/North Africa sources is a challenge that makes differentiation between proximate sources difficult and 2) we cannot exclude that unsampled geographically and chronologically closer populations are the most proximal sources. We highlighted this limitation L263-265.

Concerning the PCA interpretation, we would refrain from drawing major conclusions from a PCA alone. First, the position of low-coverage genomes is notoriously imprecise. Then, the Morocco_MN source that makes up ~80% of NUE001's ancestry is likely a proxy for a local Neolithic source. If we imagine that this source carried a slightly higher proportion of Levantine ancestry than Morocco_MN, its position in the PCA space could be shifted towards Paleolithic/Neolithic Levantine, and in this case, the Nuwayrat genome would be aligned on a Mesopotamia/local Neolithic cline.

General comments:

The use of line and page numbers would certainly help the reviewers.

We apologise for this omission and hope that reviewer 2 finds the revised version helpful.

A map with labels of the broader regions mentioned in the text, such as Fertile Crescent, Mesopotamia, etc. would be very helpful (I'm thinking perhaps something as seen in Lazaridis 2022 and Altınışık 2022). A more detailed timeline, including other periods mentioned in the text (e.g. Bronze Age is extensively mentioned) is essential.

These are very good suggestions and we modified Figs. 3 and 4 to incorporate geographic details and a timeline of cultural transition, based on Egyptian civilisation chronology.

A. Summary of the key results

Morez and colleagues present new whole genome data from an ancient individual excavated from Nuwayrat, Egypt. The authors stress the North African- and Fertile Crescent-related ancestry in the studied ancient Egyptian. This finding provides additional and important evidence on the links across North Africa and the Fertile Crescent, that now are shown to go

beyond culture and trade, but also include migration. The unusual preservation of the whole skeleton has allowed for a multidisciplinary analysis of this individual, including a thorough osteological examination that reveals a great deal about this man's occupation and a facial depiction.

B. Originality and significance: if not novel, please include reference

The analysed archaeological remains show exceptional preservation for that region and period. Genomic data from ancient Egypt has thus far only been retrieved from three individuals – and even those are of poorer quality and from a more recent period. The newly generated data will be useful for many future studies.

C. Data & methodology: validity of approach, quality of data, quality of presentation

The authors use robust and well-established methods in ancient DNA research for generating and analysing genomic data from the ancient Egyptian individual through direct shotgun sequencing.

D. Appropriate use of statistics and treatment of uncertainties

All according to well established methods frequently used in ancient DNA analyses.

E. Conclusions: robustness, validity, reliability

The overall conclusions and the findings are sound, although at times lacking a stronger discussion/contextualisation.

F. Suggested improvements: experiments, data for possible revision

I list some suggestions and comments, which I hope the authors find helpful:

Summary

"Nuwayrat (Nuerat, نويرات)." Nuwayrat (Nuerat, نويرات).

We hope you can now find this typo corrected, we could not see it in our original manuscript.

"Most of his genome is represented by North African Neolithic ancestry"

While not incorrect, the multitude of ancestries present in Neolithic North(west) Africa makes this statement ambiguous. In the distal ancestry modelling, the majority of his genome is in fact represented by Levantine ancestry, and truly, so is the "North African Neolithic ancestry" mentioned here (I assume Morocco Middle Neolithic).

Thank you for highlighting this ambiguity. We hope that the modifications on L44-45 are satisfying while keeping the information concise as requested for the abstract section. We discuss in more detail the Middle Neolithic Morocco ancestry in the text (L201-211) and back up the claim that Middle Neolithic Morocco is the current best representative for part of the Nuwayrat ancestry with further testing on potential over-fitting of Middle Neolithic Morocco in Supplementary Information 4 L138-159.

Main text

"The population has been said to be of local origin" based on what kind of evidence, if not archaeological, as that seems to be excluded, based on the next sentence?

This is indeed based on older archaeological evidence. We clarified this on L65-69. However, more recent studies showed a trade connection with the eastern Fertile Crescent. Additionally, these recent studies refer only to the diffusion of material culture. No inference on whether this was associated with demic diffusion could be made.

“Exchange continued through the late 4th millennium BCE with the growing Sumerian civilisation of Mesopotamia” Please specify the kind of exchange meant.

We modified the sentence on L69 to clarify that we are referring to cultural exchange.

“which some maintain can provide clues concerning occupation” some...authors?

We clarified this on L122.

“using a fully rotating model competition approach” It would be useful to include a very brief description of what this means.

We added a short description on L173-179.

“a single two-source model met the significance criteria ($p = 0.207$), which includes $78.4 \pm 4.0\%$ ancestry most closely related to genomes from the Middle Neolithic Moroccan site of Skhirat-Rouazi dated to 4,780-4,230 BCE (Morocco_MN), and the remainder most closely related to genomes from 9,000-8,000 BCE Neolithic Mesopotamia ($21.6 \pm 4.0\%$).” Both source populations have received geneflow from the Levant. Could this be problematic, eventually incurring into overfitting the model?

We shared this concern and detailed further qpAdm models trying to resolve this potential issue in Supplementary Information 4. The most relevant are:

1) L138-159: When replacing *Morocco_MN* with *Morocco_Epipaleolithic* in order to let another potential source for the Levantine ancestry fit more freely, we find that Nuwayrat cannot be modelled at all (no models pass rejection), even when the potential sources include *Levant_BA*. This confirms that part of the Levantine ancestry in Nuwayrat is best derived from a Morocco_MN-like population. As a consequence, Morocco_MN as source 1 should limit the possibility of wrongly picking Mesopotamia_Neolithic as the source for the eastern Fertile Crescent ancestry. Moreover, if another source population from the rotating set would be an equally good or better source for Nuwayrat, and includes the same ancestries as Mesopotamia but in different proportions, we expect this model to pass $p > 0.05$.

2) L124-137: When *Levant_BA* is added to the initial ‘proximal’ model, no model passes rejection. While this could suggest that *Mesopotamia_Neolithic* and *Levant_BA* are equally good proxies for the second source, and thus mutually excluding each other, it is interesting to note that the model Morocco_MN+Mesopotamia_N has a slightly higher ranked p-value than Morocco_MN+Levant_BA.

Overall, in light of those results, and the currently available samples, Morocco_MN+Mesopotamia_N is the best model fitting the Nuwayrat genome. Concerning, the biological interpretation, we cannot formally draw a precise time and location for the source of this ultimately eastern Fertile Crescent ancestry (as discussed on L242-265 of the Main Text).

“pre-Bronze Age genomes are not available for the rest of the Maghreb” Not true for the Maghreb.

We modified the sentence on L211-213.

"from Mota, Ethiopia, or any other individuals in central, eastern or southern Africa (Fig. 2, Extended Data Fig. 5)" I only see Ethiopia 4500BP in Fig2 and also South Africa 2200 BP in extended Data Fig. 5. It would be very interesting to see if there is shared ancestry between NUE001 and pastoral Neolithic East Africans.

We agree with the reviewer and complemented our study with a section on ancient East African modelling using NUE001 (L304-311). A more extensive report is placed in a dedicated section of Supplementary Information 4 L224-241.

"other studies suggesting gene flow from the Mesopotamian and Zagros region into surrounding regions, such as Anatolia, during the Neolithic⁴⁴" A single study is cited.

The text was amended following the reviewer's recommendation to include additional supporting references.

"wider expansion processes and an increase in mobility from the Mesopotamian region, into Egypt as well as Anatolia." It is puzzling for me how such a scenario of mobility from Mesopotamia would not bring Levantine ancestry, admixed into the migrating population on its way to Egypt. Please elaborate. Is an alternative route of expansion (not encompassing the Levant) possible or supported by other lines of evidence?

This is correct. We agree with the reviewer that this information is essential and interesting. One of the main hypothesised trade routes included seafaring through the Mediterranean and the Red Seas, and not through the Sinai Desert. We added this information on L298-301.

"To investigate the presence of Mesopotamian ancestry in Nuwayrat relative to that in Bronze Age contemporaries of the latter, we extended the qpAdm model to adjacent regions." It is not clear to me what was done, please rephrase.

We rewrote the sentence and this now reads: "Related movements may have introduced the Mesopotamian-like ancestry more recently in Egypt. We tested this by applying the same full qpAdm model to target groups from Bronze Age Anatolia and Levant" on L250-254. We hope that the reviewer finds it clearer.

"We detect ancestry from Neolithic Mesopotamia at three sites (Ebla, Baq'ah, Ashkelon), in proportions (41.8-54.8%) exceeding those in the Nuwayrat genome. However, like previous studies, we also find that all modelled Levantine Bronze Age groups trace 18.7-79.8% of their ancestry to previous Neolithic/Chalcolithic Levantine peoples^{46,48-51}. These results do not provide any evidence that the Mesopotamian ancestry in Nuwayrat arrived directly from Bronze Age Levantine groups, but we cannot exclude that intermediaries exist that are yet unsampled." I miss the rationale behind this paragraph. Why is the existence of Neolithic/Chalcolithic Levant ancestry in later (BA) Levantine groups conflicting with the presence of Mesopotamia Neolithic ancestry? Also: wouldn't evidence that Mesopotamian ancestry could have arrived through BA Levant be provided by an accepted model including BA Levant and Morocco_MN? That is not what was tested here.

We agree with the reviewer and we reworked this paragraph in order to clarify our claims and emphasize the limitations of our model given the admixed nature of the sources. Please find the amendments on L250-265.

“Like Middle Neolithic Morocco, this might indicate early shared regional ancestry.” Unclear. Ancestry from the eastern Fertile Crescent was not found in Morocco. Please rephrase. We rephrased the sentence on L284-285.

“Bronze Age Caucasus, which is similar to the Bronze Age Levant ancestry detected in the Third Intermediate Period individuals⁴⁷. Other Bronze Age West Asian populations from the Levant or Anatolia cannot be rejected as potential sources when Bronze Age Caucasus is not used as a source.” Is there evidence (archaeological, for example) of migration/contacts with BA Caucasus after the third intermediate period? Do the authors mean to suggest a possible direct migration from the Caucasus into Egypt? I think these results lack a better discussion. given the justification that BA Levant ancestry is similar to the best fitting Caucasus BA ancestry. In fact, this is a good example of how I think the model selection criteria (while necessary) results in “best models” that should be better contextualised. We regenerated these models using the more deeply sequenced version of NUE001 and increased the number of included SNPs. The updated models favour Levant_BA instead of Caucasus_BA, which is consistent with the observed increase of Levant_BA ancestry in the Third Intermediate genomes.

“Moreover, we note that there is a substantial diversity in ancestry across Egypt, as ~20% of present-day Egyptian genomes included here did not fit the model described above.” This heterogeneity could reflect some geographic structure within present-day Egypt. But there is no information on the regional origin (within Egypt) of these samples, and I think this is relevant information, omitted here.

We also think that investigating potential geographical structure would be extremely interesting. Unfortunately, this information is not publicly available from the dataset used in this study, i.e. Human Origin and Pagani *et al.* 2015.

Methods

“each skeletal element” different teeth? I am curious as for why the petrous bones were not used, if available.

The specific tooth used is detailed in Supplementary Data Table 1. We decided not to use petrous bones because the teeth were both numerous and relatively well-preserved. In many cases, if teeth are macroscopically well preserved the cementum-enriched root tips can yield DNA comparable to that from petrous bones (see Hansen *et al.* 2017 <https://doi.org/10.1371/journal.pone.0170940>). Given the high value of petrous bones for other bioarchaeological analyses, we chose to preserve them when alternative samples, like teeth, were available.

In all truth, I am (very) unfamiliar with the literature cited, but I thought this was worth noting: it was only after reading the methods and supplements that it became evident to me that the results of the osteological analyses results are novel and reported first-hand in this paper (and not a collection of results published elsewhere). I think it has to do with the reference placement in the main text – articles describing the methods are cited after the results.

These results are astonishing, so do make sure to take clear credit!

This acknowledgement is deeply appreciated and we thank the reviewer for supporting further efforts to credit the authors' work. We added a modification on L114 to clarify this.

There are 100 Egyptian individuals sequenced in the cited article (Pagani 2015), not 216. We thank the reviewer for pointing out this mistake in the text, which has been amended.

Importantly (maybe for several of my comments): What is the goal with using both proximal and distal sources for ancestry modelling? This should be better described.

Supplementary Information 4: L31-42 and L52-58 for a description of how and why the different model setups were chosen.

In a nutshell, while a distal model helps get a coarse idea of which ‘main’ divergent ancestries are present in NUE001, the idea behind keeping the distal populations and adding them to the full one allows us to gradually build a more fine-detailed model, while conserving clarity for how deeper admixture events may have shaped ancestries of the oldest genetic sample from Egypt.

We now refer to those models as ‘full’ instead of ‘proximal’ to help clarify their aim.

“We estimated ancestry proportions of the two Third Intermediate Period Egyptians” Three individuals?

There are two Third Intermediate Period Egyptian genomes and one Ptolemaic period genome from Schuenneman et al. 2017. In this study, we re-analysed only the two Third Intermediate Period Egyptians because of the noticeable contamination subsequently found in the Ptolemaic genome (contamination on the X-chromosome = 2.2-12.4%, as reported in the Allen Ancient DNA Repository metadata).

Basal Eurasian ancestry: please define CHS, ESN (also in Extended Data Fig. 7).

The definitions of CHS and ESN have been added in the legend of Extended Data Fig 7.

Conclusions

☑ “...these results reveal ancestry links to North African-related groups“ ...North African groups

Corrected L373.

Figures

Maybe not a comment to the figures per se, but I find it very confusing that some sources seem to be arbitrarily chosen as both distal and proximal.

E.g. Anatolia_BA is modelled in Fig. 2b and in Extended fig 6, with vastly different results shown – but several of the populations used as sources are common between the two figures (including those that are accepted in the other figure), even if the models are intended as proximal and distal, respectively. Is this useful? How can it be interpreted?

We agree with the reviewer that the word ‘proximal’ here is misleading. We replaced this with ‘full’ model. While the distal model intends to estimate the presence and proportion of a set of ‘deep’ lineages linked to the Epipaleolithic/Neolithic, the full model build-up from the distal one, including more proximal and related sources to get a refined understanding of the more direct sources. Since we do not have any older representative for Egypt than the Nuwayrat genome, these older samples are key to investigating deep population movements

in Egypt. We also removed *Mesopotamia_Neolithic* from the distal model, given its admixed nature and found highly consistent results for *Anatolia_BA* in the distal and full model

Fig. 3a: At first I thought that each plot represented one individual. Other models have been presented based on their (best) ranking, why was that not done here? And mostly: considering that NUE001 is a mixture of *Morocco_MN* and *Mesopotamia_N*, why including all these (redundant) sources? This plot makes it look like these individuals/populations could have very distinct demographic histories, or that a complete population replacement could have happened between NUE001 and the third intermediate period.

We added the result of the suggested model in the Supplementary Information 4 L215-223. Briefly, the rotating *qpAdm* models without *Morocco_MN* and *Mesopotamia_N* result in no model passing $p > 0.05$. We interpret this as another line of evidence showing that *Morocco_MN* ancestry was likely widespread in Egypt and not only restricted to the Nuwayrat genome so that later genomes from Northern Egypt received it. Slightly different mixture proportions of these two ancestries across Egypt can result in *Morocco_MN* being a better fit than NUE001.

We agree that for consistency, only the model selected as best fit should be displayed.

I suggest trying to simplify the *qpAdm* modelling, using a more consistent scheme – of source populations and respective colour – throughout the article.

We hope that the previous replies to the reviewer's comment helped clarify the choice of source populations in the model.

Fig 2c and SupData9: How did *Levant_BA* populations score in the *f4* statistics? I understand it is a good principle to observe chronological stratification of samples, but it would be informative. I don't think that a scenario of very recent admixture could be discarded.

We added *Levant_BA* to the *f4*-statistics. $f_4(\text{NUE001, MoroccoMN; LevantBA, JuHoan})$ derives significantly from 0 ($Z > 2$), but does not deviate as much as when *MesopotamiaN* is placed as X ($Z > 3$).

Fig 2a PCA is very small (I can't identify Caucasian or South East Asian individuals, even though these are listed in the legend) and shows exactly what is expected in a PCA with world-wide populations, so not very revealing. I think Extended Data Fig 4 is much more informative. Consider displaying it in the main text instead.

We agree with the reviewer and modified the Fig. 2 accordingly.

Extended Data Fig. 5: How many K are represented in this figure? And why was this the chosen K to display? Is this the full set of populations used for ADMIXTURE analysis?

Please present all the K of the full analysis, I didn't find it in the supplements either. I also find the colour scheme confusing, especially because there are several components that are only represented in very low proportions in some individuals (so it is hard to identify their affinities) and there are too many shades of the same colour. For example, there is a blue component in NUE001 that I don't seem to identify in any other individual?

We modified the figures and legends to add the details asked by reviewer #2. Figure 2 now displays the PCA generated on North African and West Asian genomes (previous Extended Data Fig. 4) and a newly generated ADMIXTURE plot featuring the 2X genome at $K = 14$,

which is the smallest observed CV error. We hope that the new palette and font size improve the readability of the figures.

We also added the full output of the ADMIXTURE analysis in Extended Data Fig. 4 and 5 (ancient and present-day genomes, respectively).

Supplements

Sup info 4

“The proportion of Zagros or Mesopotamian components (or Caucasus component, also similar to Zagros Neolithic⁴) increase eastward (in the Levant and Anatolia), while the Epipaleolithic Morocco ancestry disappears (Extended Data Fig. 6, Supplementary Data Table 5).” In Extended Data Fig.6, Iran-Neolithic ancestry is depicted, but it is not mentioned in the modelling methods (always Zagros). If it is the same set of samples, use a consistent label.

Correct, we are referring to Zagros and modified Extended Data Fig. 6

“ $74.0 \pm 7.6\%$ Middle Neolithic Moroccan, $21.2 \pm 4.2\%$ Neolithic Mesopotamian, and $4.7 \pm 8.1\%$ Neolithic Levantine ancestries ($p = 0.255$). This alternative model corroborates a scenario where the West Asian-related ancestry detected in the Nuwayrat individual is still most directly derived from Neolithic Mesopotamia and not from the two sources from which it derives, namely Neolithic Levantine or Caucasus/Zagros ancestries”. Do you mean to say that Neo Mesopotamian ancestry derives from both Neo Caucasus and Zagros Neo? If so, please rephrase for clarity.

We clarified L76-79.

But most importantly, why does this corroborate such scenario? This model actually has a higher p-value and would make sense from a migration route perspective.

And also: could this result reflect a bit of overfitting? The 3-source models with Morocco Middle Neolithic, Neolithic Levant, Neolithic Caucasus/Zagros fits the data as well, so I am not totally convinced about the conclusion that Nuwayrat's ancestry derives directly from Mesopotamian Neo. Including both Morocco_MN and Levant_Neolithic/Chalcolithic could have the same effect (overfitting)?

Thanks to the sequencing effort, the newly updated genome allows us to discriminate better between those models. We can now reject the three-source model Morocco_MN, Neolithic Levant and Neolithic Caucasus/Zagros. The two-way model Morocco_MN+Mesopotamia_N has now the highest ranked p-value and the three-sources models are consistent with the Morocco_MN+Mesopotamia_N, including a minor proportion of Neolithic or Chalcolithic Levant, with standard error overlapping with 0. We hope that this new result clarifies the issue raised by reviewer 2.

“This result is consistent with the previous distal model (Extended Data Fig. 6)” This model shows Iran Neolithic (Zagros?) instead of Mesopotamia Neolithic, even though the latter was also used as a distal source, so I don't think consistency can be claimed.

As mentioned in a reply to a previous comment, we removed Mesopotamia_N as a source in the distal model, to use sources with divergent ancestry. Only Caucasus_N and Zagros_N were added despite being similar because 1) the absence of Zagros_N resulted in no model passing $p > 0.05$ for NUE001 and 2) the absence of Caucasus_N resulted in no model

passing $p > 0.05$ for Anatolia_BA. Both populations are necessary in order to effectively model the current set of target populations within the same framework.

“When this reference is favoured to represent the eastern Fertile Crescent ancestry, ~13-16% ancestry is attributed to Neolithic Chalcolithic Levant.” ...Neolithic/Chalcolithic Levant?
The sentence has been removed in light of the new results.

I am surprised to see that when you try to control for overfitting the model, the “Distal model” (Neolithic Levant, Epipaleolithic Morocco and Neolithic Zagros) previously accepted no longer works. I can only imagine that this is caused by populations that are now in the left populations set being rotated of the right (outgroup) set. What are the implications, if that is the case?

Indeed, in the full rotating model, the proximal populations are added in the right/outgroup set to improve discrimination between sources. We can interpret this result as the set of left populations selected in the “distal model” is in an outgroup position compared to some of the proximal populations in the right/outgroup set, forcing more proximal ancestries to be used.

“the inclusion of Bronze Age Levant as an outgroup is expected to result in a lower p-value if the Bronze Age Levant in truth received gene flow from Nuwayrat-related sources.” How would Noth African (related to Morocco_MN) ancestry of Nuwayrat not be observed in Levant_BA if this had happened?

We agree and this paragraph has been extensively modified in light of the new results generated using the 2X coverage genome in Supplementary Information 4 on L124-137. Based on this updated genome, the model Morocco_MN+Levant_BA is also rejected and associated with a slightly lower p-value than Morocco_MN+Mesopotamia_N. We discuss the range of interpretations explaining this result in Supplementary Information 4 on L124-137.

"We find that the Bronze Age Levant can marginally be modelled as a source for Nuwayrat" I think that this result is worth further exploring. Can a recent admixture from BA Levant be completely excluded? Admixture dating (Morocco MN+Mesopotamia N / Morocco MN + Levant BA) could be revealing.

We added the result of admixture dating using DATES (Chintalapati, Patterson and Moorjani 2022) in Supplementary Information 4 (L242-257) and Supplementary Data Table S11. We extensively tried dating admixture in the Nuwayrat genome using Morocco_MN as source 1 and a set of ancient West Asian genomes as source 2 using the newly generated 2X genome and using the imputed data. We could not get an accurate fit for the ancestry covariance decay at $n_{\text{rsmd}} < 0.7$ and Z-scores > 2 , meaning we cannot interpret the admixture dating. We then discuss potential explanations for these results, which are caused by a lack of statistical power to discriminate sources in this single genome and/or multiple admixture events. Moreover, as abovementioned, the full model with Levant_BA is rejected with a lower p-value than the model Morocco_MN + Mesopotamia_N.

“We also detect ancestry from Neolithic Mesopotamia at three archaeological sites (Ebla, Baq’ah, Ashkelon)” ...at three Levantine archaeological sites...

The sentence was modified following the reviewer’s suggestion.

“A part of this ancestry ultimately derives from Caucasus Neolithic-related ancestry in Neolithic Mesopotamia (Extended Data Figure 5)8,10,11, this could explain why Neolithic

Mesopotamia can be a possible source with an increased proportion compared with the best-fit model for Nuwayrat." Unclear, please rephrase.

This sentence was removed because it became obsolete in light of the new results based on the 2X genome.

G. References: appropriate credit to previous work?
References are appropriate and credit previous work.

H. Clarity and context: lucidity of abstract/summary, appropriateness of abstract, introduction and conclusions

The manuscript is clear, easy to follow and well written. Suggested improvements are listed in section F.

Referee #3 (Remarks to the Author):

This is an interesting article and appears well written with good supporting figures, but my review focuses solely on the radiocarbon dates as requested by the Editor, as this is my area of expertise.

The radiocarbon dates are a minor part of this study, and are presented primarily to support the age of the individual whose DNA was analysed. Within the main text of the paper, the combined radiocarbon age (from 3 dates, 2 from teeth dated as part of this study, and one previously published from the femur) is given in Fig. 1d as 2851-2573 cal BCE. However, this is an over-simplification, as calibration of the combined date results in 3 narrower ranges (2851-2809 cal BCE (20.6%), 2747-2727 cal BCE (5.2%) and 2698-2573 cal BCE (69.7%). It is not statistically correct to state that, for example, 4080 ± 30 BP calibrates to 2857-2492 cal BCE at 95.4% probability. Within the main text, this may be acceptable, but the specific date ranges should be included and explained in the supplementary information. Note that the legend to Figure 1 describes it as a summary of genomic data, without mentioning the C14 date. As stated though, the radiocarbon date is not the main focus of this study.

We thank the reviewer for this helpful comment. We highlighted the trimodal probability distribution in Supplementary Information 1, Fig S1.5 and Supplementary Table 2, and made reference to this in the legend of Fig 1. While we acknowledge this limitation, the norm in archaeology is to present the maximum intercept (we use two-sigma, or 95.4%) as a single range, which we also verified with Mike Dee who previously provided assistance with C14 calibration (see further details below).

Within the supplementary data table 2, I would expect the version of calibration software used to calibrate and combine the dates (e.g. OxCal v4.4.4) and the calibration curve (presumably IntCal2020; Reimer et al (2020) to be stated. I recalibrated the dates on the teeth and the femur, and got the same overall ranges (at 95% probability) for the teeth but not for the femur. This suggests that they may have been calibrated using different software and/or datasets (especially if the femur calibration was cited directly from Dee (2016), prior to the release of IntCal20). Ideally, this should be recalibrated for effective comparison.

We thank the reviewer for pointing out the mismatch (the femur calibration was indeed outdated) and updated the calibration and manuscript accordingly. In fact, we recalibrated all dates to ensure consistency, and we further implemented outward rounding following widely applied recommendations for archaeology (see the updated Methods section). We also added the references in the Supplementary Data Table 2.

I could not access either of the references cited for the femur date (refs 64, 65) as the links took me to a Paperpile file that I could not access, and I could not easily find the documents online.

Briefly, those references are “Vanthuyne, B. Early Old Kingdom Rock Circle Cemeteries in the 15th and 16th Nomes of Upper Egypt. A Socio-archaeological Investigation of the Cemeteries in Dayr al-Barshā, Dayr Abū Ḥinnis, Benī Ḥasan al-Shurūq and Nuwayrāt. (KU Leuven, Leuven, 2017)” reporting the femur date and Dee 2016 is the extensive radiocarbon report made for the World Museum of Liverpool, but not available online.

However, I note that Mike Dee (an expert in C14 dating, especially from Ancient Egypt) has assisted with the date combination according to the acknowledgements. I wonder if he might be contacted to clarify the calibrated date range for the femur, and also to advise the authors on inclusion of the individual date ranges rather than stating the continuous calibrated range. As the CN ratios are included for the teeth, it would be helpful if Mike could also provide the CN for the femur (which should be readily available, although I would anticipate it would be in one of the papers cited) - that measurement provides confidence in that date.

We obtained CN data directly from Dr Rachel Wood (with Mike Dee’s permission) who is Director of the Oxford Radiocarbon Accelerator Unit (ORAU). Regarding the individual date ranges, Mike Dee stated in personal communication with L. G-F. “Yes, this aspect of radiocarbon dating is an ongoing challenge. Often we just ignore it and give the earliest and latest dates of the 95% probability calibration range.” But we agree with the reviewer that in this context the non-overlapping probability densities should be acknowledged, so we expand on this aspect of the calibration in Supplementary Information 1 (and see Fig S1.5) as well as in Supplementary Table 2.

I’m not sure why the date on the textile was included (line 6 of the table in Supplementary Data 2) as it is clearly not contemporaneous with the skeletal remains and is not discussed in the paper; I would remove this.

We thank the reviewer for pointing this out and we removed it.

Could 'source' (Column L) be clarified - is this a Pers. Comm with Mike Dee, or should Dee (2016), for example, be cited?

We thank the reviewer for pointing that out and we clarify the reference.

Referee #4 (Remarks to the Author):

Summary of the key results: result push back evidence of aDNA in the Nile valley by a significant margin, capturing the dawn of Egyptian civilization, a key marker in early stages of this civilization. Other rare successful result (n=3 individuals), are much later, less

comprehensive, and do not offer such insights. Although this is based on one individual, the finding is highly significant as (a) aDNA rarely survives in the Nile valley, (b) it captures a period that is key to the emergence of the Ancient Egyptian civilization and (c) the data is more comprehensive and robust than the previous study (much later, see lines 68-69 and 221-222) data. Importantly, for this early period of state formation, it is the first direct evidence of what has been hinted at by indirect genetic proxies such as dental morphology, confirming what these other data have tentatively suggested. It is also, for the first time, the key input from the fertile crescent within a stable North African population group, laying much argument to rest.

Originality and significance: if not novel, please include reference: This is highly significant despite being based on one individual (see above). In view of the rarity of aDNA in the Nile valley, the fact that this is a single individual should not be held against it. It supports what the material culture has suggested, particularly the influx of ideas from the Fertile Crescent into Egypt from the Predynastic period onwards being accompanied with some population movement (e.g. the use of Bull Imagery reveals such an influence in the famous Predynastic tomb 100 decorations, the earliest wall paintings known to date from ancient Egypt, now at the entrance of the Cairo Museum) (i.e. cultural diffusion associated with the movement and settlement of people).

Data & methodology: validity of approach, quality of data, quality of presentation: all appear robust but the aDNA is not my expertise. Other bioarchaeological interpretations are sound.

Appropriate use of statistics and treatment of uncertainties: N/A

Conclusions: robustness, validity, reliability: All appear to be very strong

Suggested improvements: experiments, data for possible revision: perhaps add a clear paragraph (just a few lines) on how the data compares/differs from Sudanese and other 'sub-saharan' populations.

We thank the reviewer for this suggestion and added ancestry modelling of ancient East African genomes using the Nuwayrat genome as a source (L304-311). The extensive report is a section of Supplementary Information 4 L249-266.

I would also expand (again, just a few lines) on the population shift that appears to occur between the Mesolithic and Neolithic (something that is alluded to but would benefit the clarification for the general readers).

We thank the reviewer for this suggestion and added further details in L292-295.

Also perhaps expand/clarify (one or two line) what indirect sources of population biology affinity and relatedness have suggested to date (dental morphology etc...).

We clarify the sentence L75-78.

References: appropriate credit to previous work? yes

Clarity and context: lucidity of abstract/summary, appropriateness of abstract, introduction and conclusions. All appear to be clear and appropriate

We thank all the reviewers for providing constructive and positive feedback on our work. They were deeply appreciated. We specifically reply to reviewers 2 and 4 below who raised further points to discuss.

Referee #1 (Remarks to the Author):

The authors have addressed my concerns and I have no further feedback. I recommend this manuscript for publication.

Referee #2 (Remarks to the Author):

I am overall very happy with the reviewed version of this manuscript. The authors have made significant improvements, not only according to reviewers advice, but also (and commendably) increased the genomic coverage of the Nuwayrat individual to 2x. This enabled improved results, including clearer ancestry modelling results and resolving phenotypic traits, for example. The main text is notably clearer and the discussion of the results much better contextualised and matured.

I noted, however, a couple of minor details that should be fixed before publication.

First and most importantly:

I received a version of the Supplementary Data Tables that has not been updated. For example, supplementary data table 5 still reports distal models using Mesopotamia Neolithic as a source, which as I understand, was not used as a distal source in this reviewed version. Also, still using "Iran_Neolithic" in this table; the ancestry proportions in sup data table 6 do not correspond to those reported in the main text. Please make sure to update this file. I apologize for this issue. It seems that I mistakenly deposited the outdated supplementary data tables.

L259 Please specify that "the initial full qpAdm model" is the initial full qpAdm model for NUE001. It was not completely clear to me, as this sentence follows some modelling results for Bronze Age Levantine sites.

We thank the reviewer for pointing out this miscommunication. This now reads 'However, the initial full qpAdm model extended to include Bronze Age Levant as a potential source can effectively be rejected for the Nuwayrat genome ($p = 0.013$, see Supplementary Information 4).'

L285 "Early Neolithic shared regional ancestry" It should be clearer that regional here means across North Africa. To illustrate this, it would be useful and interesting to refer to the distal qpAdm ancestry modelling results (which are now completely omitted in the main text). E.g., refer to Extended Data Fig 6 here, that shows broadly similar ancestry components for NUE001 and Morocco_MN, in spite of the absence of Zagros Neolithic in Morocco.

We thank the reviewer for pointing out the lack of clarity. This sentence introduces the discussion of the timing of shared regional ancestry between West Asia and Egypt, in other words, when the Mesopotamia-like ancestry entered Egypt. We clarify this sentence as 'Archaeological evidence lends support for Early Neolithic shared regional ancestry between Egypt and West Asia'

L547 "Caucasus_Neolithic wer used as rotating sources" ... *were* used as rotating sources

We thank the reviewer for informing us of this typo and we corrected it.

Referee #4 (Remarks to the Author):

Changes are appropriate and have clarified several elements of the paper, which is now stronger, no further comments.

Referee #5 (Remarks to the Author):

I was asked to check if the authors have answered correctly and sufficiently to the reviewer #3 (the Radiocarbon Dating scholar) points.

They authors answer the points of the reviewer and this would probably be sufficient to accept the paper for publication.

I have another suggestion in order to retrieve more information from the radiocarbon dates obtained and reduce the calibrated range.

The authors could build different Bayesian models considering that:

1. dentine 14C reflects the first 10-15 years of life
2. the person is around 60 years old
3. the femur 14C reflects the last 10-15 years of life.
- 4 use a gap or different gaps and see what kind of calibrated range become most probable.

Although the radiocarbon dating is not the major focus of this paper, yet the authors do mention the possible synchronization with the archaeological and historical record. I did the model and in my opinion it has improved the chronological issue.

Such suggestion is not a "MUST" to get the paper accepted, but it would improve in general the meaning of the paper. Hope this helps.

Elisabetta Boaretto

Thank you for the opportunity to test this suggestion, which in principle is a very good idea. However, we find that a more detailed approximation of femur collagen turnover reveals real gaps that are far smaller than those suggested here, which in turn results in less evident improvements in overall calibration. Moreover, given the uncertainties inherent to precisely estimate the true gap ages, in combination with the uncertainties in the osteological determination of his age-at-death, the C14 calibrations in the current manuscript should be the most robust and assumption-free. We have updated Supp Text 1 with the following text:

“We merged OxA-33186 with two new dates (Beta – 635236 and Beta - 635237) obtained from tooth collagen of two teeth, one of which we generated the majority of genome data from (Supplementary Data Table S2). The combined date (see Methods) yielded a conventional radiocarbon age of 4098 ± 19 BP, which calibrates to 2855-2570 cal BCE (95.4% probability). However, the conventional radiocarbon age intersects with the calibration curve over two so-called wiggles and a weak plateau (i.e. natural variations in atmospheric C14) which results in trimodal probability distribution (Fig S1.5). This means that the maximum intercept range (at 95.4% probability) is conservative.

We attempted to resolve these issues by leveraging the fact that dentine forms over a limited period and does not turn over later in life, meaning that no new collagen is synthesised in those tissues once fully formed. Therefore, C14 measurements on collagen from dentine

represent an average of the period of tissue formation. Conversely, femur collagen is gradually replaced ('turned over'), meaning that a certain proportion of older collagen is continuously replaced with newly synthesised collagen. The collagen in a femur of a 45-60-year-old male should therefore represent an average formation age that is closer to the age of death than the M3 collagen. This age-gap can then be implemented in Bayesian modelling during C14 calibration. The average formation age for M3 collagen is approximately 20 years of age^{13,14}.

However, upon closer review of the collagen turn-over rates in femurs, we find that adult femurs will always contain a substantial proportion of collagen synthesised prior to the age of 20-25¹⁵ such that an age-weighted average in adult femur collagen will always be markedly younger than the age-at-death. Specifically, Hedges et al. (2007)¹⁵ estimated that collagen turn-over rates between the ages of ~10-20 are much higher (~15-25% for the best-fit model) than from ages ~25-80, during which it gradually decreases from ~3-1.5%. Moreover, they also observed up to 30% individual variation in turn-over rates. Therefore, the precise amount of pre-age 20 collagen that is retained at any given age in adulthood is difficult to determine with precision; it is also not known whether there is variability in turn-over rates for age-specific fractions in the total collagen pool. Consequently, since adult femur collagen will always contain a substantial proportion of collagen formed in adolescence and/or before the age of 20, it is not suitable for Bayesian gap modelling; in fact, it is likely that part of the collagen from which the C14 date was derived formed at an earlier age (i.e. is older in C14 years) than the collagen from the M3 tooth.

¹³Berkovitz, B. K. B (ed). *Why Should It Matter How Long Our Ancestors' Teeth Took To Develop?* in *Nothing but the Tooth*. 175-188. (Elsevier: 2013)

¹⁴Putul, M., Konwar, R., Dutta, M., Basumatary, B., Rajbongshi, M. C., Thakuria, K. D., Sarma, B. Assessment of Age at the Stages of the Eruption of Third Molar Teeth among the People of North-Eastern India. *Biomed Research International*. **1**, 9714121 (2021).

¹⁵Hedges, R. E., Clement, J. G., Thomas, C. D., O'connell, T. C. Collagen turnover in the adult femoral mid-shaft: modeled from anthropogenic radiocarbon tracer measurements. *American Journal of Physical Anthropology*. **133**(2):808-16 (2007)."

Referee #6 (Remarks to the Author):

A. This article presents new and exciting genetic data from the Early Dynastic/Old Kingdom. As the authors clearly state, this pushes back our genetic understanding of ancient Egypt significantly. I very much appreciate the multi-method approach, including aDNA, isotopes, osteology, etc. Approximately 20% of this individual's genetic ancestry can be traced to the Near East. While this isn't altogether surprising, given the history of ancient Egypt--it is important to be able to quantify this connection and present empirical evidence for genetic relationships (even at Nuwayrat).

B. Of course, there have been many attempts to study aDNA in ancient Egyptian individuals. However, this has been frequently unsuccessful, due to preservation issues. Thus, the whole-genome data presented here from such an early date (Early Dynastic/Old Kingdom) is truly original and significant.

C. The data and methodology, are all sound. I'm specifically focusing on the osteological approach, but I have no suggestions here. Every method and citation is appropriate and applied accordingly.

D. Yes.

E. The conclusions are appropriate and fitting. I like the added value of the unique burial practice (jar burial), may have played a special role in preserving the aDNA. This, in my opinion, could help guide future aDNA research in the Nile Valley.

F. None.

G. Yes; again, I'm focusing mainly on osteology, but all references are good.

H. Very clear and well contextualized.

Great article--very nice work! and a great team!

Referees' comments

Referee expertise:

Referee #1: population genetics, aDNA

Referee #2: population genetics, aDNA

Referee #3: radiocarbon dating

Referee #4: Egyptology

Referees' comments:

Referee #1 (Remarks to the Author):

Morez et al. present a 1.51X whole genome of an individual from Nuwayrat, Egypt dating to the Early Dynastic-Old Kingdom period. This makes this genome the oldest genomic sequence from Egypt, thus enabling first insights - barring sample size limitation - into the genetic diversity during the early dynastic periods. The manuscript engages primarily with the genomic data and the presented analyses are fairly standard for an ancient DNA paper dealing with a single low coverage genome that precludes more involved population genetics methods. Based on the genomic analyses, the authors report that the sequenced Nuwayrat individual had genetic ancestry related primarily to North African Neolithic and eastern Fertile Crescent groups (best proxy being Neolithic Mesopotamia). The Nuwayrat individual's ancestry may have contributed in part to later Third Intermediate period genomes as well as several present-day Egyptians, though these groups have received substantial and later gene flow from genetic sources related to Bronze Age Levant and ancient East African groups, respectively. Additionally, there are osteological and facial reconstruction analyses included, the former suggesting that despite the burial context hinting at this individual's higher social class they were likely involved in a labor-intensive occupation. Overall, this manuscript pushes past the methodological challenges inherent in a region such as Egypt and generates whole-genome data, which is commendable.

I have been through the genomic analyses presented in this manuscript. I am quite happy with the careful manner in which the data QC and filtering has been performed. I am also satisfied with the descriptions and implementation of the analyses that are feasible given the generated data. The authors have been thoughtful in the qpAdm analytical set-up, which forms the core of the results in this study. I believe they have accounted for the distal and proximal sources with the available aDNA data and show fairly convincingly the sources that are able to model the Nuwayrat individual, in addition to considering alternate models and possibilities of overfitting due to one of the sources being a compound in itself. I do not have any major concerns as such but list a few minor suggestions/questions below:

We thank the reviewer for their positive comments and constructive feedback.

Title: "Whole-genome ancestry" sounds a little awkward to me, perhaps: Whole-genome sequencing of an Old Kingdom Egyptian.

We thank the reviewer for their suggestion regarding the title. However, we feel that 'Whole-genome ancestry' better captures the focus of our study on the ancestral insights derived from the genome, which goes beyond sequencing alone.

Supp Table 1 lists information on 11 libraries but main text talks about 7 extracts - did some extracts have multiple libraries? Can the dropped libraries be marked clearly? Several columns in Supp Table 1 are somewhat unclear - it would be good to have descriptions of the headers.

Two libraries were sequenced several times (information added L148). The information on the two libraries that were discarded is added to L148. We also added a column in Supplementary Data Table 1 to indicate the library ID and the sequencing platform used to generate each sequencing run. We also added the description of the columns in Supplementary Data Table 1 (L498-533).

I was unable to locate a figure corresponding to the ROH analysis.

Based on the more extensively sequenced genome (2X), no ROH segments longer than 4cM were detected. We modified this result on L164-165.

Please standardize the labels for Zagros_N and Iran_N, which are used interchangeably in the qpAdm analysis (e.g. in Extended Data Fig 6 where both are used to indicate source in the figure and text legends).

Thank you. We updated the figure with 'Zagros_Neolithic'.

I was unable to locate the qpAdm results for the tests noted on page 6 (overfitting due to the use of Middle Neolithic Morocco) and page 7 (BA Levantine as sources for the Nuwayrat individual).

Concerning 'overfitting due to Morocco_MN', we detail the qpAdm models in Supplementary Information 4 L138-159. Concerning the tests on 'BA Levantine as sources', we detail the qpAdm models L124-137 of Supplementary Information 4 and point to it in the main text on L259-263. However, in both cases, no model passed the significance threshold, so no results are provided in the Supplementary Data Table.

In the supplementary text describing the f4 ratio for investigating Basal Eurasian, it should note Extended Data Fig 7 instead of 6.

We corrected the text according to the reviewer's comment.

Referee #2 (Remark to the Author):

Does the manuscript have flaws which should prohibit its publication? No.

To my knowledge, the conclusions are original.

The results presented are of immediate interest to geneticists, archaeologists, and anthropologists.

Outstanding features:

Morez and colleagues present new whole genome data from an ancient individual excavated from Nuwayrat, Egypt. The analysed archaeological remains show exceptional preservation for that region and period. Genomic data from ancient Egypt has thus far only been retrieved from three individuals – and even those are of poorer quality and from a more recent period. The preservation of the whole skeleton has allowed for a multidisciplinary analysis of this individual, including a thorough osteological examination that reveals a great deal about this man's life, including his potential occupation, and a facial depiction.

General notes:

I applaud the authors for generating complete genome data for this rare individual, instead of enriching for a SNP panel.

We thank the reviewer for pointing this out, and we completely agree with the added value of complete genome data, reflected in our reference to "Whole-genome" in the title.

The authors caution for the fact that this study is based on a single individual. Small sample sizes are common in ancient DNA studies, especially those based on sites with human remains with documented low molecular preservation, such as Egypt. The assumption for extrapolating such results to a population level is usually based on the individuals' expected representativeness of the general population genetic background. My concerns about the generalisation of the conclusions in this study are based on this individual's potential higher social status, which could hinder that extrapolation. I think that this should be more obviously stressed in the text.

We agree, and thank the reviewer for commending our discussion on the limited sample size. We have extended the text about the limitations due to social status.

The text now reads: "While our analyses are limited to a single Egyptian individual who, based on his relatively high status burial, may not be representative of the general population, [...]" on L370-372.

I am surprised to note the absence of stable isotope analyses such as dietary isotope analysis and/or strontium analysis. The rarity of the specimen would justify investigating these aspects, which would in turn provide a greater depth to the manuscript. These could substantiate the claim of the higher social status of this individual, by possible comparison of

dietary patterns with contemporary Egyptians and perhaps resolve if the individual was born locally or was a migrant.

We agree. In the revised version of the manuscript, we now include an extensive report of several isotope analyses ($\delta^{13}\text{C}$, $\delta^{15}\text{N}$, $\delta^{18}\text{O}$, $^{87}\text{Sr}/^{86}\text{Sr}$) on L130-139 in the main text and in Supplementary Information 5. Overall, the results are consistent with the individual being local.

I would expect, in the context of a culture as well-studied as the ancient Egypt (by many disciplines before archaeogenomics), a more robust and contextualised discussion, including perspectives and background from available literature/evidence (such as archaeological). I think that improving this aspect would round up the paper nicely and make it more interesting to the general audience, considering how popular ancient Egypt is. As an example, the conclusions paragraph is rather dull and adds little to no information to the manuscript.

“These genetic links have ramifications for discerning early Dynastic population history, opening up the possibility that the incorporation of objects, images and ideas from outside Egypt was concomitant with some settlement of people.”

Also, in the Summary “contacts between Egypt and the eastern Fertile Crescent were not limited to objects and imagery”

Could the authors give examples? Is there (archaeological) evidence of objects, images or ideas that were incorporated from the regions that seem to have been connected to Egypt through the movement of people, given the genomic results?

We have expanded sections dealing with wider archaeological data and context and included additional references L52-53, L298-301 and L374-379, in addition to material previously included within the main text (e.g., L58-74, L242-249).

Lastly, and related to the previous point, I am not totally convinced about Mesopotamia Neolithic ancestry being the most parsimonious source of ancestry present in NUE001. On PCA (Extended Data Fig 4), for example, NUE001 is clearly positioned between Morocco MN and Levant Neolithic/Chalcolithic, not towards Mesopotamian Neolithic. Unsupervised ADMIXTURE could be very informative, but unfortunately the figure shown is undecipherable and unhelpful. It is cautioned that “surrounding regions” are also potential sources. However, a rather strong emphasis is put on Neolithic Mesopotamia and I am concerned that this result is based mostly on the criteria implemented for model selection in the qpAdm framework, namely the minimal number of source populations criterium. Noticeably, the model selection criteria used follow recently published guidelines for the application of a qpAdm framework. However, at times two-source models are preferred to three-source models, even when the latter could possibly make more sense from a population history dynamics perspective. This seems particularly relevant here given the complex Fertile Crescent demographic history and the (already) admixed nature of the source populations considered – especially proximal. Or perhaps all that is missing is contextualizing the dynamics behind the arrival of Mesopotamia Neolithic-related ancestry to Egypt – and how/why geographically and chronologically closer populations don’t seem to be better proxies for that ancestry. Please see more specific comments/suggestions below.

We agree with this concern. In the revised version, we fully updated all analyses using newly

generated sequences, now reaching 2X coverage, which increased our statistical power to resolve this result. The new key result is that only a single two-source model passes rejection, which includes Morocco_MN and Mesopotamia_N. In turn, the two three-source models passing rejection confirm the two-source model, with Morocco_MN- and Mesopotamia_N-related ancestries present in similar proportion, with a small addition of Levant_N/ChI (4.7+/- 8.2% or 1.1+/-8.7%). Given these results and that of the distal model (Extended Data Fig 6), we can confirm the presence of ancestry ultimately deriving from the eastern Fertile Crescent. We chose the term 'eastern Fertile Crescent' in our interpretation, instead of Mesopotamia, to emphasise the uncertainty of precisely determining the source population.

Dating the admixture event would be key to answering questions related to the timing and location of the true source here. While we lack the statistical power to date the admixture event confidently, archaeological evidence is consistent with both an older and/or a more recent genetic contact; we discuss both possibilities in the main text (L242-251, L266-303). We agree with reviewer 2 that, despite that our model points toward Mesopotamia Neolithic as the best minority source for the Nuwayrat ancestry (with Morocco_MN the majority source) given the current set of reference populations, 1) the admixed nature of the BA and pre-BA West Asia/North Africa sources is a challenge that makes differentiation between proximate sources difficult and 2) we cannot exclude that unsampled geographically and chronologically closer populations are the most proximal sources. We highlighted this limitation L263-265.

Concerning the PCA interpretation, we would refrain from drawing major conclusions from a PCA alone. First, the position of low-coverage genomes is notoriously imprecise. Then, the Morocco_MN source that makes up ~80% of NUE001's ancestry is likely a proxy for a local Neolithic source. If we imagine that this source carried a slightly higher proportion of Levantine ancestry than Morocco_MN, its position in the PCA space could be shifted towards Paleolithic/Neolithic Levantine, and in this case, the Nuwayrat genome would be aligned on a Mesopotamia/local Neolithic cline.

General comments:

The use of line and page numbers would certainly help the reviewers.

We apologise for this omission and hope that reviewer 2 finds the revised version helpful.

A map with labels of the broader regions mentioned in the text, such as Fertile Crescent, Mesopotamia, etc. would be very helpful (I'm thinking perhaps something as seen in Lazaridis 2022 and Altınışık 2022). A more detailed timeline, including other periods mentioned in the text (e.g. Bronze Age is extensively mentioned) is essential.

These are very good suggestions and we modified Figs. 3 and 4 to incorporate geographic details and a timeline of cultural transition, based on Egyptian civilisation chronology.

A. Summary of the key results

Morez and colleagues present new whole genome data from an ancient individual excavated from Nuwayrat, Egypt. The authors stress the North African- and Fertile Crescent-related ancestry in the studied ancient Egyptian. This finding provides additional and important evidence on the links across North Africa and the Fertile Crescent, that now are shown to go

beyond culture and trade, but also include migration. The unusual preservation of the whole skeleton has allowed for a multidisciplinary analysis of this individual, including a thorough osteological examination that reveals a great deal about this man's occupation and a facial depiction.

B. Originality and significance: if not novel, please include reference

The analysed archaeological remains show exceptional preservation for that region and period. Genomic data from ancient Egypt has thus far only been retrieved from three individuals – and even those are of poorer quality and from a more recent period. The newly generated data will be useful for many future studies.

C. Data & methodology: validity of approach, quality of data, quality of presentation

The authors use robust and well-established methods in ancient DNA research for generating and analysing genomic data from the ancient Egyptian individual through direct shotgun sequencing.

D. Appropriate use of statistics and treatment of uncertainties

All according to well established methods frequently used in ancient DNA analyses.

E. Conclusions: robustness, validity, reliability

The overall conclusions and the findings are sound, although at times lacking a stronger discussion/contextualisation.

F. Suggested improvements: experiments, data for possible revision

I list some suggestions and comments, which I hope the authors find helpful:

Summary

"Nuwayrat (Nuerat, نويرات)." Nuwayrat (Nuerat, نويرات).

We hope you can now find this typo corrected, we could not see it in our original manuscript.

"Most of his genome is represented by North African Neolithic ancestry"

While not incorrect, the multitude of ancestries present in Neolithic North(west) Africa makes this statement ambiguous. In the distal ancestry modelling, the majority of his genome is in fact represented by Levantine ancestry, and truly, so is the "North African Neolithic ancestry" mentioned here (I assume Morocco Middle Neolithic).

Thank you for highlighting this ambiguity. We hope that the modifications on L44-45 are satisfying while keeping the information concise as requested for the abstract section. We discuss in more detail the Middle Neolithic Morocco ancestry in the text (L201-211) and back up the claim that Middle Neolithic Morocco is the current best representative for part of the Nuwayrat ancestry with further testing on potential over-fitting of Middle Neolithic Morocco in Supplementary Information 4 L138-159.

Main text

"The population has been said to be of local origin" based on what kind of evidence, if not archaeological, as that seems to be excluded, based on the next sentence?

This is indeed based on older archaeological evidence. We clarified this on L65-69. However, more recent studies showed a trade connection with the eastern Fertile Crescent. Additionally, these recent studies refer only to the diffusion of material culture. No inference on whether this was associated with demic diffusion could be made.

“Exchange continued through the late 4th millennium BCE with the growing Sumerian civilisation of Mesopotamia” Please specify the kind of exchange meant.

We modified the sentence on L69 to clarify that we are referring to cultural exchange.

“which some maintain can provide clues concerning occupation” some...authors?

We clarified this on L122.

“using a fully rotating model competition approach” It would be useful to include a very brief description of what this means.

We added a short description on L173-179.

“a single two-source model met the significance criteria ($p = 0.207$), which includes $78.4 \pm 4.0\%$ ancestry most closely related to genomes from the Middle Neolithic Moroccan site of Skhirat-Rouazi dated to 4,780-4,230 BCE (Morocco_MN), and the remainder most closely related to genomes from 9,000-8,000 BCE Neolithic Mesopotamia ($21.6 \pm 4.0\%$).” Both source populations have received geneflow from the Levant. Could this be problematic, eventually incurring into overfitting the model?

We shared this concern and detailed further qpAdm models trying to resolve this potential issue in Supplementary Information 4. The most relevant are:

1) L138-159: When replacing *Morocco_MN* with *Morocco_Epipaleolithic* in order to let another potential source for the Levantine ancestry fit more freely, we find that Nuwayrat cannot be modelled at all (no models pass rejection), even when the potential sources include *Levant_BA*. This confirms that part of the Levantine ancestry in Nuwayrat is best derived from a Morocco_MN-like population. As a consequence, Morocco_MN as source 1 should limit the possibility of wrongly picking Mesopotamia_Neolithic as the source for the eastern Fertile Crescent ancestry. Moreover, if another source population from the rotating set would be an equally good or better source for Nuwayrat, and includes the same ancestries as Mesopotamia but in different proportions, we expect this model to pass $p > 0.05$.

2) L124-137: When *Levant_BA* is added to the initial ‘proximal’ model, no model passes rejection. While this could suggest that *Mesopotamia_Neolithic* and *Levant_BA* are equally good proxies for the second source, and thus mutually excluding each other, it is interesting to note that the model Morocco_MN+Mesopotamia_N has a slightly higher ranked p-value than Morocco_MN+Levant_BA.

Overall, in light of those results, and the currently available samples, Morocco_MN+Mesopotamia_N is the best model fitting the Nuwayrat genome. Concerning, the biological interpretation, we cannot formally draw a precise time and location for the source of this ultimately eastern Fertile Crescent ancestry (as discussed on L242-265 of the Main Text).

“pre-Bronze Age genomes are not available for the rest of the Maghreb” Not true for the Maghreb.

We modified the sentence on L211-213.

"from Mota, Ethiopia, or any other individuals in central, eastern or southern Africa (Fig. 2, Extended Data Fig. 5)" I only see Ethiopia 4500BP in Fig2 and also South Africa 2200 BP in extended Data Fig. 5. It would be very interesting to see if there is shared ancestry between NUE001 and pastoral Neolithic East Africans.

We agree with the reviewer and complemented our study with a section on ancient East African modelling using NUE001 (L304-311). A more extensive report is placed in a dedicated section of Supplementary Information 4 L224-241.

"other studies suggesting gene flow from the Mesopotamian and Zagros region into surrounding regions, such as Anatolia, during the Neolithic⁴⁴" A single study is cited.

The text was amended following the reviewer's recommendation to include additional supporting references.

"wider expansion processes and an increase in mobility from the Mesopotamian region, into Egypt as well as Anatolia." It is puzzling for me how such a scenario of mobility from Mesopotamia would not bring Levantine ancestry, admixed into the migrating population on its way to Egypt. Please elaborate. Is an alternative route of expansion (not encompassing the Levant) possible or supported by other lines of evidence?

This is correct. We agree with the reviewer that this information is essential and interesting. One of the main hypothesised trade routes included seafaring through the Mediterranean and the Red Seas, and not through the Sinai Desert. We added this information on L298-301.

"To investigate the presence of Mesopotamian ancestry in Nuwayrat relative to that in Bronze Age contemporaries of the latter, we extended the qpAdm model to adjacent regions." It is not clear to me what was done, please rephrase.

We rewrote the sentence and this now reads: "Related movements may have introduced the Mesopotamian-like ancestry more recently in Egypt. We tested this by applying the same full qpAdm model to target groups from Bronze Age Anatolia and Levant" on L250-254. We hope that the reviewer finds it clearer.

"We detect ancestry from Neolithic Mesopotamia at three sites (Ebla, Baq'ah, Ashkelon), in proportions (41.8-54.8%) exceeding those in the Nuwayrat genome. However, like previous studies, we also find that all modelled Levantine Bronze Age groups trace 18.7-79.8% of their ancestry to previous Neolithic/Chalcolithic Levantine peoples^{46,48-51}. These results do not provide any evidence that the Mesopotamian ancestry in Nuwayrat arrived directly from Bronze Age Levantine groups, but we cannot exclude that intermediaries exist that are yet unsampled." I miss the rationale behind this paragraph. Why is the existence of Neolithic/Chalcolithic Levant ancestry in later (BA) Levantine groups conflicting with the presence of Mesopotamia Neolithic ancestry? Also: wouldn't evidence that Mesopotamian ancestry could have arrived through BA Levant be provided by an accepted model including BA Levant and Morocco_MN? That is not what was tested here.

We agree with the reviewer and we reworked this paragraph in order to clarify our claims and emphasize the limitations of our model given the admixed nature of the sources. Please find the amendments on L250-265.

“Like Middle Neolithic Morocco, this might indicate early shared regional ancestry.” Unclear. Ancestry from the eastern Fertile Crescent was not found in Morocco. Please rephrase. We rephrased the sentence on L284-285.

“Bronze Age Caucasus, which is similar to the Bronze Age Levant ancestry detected in the Third Intermediate Period individuals⁴⁷. Other Bronze Age West Asian populations from the Levant or Anatolia cannot be rejected as potential sources when Bronze Age Caucasus is not used as a source.” Is there evidence (archaeological, for example) of migration/contacts with BA Caucasus after the third intermediate period? Do the authors mean to suggest a possible direct migration from the Caucasus into Egypt? I think these results lack a better discussion. given the justification that BA Levant ancestry is similar to the best fitting Caucasus BA ancestry. In fact, this is a good example of how I think the model selection criteria (while necessary) results in “best models” that should be better contextualised. We regenerated these models using the more deeply sequenced version of NUC001 and increased the number of included SNPs. The updated models favour Levant_BA instead of Caucasus_BA, which is consistent with the observed increase of Levant_BA ancestry in the Third Intermediate genomes.

“Moreover, we note that there is a substantial diversity in ancestry across Egypt, as ~20% of present-day Egyptian genomes included here did not fit the model described above.” This heterogeneity could reflect some geographic structure within present-day Egypt. But there is no information on the regional origin (within Egypt) of these samples, and I think this is relevant information, omitted here.

We also think that investigating potential geographical structure would be extremely interesting. Unfortunately, this information is not publicly available from the dataset used in this study, i.e. Human Origin and Pagani *et al.* 2015.

Methods

“each skeletal element” different teeth? I am curious as for why the petrous bones were not used, if available.

The specific tooth used is detailed in Supplementary Data Table 1. We decided not to use petrous bones because the teeth were both numerous and relatively well-preserved. In many cases, if teeth are macroscopically well preserved the cementum-enriched root tips can yield DNA comparable to that from petrous bones (see Hansen *et al.* 2017 <https://doi.org/10.1371/journal.pone.0170940>). Given the high value of petrous bones for other bioarchaeological analyses, we chose to preserve them when alternative samples, like teeth, were available.

In all truth, I am (very) unfamiliar with the literature cited, but I thought this was worth noting: it was only after reading the methods and supplements that it became evident to me that the results of the osteological analyses results are novel and reported first-hand in this paper (and not a collection of results published elsewhere). I think it has to do with the reference placement in the main text – articles describing the methods are cited after the results.

These results are astonishing, so do make sure to take clear credit!

This acknowledgement is deeply appreciated and we thank the reviewer for supporting further efforts to credit the authors' work. We added a modification on L114 to clarify this.

There are 100 Egyptian individuals sequenced in the cited article (Pagani 2015), not 216. We thank the reviewer for pointing out this mistake in the text, which has been amended.

Importantly (maybe for several of my comments): What is the goal with using both proximal and distal sources for ancestry modelling? This should be better described.

Supplementary Information 4: L31-42 and L52-58 for a description of how and why the different model setups were chosen.

In a nutshell, while a distal model helps get a coarse idea of which 'main' divergent ancestries are present in NUE001, the idea behind keeping the distal populations and adding them to the full one allows us to gradually build a more fine-detailed model, while conserving clarity for how deeper admixture events may have shaped ancestries of the oldest genetic sample from Egypt.

We now refer to those models as 'full' instead of 'proximal' to help clarify their aim.

"We estimated ancestry proportions of the two Third Intermediate Period Egyptians" Three individuals?

There are two Third Intermediate Period Egyptian genomes and one Ptolemaic period genome from Schuenneman et al. 2017. In this study, we re-analysed only the two Third Intermediate Period Egyptians because of the noticeable contamination subsequently found in the Ptolemaic genome (contamination on the X-chromosome = 2.2-12.4%, as reported in the Allen Ancient DNA Repository metadata).

Basal Eurasian ancestry: please define CHS, ESN (also in Extended Data Fig. 7).

The definitions of CHS and ESN have been added in the legend of Extended Data Fig 7.

Conclusions

"...these results reveal ancestry links to North African-related groups" ...North African groups

Corrected L373.

Figures

Maybe not a comment to the figures per se, but I find it very confusing that some sources seem to be arbitrarily chosen as both distal and proximal.

E.g. Anatolia_BA is modelled in Fig. 2b and in Extended fig 6, with vastly different results shown – but several of the populations used as sources are common between the two figures (including those that are accepted in the other figure), even if the models are intended as proximal and distal, respectively. Is this useful? How can it be interpreted?

We agree with the reviewer that the word 'proximal' here is misleading. We replaced this with 'full' model. While the distal model intends to estimate the presence and proportion of a set of 'deep' lineages linked to the Epipaleolithic/Neolithic, the full model build-up from the distal one, including more proximal and related sources to get a refined understanding of the more direct sources. Since we do not have any older representative for Egypt than the

Nuwayrat genome, these older samples are key to investigating deep population movements in Egypt. We also removed *Mesopotamia_Neolithic* from the distal model, given its admixed nature and found highly consistent results for *Anatolia_BA* in the distal and full model

Fig. 3a: At first I thought that each plot represented one individual. Other models have been presented based on their (best) ranking, why was that not done here? And mostly: considering that NUE001 is a mixture of Morocco_MN and Mesopotamia_N, why including all these (redundant) sources? This plot makes it look like these individuals/populations could have very distinct demographic histories, or that a complete population replacement could have happened between NUE001 and the third intermediate period.

We added the result of the suggested model in the Supplementary Information 4 L215-223. Briefly, the rotating qpAdm models without *Morocco_MN* and *Mesopotamia_N* result in no model passing $p > 0.05$. We interpret this as another line of evidence showing that *Morocco_MN* ancestry was likely widespread in Egypt and not only restricted to the Nuwayrat genome so that later genomes from Northern Egypt received it. Slightly different mixture proportions of these two ancestries across Egypt can result in *Morocco_MN* being a better fit than NUE001.

We agree that for consistency, only the model selected as best fit should be displayed.

I suggest trying to simplify the qpAdm modelling, using a more consistent scheme – of source populations and respective colour – throughout the article.

We hope that the previous replies to the reviewer's comment helped clarify the choice of source populations in the model.

Fig 2c and SupData9: How did Levant_BA populations score in the f4 statistics? I understand it is a good principle to observe chronological stratification of samples, but it would be informative. I don't think that a scenario of very recent admixture could be discarded.

We added Levant_BA to the f4-statistics. $f_4(\text{NUE001, MoroccoMN; LevantBA, JuHoan})$ derives significantly from 0 ($Z > 2$), but does not deviate as much as when MesopotamiaN is placed as X ($Z > 3$).

Fig 2a PCA is very small (I can't identify Caucasian or South East Asian individuals, even though these are listed in the legend) and shows exactly what is expected in a PCA with world-wide populations, so not very revealing. I think Extended Data Fig 4 is much more informative. Consider displaying it in the main text instead.

We agree with the reviewer and modified the Fig. 2 accordingly.

Extended Data Fig. 5: How many K are represented in this figure? And why was this the chosen K to display? Is this the full set of populations used for ADMIXTURE analysis?

Please present all the K of the full analysis, I didn't find it in the supplements either. I also find the colour scheme confusing, especially because there are several components that are only represented in very low proportions in some individuals (so it is hard to identify their affinities) and there are too many shades of the same colour. For example, there is a blue component in NUE001 that I don't seem to identify in any other individual?

We modified the figures and legends to add the details asked by reviewer #2. Figure 2 now displays the PCA generated on North African and West Asian genomes (previous Extended

Data Fig. 4) and a newly generated ADMIXTURE plot featuring the 2X genome at $K = 14$, which is the smallest observed CV error. We hope that the new palette and font size improve the readability of the figures.

We also added the full output of the ADMIXTURE analysis in Extended Data Fig. 4 and 5 (ancient and present-day genomes, respectively).

Supplements

Sup info 4

“The proportion of Zagros or Mesopotamian components (or Caucasus component, also similar to Zagros Neolithic4) increase eastward (in the Levant and Anatolia), while the Epipaleolithic Morocco ancestry disappears (Extended Data Fig. 6, Supplementary Data Table 5).” In Extended Data Fig.6, Iran-Neolithic ancestry is depicted, but it is not mentioned in the modelling methods (always Zagros). If it is the same set of samples, use a consistent label.

Correct, we are referring to Zagros and modified Extended Data Fig. 6

“ $74.0 \pm 7.6\%$ Middle Neolithic Moroccan, $21.2 \pm 4.2\%$ Neolithic Mesopotamian, and $4.7 \pm 8.1\%$ Neolithic Levantine ancestries ($p = 0.255$). This alternative model corroborates a scenario where the West Asian-related ancestry detected in the Nuwayrat individual is still most directly derived from Neolithic Mesopotamia and not from the two sources from which it derives, namely Neolithic Levantine or Caucasus/Zagros ancestries”. Do you mean to say that Neo Mesopotamian ancestry derives from both Neo Caucasus and Zagros Neo? If so, please rephrase for clarity.

We clarified L76-79.

But most importantly, why does this corroborate such scenario? This model actually has a higher p-value and would make sense from a migration route perspective.

And also: could this result reflect a bit of overfitting? The 3-source models with Morocco Middle Neolithic, Neolithic Levant, Neolithic Caucasus/Zagros fits the data as well, so I am not totally convinced about the conclusion that Nuwayrat’s ancestry derives directly from Mesopotamian Neo. Including both Morocco_MN and Levant_Neolithic/Chalcolithic could have the same effect (overfitting)?

Thanks to the sequencing effort, the newly updated genome allows us to discriminate better between those models. We can now reject the three-source model Morocco_MN, Neolithic Levant and Neolithic Caucasus/Zagros. The two-way model Morocco_MN+Mesopotamia_N has now the highest ranked p-value and the three-sources models are consistent with the Morocco_MN+Mesopotamia_N, including a minor proportion of Neolithic or Chalcolithic Levant, with standard error overlapping with 0. We hope that this new result clarifies the issue raised by reviewer 2.

“This result is consistent with the previous distal model (Extended Data Fig. 6)” This model shows Iran Neolithic (Zagros?) instead of Mesopotamia Neolithic, even though the latter was also used as a distal source, so I don’t think consistency can be claimed.

As mentioned in a reply to a previous comment, we removed Mesopotamia_N as a source in the distal model, to use sources with divergent ancestry. Only Caucasus_N and Zagros_N were added despite being similar because 1) the absence of Zagros_N resulted in no model

passing $p > 0.05$ for NUE001 and 2) the absence of Caucasus_N resulted in no model passing $p > 0.05$ for Anatolia_BA. Both populations are necessary in order to effectively model the current set of target populations within the same framework.

“When this reference is favoured to represent the eastern Fertile Crescent ancestry, ~13-16% ancestry is attributed to Neolithic Chalcolithic Levant.” ...Neolithic/Chalcolithic Levant?
The sentence has been removed in light of the new results.

I am surprised to see that when you try to control for overfitting the model, the “Distal model” (Neolithic Levant, Epipaleolithic Morocco and Neolithic Zagros) previously accepted no longer works. I can only imagine that this is caused by populations that are now in the left populations set being rotated of the right (outgroup) set. What are the implications, if that is the case?

Indeed, in the full rotating model, the proximal populations are added in the right/outgroup set to improve discrimination between sources. We can interpret this result as the set of left populations selected in the “distal model” is in an outgroup position compared to some of the proximal populations in the right/outgroup set, forcing more proximal ancestries to be used.

“the inclusion of Bronze Age Levant as an outgroup is expected to result in a lower p-value if the Bronze Age Levant in truth received gene flow from Nuwayrat-related sources.” How would North African (related to Morocco_MN) ancestry of Nuwayrat not be observed in Levant_BA if this had happened?

We agree and this paragraph has been extensively modified in light of the new results generated using the 2X coverage genome in Supplementary Information 4 on L124-137. Based on this updated genome, the model Morocco_MN+Levant_BA is also rejected and associated with a slightly lower p-value than Morocco_MN+Mesopotamia_N. We discuss the range of interpretations explaining this result in Supplementary Information 4 on L124-137.

“We find that the Bronze Age Levant can marginally be modelled as a source for Nuwayrat” I think that this result is worth further exploring. Can a recent admixture from BA Levant be completely excluded? Admixture dating (Morocco MN+Mesopotamia N / Morocco MN + Levant BA) could be revealing.

We added the result of admixture dating using DATES (Chintalapati, Patterson and Moorjani 2022) in Supplementary Information 4 (L242-257) and Supplementary Data Table S11. We extensively tried dating admixture in the Nuwayrat genome using Morocco_MN as source 1 and a set of ancient West Asian genomes as source 2 using the newly generated 2X genome and using the imputed data. We could not get an accurate fit for the ancestry covariance decay at $nrsmd < 0.7$ and Z-scores > 2 , meaning we cannot interpret the admixture dating. We then discuss potential explanations for these results, which are caused by a lack of statistical power to discriminate sources in this single genome and/or multiple admixture events. Moreover, as abovementioned, the full model with Levant_BA is rejected with a lower p-value than the model Morocco_MN + Mesopotamia_N.

“We also detect ancestry from Neolithic Mesopotamia at three archaeological sites (Ebla, Baq’ah, Ashkelon)” ...at three Levantine archaeological sites...
The sentence was modified following the reviewer’s suggestion.

“A part of this ancestry ultimately derives from Caucasus Neolithic-related ancestry in

Neolithic Mesopotamia (Extended Data Figure 5)8,10,11, this could explain why Neolithic Mesopotamia can be a possible source with an increased proportion compared with the best-fit model for Nuwayrat.” Unclear, please rephrase.

This sentence was removed because it became obsolete in light of the new results based on the 2X genome.

G. References: appropriate credit to previous work?

References are appropriate and credit previous work.

H. Clarity and context: lucidity of abstract/summary, appropriateness of abstract, introduction and conclusions

The manuscript is clear, easy to follow and well written. Suggested improvements are listed in section F.

Referee #3 (Remarks to the Author):

This is an interesting article and appears well written with good supporting figures, but my review focuses solely on the radiocarbon dates as requested by the Editor, as this is my area of expertise.

The radiocarbon dates are a minor part of this study, and are presented primarily to support the age of the individual whose DNA was analysed. Within the main text of the paper, the combined radiocarbon age (from 3 dates, 2 from teeth dated as part of this study, and one previously published from the femur) is given in Fig. 1d as 2851-2573 cal BCE. However, this is an over-simplification, as calibration of the combined date results in 3 narrower ranges (2851-2809 cal BCE (20.6%), 2747-2727 cal BCE (5.2%) and 2698-2573 cal BCE (69.7%). It is not statistically correct to state that, for example, 4080 ± 30 BP calibrates to 2857-2492 cal BCE at 95.4% probability. Within the main text, this may be acceptable, but the specific date ranges should be included and explained in the supplementary information. Note that the legend to Figure 1 describes it as a summary of genomic data, without mentioning the C14 date. As stated though, the radiocarbon date is not the main focus of this study.

We thank the reviewer for this helpful comment. We highlighted the trimodal probability distribution in Supplementary Information 1, Fig S1.5 and Supplementary Table 2, and made reference to this in the legend of Fig 1. While we acknowledge this limitation, the norm in archaeology is to present the maximum intercept (we use two-sigma, or 95.4%) as a single range, **SENTENCE REDACTED**

Within the supplementary data table 2, I would expect the version of calibration software used to calibrate and combine the dates (e.g. OxCal v4.4.4) and the calibration curve (presumably IntCal2020; Reimer et al (2020) to be stated. I recalibrated the dates on the teeth and the femur, and got the same overall ranges (at 95% probability) for the teeth but not for the femur. This suggests that they may have been calibrated using different software and/or datasets (especially if the femur calibration was cited directly from Dee (2016), prior to the release of IntCal20). Ideally, this should be recalibrated for effective comparison.

We thank the reviewer for pointing out the mismatch (the femur calibration was indeed outdated) and updated the calibration and manuscript accordingly. In fact, we recalibrated all dates to ensure consistency, and we further implemented outward rounding following widely applied recommendations for archaeology (see the updated Methods section). We also added the references in the Supplementary Data Table 2.

I could not access either of the references cited for the femur date (refs 64, 65) as the links took me to a Paperpile file that I could not access, and I could not easily find the documents online.

Briefly, those references are “Vanthuyne, B. Early Old Kingdom Rock Circle Cemeteries in the 15th and 16th Nomes of Upper Egypt. A Socio-archaeological Investigation of the Cemeteries in Dayr al-Barshā, Dayr Abū Ḥinnis, Benī Ḥasan al-Shurūq and Nuwayrāt. (KU Leuven, Leuven, 2017)” reporting the femur date and Dee 2016 is the extensive radiocarbon report made for the World Museum of Liverpool, but not available online.

PARAGRAPHS REDACTED

I'm not sure why the date on the textile was included (line 6 of the table in Supplementary Data 2) as it is clearly not contemporaneous with the skeletal remains and is not discussed in the paper; I would remove this.

We thank the reviewer for pointing this out and we removed it.

Could 'source' (Column L) be clarified - is this a Pers. Comm with Mike Dee, or should Dee (2016), for example, be cited?

We thank the reviewer for pointing that out and we clarify the reference.

Referee #4 (Remarks to the Author):

Summary of the key results: result push back evidence of aDNA in the Nile valley by a significant margin, capturing the dawn of Egyptian civilization, a key marker in early stages of this civilization. Other rare successful result (n=3 individuals), are much later, less comprehensive, and do not offer such insights. Although this is based on one individual, the finding is highly significant as (a) aDNA rarely survives in the Nile valley, (b) it captures a period that is key to the emergence of the Ancient Egyptian civilization and (c) the data is more comprehensive and robust than the previous study (much later, see lines 68-69 and 221-222) data. Importantly, for this early period of state formation, it is the first direct evidence of what has been hinted at by indirect genetic proxies such as dental morphology, confirming what these other data have tentatively suggested. It is also, for the first time, the key input from the fertile crescent within a stable North African population group, laying much arguments to rest.

Originality and significance: if not novel, please include reference: This is highly significant despite being based on one individual (see above). In view of the rarity of aDNA in the Nile valley, the fact that this is a single individual should not be held against it. It supports what the material culture has suggested, particularly the influx of ideas from the Fertile Crescent

into Egypt from the Predynastic period onwards being accompanied with some population movement (e.g. the use of Bull Imagery reveals such an influence in the famous Predynastic tomb 100 decorations, the earliest wall paintings known to date from ancient Egypt, now at the entrance of the Cairo Museum) (i.e. cultural diffusion associated with the movement and settlement of people).

Data & methodology: validity of approach, quality of data, quality of presentation: all appear robust but the aDNA is not my expertise. Other bioarchaeological interpretations are sound.

Appropriate use of statistics and treatment of uncertainties: N/A

Conclusions: robustness, validity, reliability: All appear to be very strong

Suggested improvements: experiments, data for possible revision: perhaps add a clear paragraph (just a few lines) on how the data compares/differs from Sudanese and other 'sub-saharan' populations.

We thank the reviewer for this suggestion and added ancestry modelling of ancient East African genomes using the Nuwayrat genome as a source (L304-311). The extensive report is a section of Supplementary Information 4 L249-266.

I would also expand (again, just a few lines) on the population shift that appears to occur between the Mesolithic and Neolithic (something that is alluded to but would benefit the clarification for the general readers).

We thank the reviewer for this suggestion and added further details in L292-295.

Also perhaps expand/clarify (one or two line) what indirect sources of population biology affinity and relatedness have suggested to date (dental morphology etc...).

We clarify the sentence L75-78.

References: appropriate credit to previous work? yes

Clarity and context: lucidity of abstract/summary, appropriateness of abstract, introduction and conclusions. All appear to be clear and appropriate

We thank all the reviewers for providing constructive and positive feedback on our work. They were deeply appreciated. We specifically reply to reviewers 2 and 4 below who raised further points to discuss.

Referee #1 (Remarks to the Author):

The authors have addressed my concerns and I have no further feedback. I recommend this manuscript for publication.

Referee #2 (Remarks to the Author):

I am overall very happy with the reviewed version of this manuscript. The authors have made significant improvements, not only according to reviewers advice, but also (and commendably) increased the genomic coverage of the Nuwayrat individual to 2x. This enabled improved results, including clearer ancestry modelling results and resolving phenotypic traits, for example. The main text is notably clearer and the discussion of the results much better contextualised and matured.

I noted, however, a couple of minor details that should be fixed before publication.

First and most importantly:

I received a version of the Supplementary Data Tables that has not been updated. For example, supplementary data table 5 still reports distal models using Mesopotamia Neolithic as a source, which as I understand, was not used as a distal source in this reviewed version. Also, still using "Iran_Neolithic" in this table; the ancestry proportions in sup data table 6 do not correspond to those reported in the main text. Please make sure to update this file.

I apologize for this issue. It seems that I mistakenly deposited the outdated supplementary data tables.

L259 Please specify that "the initial full qpAdm model" is the initial full qpAdm model for NUE001. It was not completely clear to me, as this sentence follows some modelling results for Bronze Age Levantine sites.

We thank the reviewer for pointing out this miscommunication. This now reads 'However, the initial full qpAdm model extended to include Bronze Age Levant as a potential source can effectively be rejected for the Nuwayrat genome ($p = 0.013$, see Supplementary Information 4).'

L285 "Early Neolithic shared regional ancestry" It should be clearer that regional here means across North Africa. To illustrate this, it would be useful and interesting to refer to the distal qpAdm ancestry modelling results (which are now completely omitted in the main text). E.g., refer to Extended Data Fig 6 here, that shows broadly similar ancestry components for NUE001 and Morocco_MN, in spite of the absence of Zagros Neolithic in Morocco.

We thank the reviewer for pointing out the lack of clarity. This sentence introduces the discussion of the timing of shared regional ancestry between West Asia and Egypt, in other words, when the Mesopotamia-like ancestry entered Egypt. We clarify this sentence as 'Archaeological evidence lends support for Early Neolithic shared regional ancestry between Egypt and West Asia'

L547 "Caucasus_Neolithic wer used as rotating sources" ... *were* used as rotating sources

We thank the reviewer for informing us of this typo and we corrected it.

Referee #4 (Remarks to the Author):

Changes are appropriate and have clarified several elements of the paper, which is now stronger, no further comments.

Referee #5 (Remarks to the Author):

I was asked to check if the authors have answered correctly and sufficiently to the reviewer #3 (the Radiocarbon Dating scholar) points.

They authors answer the points of the reviewer and this would probably be sufficient to accept the paper for publication.

I have another suggestion in order to retrieve more information from the radiocarbon dates obtained and reduce the calibrated range.

The authors could build different Bayesian models considering that:

1. dentine 14C reflects the first 10-15 years of life
2. the person is around 60 years old
3. the femur 14C reflects the last 10-15 years of life.
- 4 use a gap or different gaps and see what kind of calibrated range become most probable.

Although the radiocarbon dating is not the major focus of this paper, yet the authors do mention the possible synchronization with the archaeological and historical record. I did the model and in my opinion it has improved the chronological issue.

Such suggestion is not a "MUST" to get the paper accepted, but it would improve in general the meaning of the paper. Hope this helps.

Elisabetta Boaretto

Thank you for the opportunity to test this suggestion, which in principle is a very good idea. However, we find that a more detailed approximation of femur collagen turnover reveals real gaps that are far smaller than those suggested here, which in turn results in less evident improvements in overall calibration. Moreover, given the uncertainties inherent to precisely estimate the true gap ages, in combination with the uncertainties in the osteological determination of his age-at-death, the C14 calibrations in the current manuscript should be the most robust and assumption-free. We have updated Supp Text 1 with the following text:

"We merged OxA-33186 with two new dates (Beta – 635236 and Beta - 635237) obtained from tooth collagen of two teeth, one of which we generated the majority of genome data from (Supplementary Data Table S2). The combined date (see Methods) yielded a conventional radiocarbon age of 4098 ± 19 BP, which calibrates to 2855-2570 cal BCE (95.4% probability). However, the conventional radiocarbon age intersects with the calibration curve over two so-called wiggles and a weak plateau (i.e. natural variations in atmospheric C14) which results in trimodal probability distribution (Fig S1.5). This means that the maximum intercept range (at 95.4% probability) is conservative.

We attempted to resolve these issues by leveraging the fact that dentine forms over a limited period and does not turn over later in life, meaning that no new collagen is synthesised in those tissues once fully formed. Therefore, C14 measurements on collagen from dentine

represent an average of the period of tissue formation. Conversely, femur collagen is gradually replaced ('turned over'), meaning that a certain proportion of older collagen is continuously replaced with newly synthesised collagen. The collagen in a femur of a 45-60-year-old male should therefore represent an average formation age that is closer to the age of death than the M3 collagen. This age-gap can then be implemented in Bayesian modelling during C14 calibration. The average formation age for M3 collagen is approximately 20 years of age^{13,14}.

However, upon closer review of the collagen turn-over rates in femurs, we find that adult femurs will always contain a substantial proportion of collagen synthesised prior to the age of 20-25¹⁵ such that an age-weighted average in adult femur collagen will always be markedly younger than the age-at-death. Specifically, Hedges et al. (2007)¹⁵ estimated that collagen turn-over rates between the ages of ~10-20 are much higher (~15-25% for the best-fit model) than from ages ~25-80, during which it gradually decreases from ~3-1.5%. Moreover, they also observed up to 30% individual variation in turn-over rates. Therefore, the precise amount of pre-age 20 collagen that is retained at any given age in adulthood is difficult to determine with precision; it is also not known whether there is variability in turn-over rates for age-specific fractions in the total collagen pool. Consequently, since adult femur collagen will always contain a substantial proportion of collagen formed in adolescence and/or before the age of 20, it is not suitable for Bayesian gap modelling; in fact, it is likely that part of the collagen from which the C14 date was derived formed at an earlier age (i.e. is older in C14 years) than the collagen from the M3 tooth.

¹³Berkovitz, B. K. B (ed). *Why Should It Matter How Long Our Ancestors' Teeth Took To Develop?* in *Nothing but the Tooth*. 175-188. (Elsevier: 2013)

¹⁴Putul, M., Konwar, R., Dutta, M., Basumatary, B., Rajbongshi, M. C., Thakuria, K. D., Sarma, B. Assessment of Age at the Stages of the Eruption of Third Molar Teeth among the People of North-Eastern India. *Biomed Research International*. **1**, 9714121 (2021).

¹⁵Hedges, R. E., Clement, J. G., Thomas, C. D., O'connell, T. C. Collagen turnover in the adult femoral mid-shaft: modeled from anthropogenic radiocarbon tracer measurements. *American Journal of Physical Anthropology*. **133**(2):808-16 (2007)."

Referee #6 (Remarks to the Author):

A. This article presents new and exciting genetic data from the Early Dynastic/Old Kingdom. As the authors clearly state, this pushes back our genetic understanding of ancient Egypt significantly. I very much appreciate the multi-method approach, including aDNA, isotopes, osteology, etc. Approximately 20% of this individual's genetic ancestry can be traced to the Near East. While this isn't altogether surprising, given the history of ancient Egypt--it is important to be able to quantify this connection and present empirical evidence for genetic relationships (even at Nuwayrat).

B. Of course, there have been many attempts to study aDNA in ancient Egyptian individuals. However, this has been frequently unsuccessful, due to preservation issues. Thus, the whole-genome data presented here from such an early date (Early Dynastic/Old Kingdom) is truly original and significant.

C. The data and methodology, are all sound. I'm specifically focusing on the osteological approach, but I have no suggestions here. Every method and citation is appropriate and applied accordingly.

D. Yes.

E. The conclusions are appropriate and fitting. I like the added value of the unique burial practice (jar burial), may have played a special role in preserving the aDNA. This, in my opinion, could help guide future aDNA research in the Nile Valley.

F. None.

G. Yes; again, I'm focusing mainly on osteology, but all references are good.

H. Very clear and well contextualized.

Great article--very nice work! and a great team!